# Widespread and increased drilling of wells into fossil aquifers in the USA

Merhawi GebreEgziabher ⬤ [1✉], Scott Jasechko ⬤ [1] & Debra Perrone ⬤ [2]

Most stored groundwater is 'fossil' in its age, having been under the ground for more than ~12 thousand years. Mapping where wells tap fossil aquifers is relevant for water quality and quantity management. Nevertheless, the prevalence of wells that tap fossil aquifers is not known. Here we show that wells that are sufficiently deep to tap fossil aquifers are widespread, though they remain outnumbered by shallower wells in most areas. Moreover, the proportion of newly drilled wells that are deep enough to tap fossil aquifers has increased over recent decades. However, this widespread and increased drilling of wells into fossil aquifers is not necessarily associated with groundwater depletion, emphasizing that the presence of fossil groundwater does not necessarily indicate a non-renewable water supply. Our results highlight the importance of safeguarding fossil groundwater quality and quantity to meet present and future water demands.

[1] Bren School of Environmental Science and Management, University of California, Santa Barbara, California, CA 93106, USA. [2] Environmental Studies Program, University of California, Santa Barbara, California, CA 93106, USA. ✉email: gebremichael@ucsb.edu

Fossil groundwater—defined as groundwater that has been underground for more than 12 thousand years—likely comprises more than half of global groundwater stored within 1000 m of the land surface[1]. Our current understanding of fossil groundwater distributions is based primarily on well water radioisotope measurements. Radioisotope measurements have identified fossil groundwater in over one hundred aquifers around the globe, including the Nubian Sandstone Aquifer System (Chad, Egypt, Libya, Sudan[2]), the Karoo Aquifer (South Africa[3]), the Paris Basin (France[4]), the Chalk Aquifer (United Kingdom[5]), the Great Artesian Basin (Australia[6]), and the North China Plain (China[7]). Fossil groundwater expectedly exists in portions of most, if not all, aquifer systems, even those where adequate radioisotope data are lacking to confirm the presence of fossil groundwater.

Identifying places where fossil groundwater withdrawals are common or increasing over time is important because older and younger well waters often have different susceptibilities to contaminants, including fluoride (e.g., Brazil's Botucatu Aquifer[8]), arsenic (Mexico's Comarca Lagunera Granular Aquifer[9]), salinity (Tunisia's Sfax Basin[10]) and nitrate (e.g., California's Central Valley[11]). Despite the importance of identifying wells that pump fossil groundwater for understanding contamination risk, little is known about the prevalence of wells that pump fossil groundwater. One reason for this is that well water radioisotope measurements and continental-scale well construction depth data are not systematically collected, hindering comparison.

Understanding where wells access fossil groundwater has implications beyond well water quality assessments. First, fossil groundwater can capture human interest, garnering intrinsic and economic values beyond those ascribed to younger water (Supplementary Note 7). Second, fossil groundwater that discharges at springs or into lowland streams can play a critical role in sustaining vulnerable ecosystems[12]. Third, mapping wells that tap fossil aquifers can enable a better understanding of the prevalence of communities that rely on fossil groundwater resources[13]. Although multiple studies have commented on the sustainability of fossil groundwater use[14–18], the spatiotemporal patterns of fossil groundwater use and groundwater depletion remain unclear, partly because of a lack of geospatial data with locally relevant information for aquifer boundaries.

The United States has sufficient data to evaluate both the spatiotemporal patterns of wells deep enough to tap fossil groundwater, and the spatial relationships between fossil groundwater use and groundwater level changes over time. Compared to other countries, the US has relatively dense well water radioisotope measurements[1]. Similarly, the quality of US well construction depth data is relatively good[19]. These continental-scale datasets have never been merged at continental-scale, but provide an opportunity to analyze spatiotemporal patterns of wells of sufficient depth to tap fossil groundwater if merged with a new geospatial dataset that has locally relevant information about aquifers in the US (Fig. 1a). The lack of locally relevant aquifer geospatial data has constrained the number of aquifer systems evaluated in previous studies[1,19].

Here, we combine (a) radioisotope-based fossil groundwater prevalence data[1], (b) well-drilling data[19], (c) groundwater level data, and (d) a novel geodatabase consisting of 440 aquifer systems, providing locally relevant study areas in the contiguous US. Together, these data are used to meet three objectives.

(i) Our first objective is to evaluate the spatial distributions of wells accessing fossil groundwater, using wells deeper than $200 \pm 100$ m as a proxy for fossil water access (Results section entitled: Fossil groundwater accessed across US aquifers). We use deep wells as a proxy for fossil water access because densely distributed well depth data are available for the great majority of our study aquifers and wells that have been drilled deeper than $200 \pm 100$ m tend to draw some fossil groundwater when pumped (Supplementary Fig. 1; Fig. 1). Furthermore, continent-wide and densely distributed groundwater $^{14}$C measurements are not available. As a result, we cannot evaluate locally relevant depths below which fossil water dominates storage in each aquifer system (Methods section entitled: Limitations to our results due to the lack of adequate groundwater age data).

(ii) Our second objective is to test whether the frequency with which wells access fossil groundwater has increased or decreased over time (Results section entitled: Fossil groundwater accessed more frequently over time).

(iii) Our third objective is to test if groundwater level declines are disproportionately common where wells are sufficiently deep to access fossil groundwater (Results section entitled: Fossil-groundwater-use hotspots do not always co-occur with groundwater depletion hotspots).

## Results

**Fossil groundwater accessed across the US**. To better understand the spatial distribution of wells that potentially access fossil groundwater, we analyzed ~5.3 million groundwater wells in 440 aquifer systems across the contiguous United States. We interpret the prevalence of deep wells—defined as wells deeper than $200 \pm 100$ m—as a proxy for the prevalence with which wells tap fossil groundwater. Previous calculations of fossil groundwater prevalence in four thousand wells across the US[1] demonstrated that fossil groundwater is common in wells with depths that exceed $200 \pm 100$ m (Supplementary Fig. 1). Our meta-analysis of studies that report on the occurrence of fossil groundwater in US aquifers supports this finding (Fig. 1 panels b-m), and highlights that fossil groundwater occurs in a wide variety of hydrogeologic settings (Fig. 1; meta-analysis in Supplementary Table 1; Supplementary Figs. 9 and 10 for the spatial relationship of fossil well water presented in ref. [1] and the meta-analysis presented here).

We analyzed spatial patterns of wells that are deeper than $200 \pm 100$ m, and, therefore, likely to pump fossil groundwater. We show that wells drilled deeper than 100 m, 200 m, and 300 m are widespread in many US aquifers (Fig. 2a–c). Each point in Fig. 2a-c presents the proportion of wells within an aquifer system that are deeper than 100 m (Fig. 2a), 200 m (Fig. 2b), or 300 m (Fig. 2c); that is, each of the points in Fig. 2a-c represents the proportion of all drilled wells that are deeper than $200 \pm 100$ m within one of the aquifer polygons displayed in Fig. 1a. Specifically, more than one-in-ten wells have depths that exceed 100 m in 67% of our study aquifers, exceed 200 m in 17% of aquifers, and exceed 300 m in 4.6% of aquifers (Fig. 2). More than one-in-five wells have depths that exceed 100 m in 49% of our study aquifers, exceed 200 m in 8.0% of study aquifers, and exceed 300 m in 2.1% of our study aquifers (Fig. 2). Nevertheless, deep wells that tap fossil aquifers are far outnumbered by shallower wells with depths less than $200 \pm 100$ m in most aquifers.

Aquifer systems that have high proportions of wells with depths exceeding $200 \pm 100$ m include layered sedimentary aquifer systems in the northern Great Plains (e.g., eastern portion of South Dakota, where wells tap the Dakota Aquifer), southern Texas (e.g., Carrizo-Wilcox Aquifer System), central Texas (e.g., the Stockton Plateau and the Balcones Fault Zone, each part of the broader Edwards-Trinity Aquifer System), and alluvial basins in Arizona (e.g., Picacho Basin, Maricopa-Stanfield Basin; Harquahalla Basin, and Little Chino Valley), Nevada (e.g.,

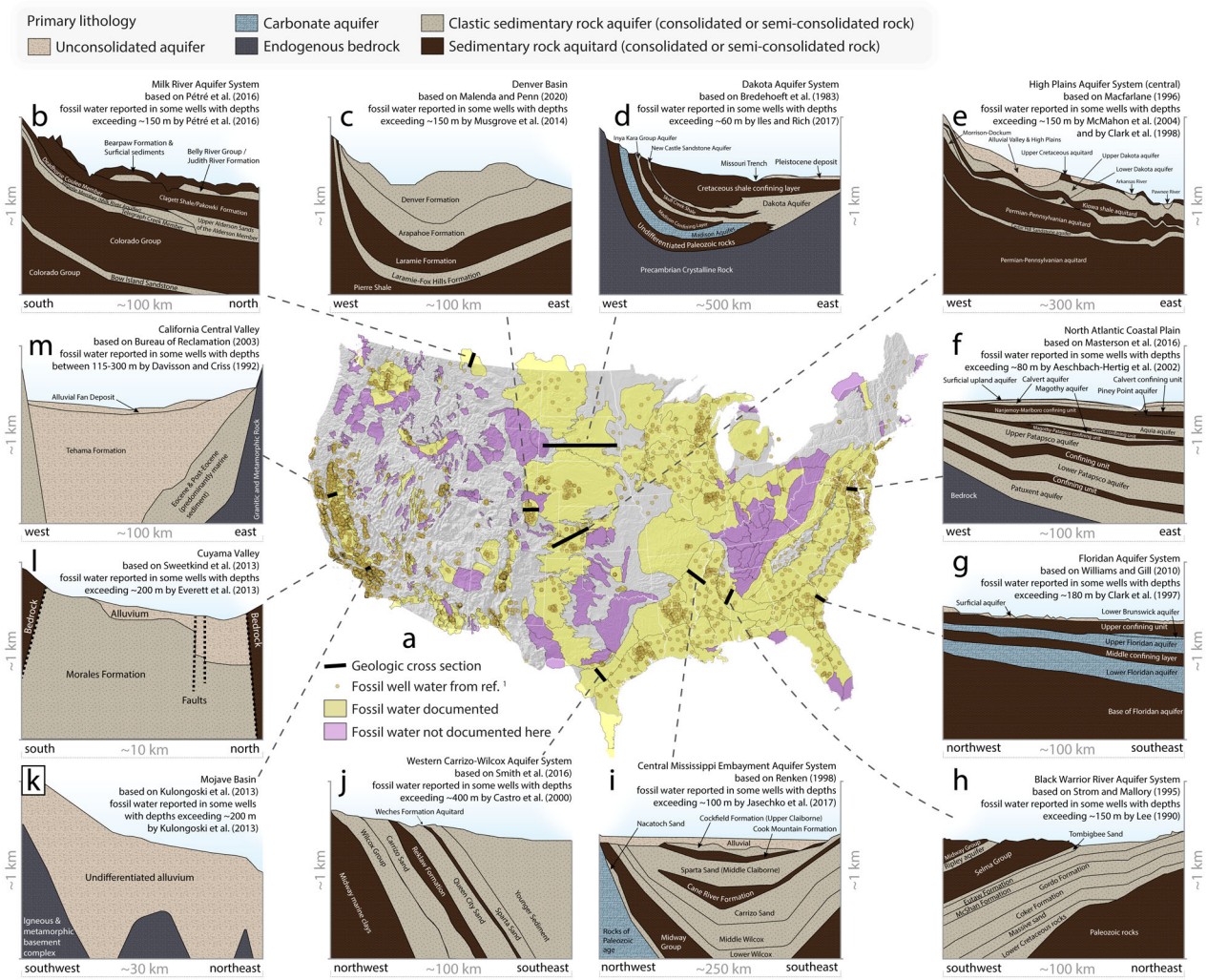

**Fig. 1 Boundaries of our newly created United States Aquifer Database and documented occurrence of fossil groundwater in the US. a** Boundaries of our 440 locally relevant study areas delineated after reviewing hundreds of primary literature sources describing aquifer boundaries (see Supplementary Table 4 for references). Yellow polygons represent aquifer systems where fossil water has been identified by ref. [1] or in our meta-analysis. For a comparison of the spatial distribution of fossil aquifers identified by ref. [1] and those identified in our meta-analysis see Supplementary Figs. 9 and 10. Pink polygons represent aquifer systems included in our analyses but where we have not found document of fossil groundwater. Gold circles represent wells where fossil groundwater has been identified in ref. [1]. **b** The Milk River Aquifer System is dominated by clastic sedimentary aquitards and aquifers, with fossil groundwater reported in some wells with depths exceeding ~150 m[58]. **c** The Denver Basin is a multi-layered clastic sedimentary aquifer system, with fossil groundwater reported in some wells with depths exceeding ~150 m[59]. **d** The Dakota Aquifer System is comprised of carbonate and clastic sedimentary rocks overlying endogenous bedrock; fossil groundwater has been reported in some wells in southeastern South Dakota at depths exceeding ~60 m[60] and also in some wells in Nebraska with depths exceeding ~170 m[61]. **e** The central portion of the High Plains Aquifer System consists of unconsolidated deposits overlying sedimentary rocks (mostly clastic rocks; e.g., sandstones and mudstones of the Dakota Formation), with fossil groundwater reported in some wells with depths exceeding ~150 m[62]. **f** The North Atlantic Coastal Plain is a multi-layered sedimentary aquifer system underlain by endogenous bedrock, with fossil groundwater reported in some wells with depths exceeding ~80 m[63]. **g** The Floridan Aquifer System consists of a surficial aquifer that is underlain by sedimentary rocks including widespread carbonate aquifers interbedded with confining layers, with fossil water reported in some wells with depths exceeding ~180 m[64]. **h** The Black Warrior River Aquifer System is dominated by clastic consolidated or semi-consolidated aquifers and Paleozoic bedrock, with fossil groundwater reported in some wells with depths exceeding ~150 m[21]. **i** The central portion of the Mississippi Embayment Aquifer System consists of unconsolidated alluvium overlying consolidated clastic sedimentary rocks, with fossil groundwater reported in some wells with depths exceeding ~100 m[1]. **j** The western portion of the Carrizo-Wilcox Aquifer System is a multi-layered sedimentary aquifer system, with fossil water reported at depths exceeding ~400 m[65]. **k** The Mojave Basin consists of alluvium overlying endogenous rock, with fossil groundwater reported in some wells with depths exceeding ~200 m[66]. **l** The Cuyama Valley is comprised of alluvial materials overlying (semi)consolidated clastic bedrock, with fossil water reported in some wells with depths exceeding ~200 m[20]. **m** The northern portion of California's Central Valley Aquifer System is comprised of alluvial materials overlying (semi)consolidated clastic bedrock, with fossil water reported in some wells with depths of 115 to 300 m[67]. Each of the 12 cross sections (panels b-m) are based on descriptions and figures presented by refs. [20,58,68–77]. See Supplementary Tables 5-16 for detailed descriptions of hydrostratigraphy; see Supplementary Figs. 9-10 for alternate and enlarged versions of this figure.

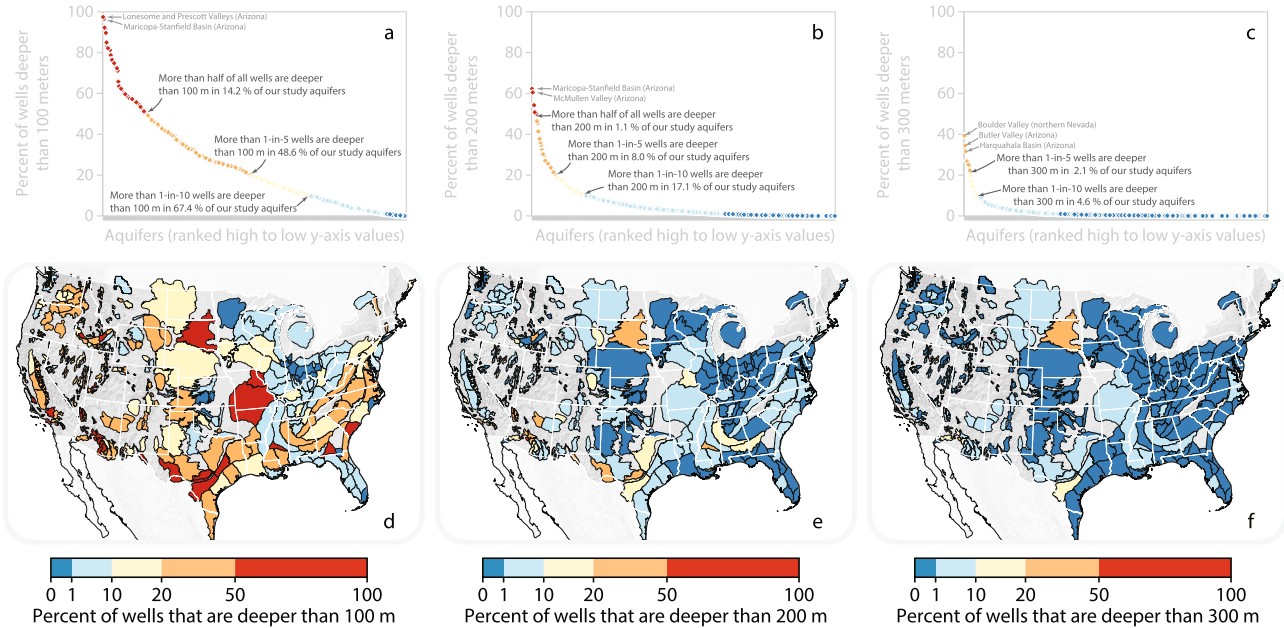

**Fig. 2 The fraction of wells that are deeper than 100 m, 200 m or 300 m in US aquifer systems.** Panels (**a**–**c**) present the fraction of wells deeper than 100 m, 200 m or 300 m for each aquifer (i.e., each diamond represents the analysis of all wells within a particular aquifer). The data are ranked from highest (i.e., largest proportion of wells within an aquifer that are deeper than 100 m (panel **a**), 200 m (panel **b**) or 300 m (panel **c**)) to lowest y-axis values. The red, orange, light yellow, light blue and dark blue diamonds in panels **a**, **b**, and **c** represent the percentage of all wells within a given aquifer system's boundaries that have been drilled deeper than 100 (panel **a**), 200 (panel **b**) or 300 m (panel **c**). Panels (**d**–**e**) display spatial patterns of the fraction of wells deeper than 100 m, 200 m or 300 m. Each polygon represents one aquifer system. Red and orange shades mark aquifers with higher proportions of wells constructed to deep depths.

Boulder Valley, Coyote Springs Valley) and California (e.g., Cuyama Valley, Los Angeles Basin, and Santa Clara-Calleguas Basin; Fig. 2d-f). By contrast, areas where the vast majority of wells are shallower than 200 ± 100 m include the Puget Sound Lowlands (Washington), Great Bend Prairie, and Equus Beds of the east-central High Plains Aquifer System (Kansas), Central Lowland Till Plain aquifers (Illinois, Indiana, Ohio, Kentucky), the Michigan Basin (Michigan), the Cape Cod Aquifer System (Massachusetts), and the Biscayne Aquifer System (Florida; Fig. 2).

**Fossil groundwater accessed more frequently over time.** We tested for changes in the prevalence with which wells tap fossil aquifers over time. We analyzed temporal changes in the fraction of newly constructed wells drilled deeper than a nearby well that is known to draw fossil water. Specifically, we identified wells known to draw some fossil groundwater (>0% as calculated by ref. [1]); fossil well waters can be identified by attributing low $^{14}C$ activities to radioactive decay, after accounting for other potential carbon sources on the basis of $^{13}C/^{12}C$ measurements (for further details see Supplementary Note 1.2 and the methods section within ref. [1]). Next, we define 'study areas' as areas within a 20 km radius of a well that is known to draw fossil water (see Supplementary Fig. 2). We identified newly drilled wells that are located within each study area. We then calculated the proportion of newly drilled wells that have depths that are deeper than the nearby well that draws fossil water. Last, we quantified temporal variations in the frequency with which new wells are constructed deeper than the nearby well known to draw some fossil water (see Supplementary Fig. 2 for a schematic of statistical analysis). We analyzed five-time intervals: 1950–1975, 1975–2000, 2000–2015, 1950–2015, and 1975–2015.

We show that the proportion of wells tapping fossil aquifers has increased over time in more places than it has decreased

over time (Table 1). Specifically, we show that study areas where the proportion of wells tapping fossil aquifers has increased over time are ~1.4–3.4 times more common than study areas where the proportion of wells tapping fossil aquifers has decreased over time (Table 1). We conclude that the proportion of newly drilled wells that are sufficiently deep to tap fossil aquifers has increased over time in more places than it has decreased in the US.

On an aquifer-by-aquifer basis, we identified 36 aquifer systems that contain at least five wells known to draw fossil water with sufficient nearby (<20 km) well-drilling data for analyses. We find evidence for an increase over time in the proportion of wells tapping fossil aquifers in the Black Warrior River Aquifer System, the Central High Plains and California's Santa Rosa Valley (all of which have median rank correlation coefficients exceeding zero for all studied time intervals; Fig. 3).

In addition to the above analysis, we completed a complementary analysis for each of our aquifer systems by calculating temporal variations in the proportion of newly drilled wells that exceed 200 ± 100 m (Supplementary Note 2.2). Specifically, we calculated correlation coefficients of the rank transforms of groundwater well construction year versus the proportion of all wells drilled within a given year that are deeper than 200 ± 100 m. These Spearman rank correlation coefficients (i.e., ρ values) demonstrate that the fraction of new wells that are drilled deeper than 200 ± 100 m has increased over time in more than half of all aquifers with sufficient data for analyses (Supplementary Figs. 3 and 4).

Among aquifer systems with sufficient data, 1.2–14 times more aquifers show an increase over time in the fraction of wells drilled deeper than 200 ± 100 m than those that show a decrease (Supplementary Fig. 4; Supplementary Table 2). If we limit our analyses to consider only significant correlations (Spearman P values

**Table 1 Temporal variations in the proportion newly drilled wells that are deeper than a nearby (within 20 km) well that pumps fossil water (Supplementary Fig. 2 for schematic of analysis).**

| Time interval | Total number of areas* analyzed | Number of areas* where the rank correlation coefficient is greater than zero (i.e., the proportion of wells being drilled deeper than a nearby well that is known to contain fossil water is, if anything, increasing over time) | Number of areas* where the rank correlation coefficient is less than or equal to zero (i.e., the proportion of wells being drilled deeper than a nearby well that is known to contain fossil water is not increasing over time) | Number of areas where the rank correlation coefficient is undefined | (the number of areas with rank correlation coefficient of greater than zero) divided by (the number of areas with rank correlation coefficient of less than zero) |
|---|---|---|---|---|---|
| 1950-1975 | 436 | 230 | 144 | 62 | 1.6 |
| 1975-2000 | 703 | 400 | 190 | 113 | 2.1 |
| 2000-2015 | 855 | 433 | 314 | 108 | 1.4 |
| 1950-2015 | 427 | 303 | 100 | 24 | 3.0 |
| 1975-2015 | 643 | 451 | 131 | 61 | 3.4 |

* defined as land area within a 20 km radius of a well that has been identified† to pump fossil water.

of less than 0.05), our conclusion becomes stronger (see large diamond symbols denoting significant ($P$ value < 0.05) correlations in Supplementary Fig. 3). This complementary analysis supports our finding: there is an increase in the proportion of wells being constructed to deep depths where fossil groundwater is common in the majority of aquifers that we studied (see Supplementary Fig. 4). We conclude that the proportion of wells tapping fossil groundwater resources is likely increasing across the US.

**Fossil-groundwater-use hotspots do not always co-occur with groundwater depletion hotspots**. We compared spatial patterns of groundwater level variations over time (Fig. 4) and the proportion of wells within aquifers that have depths exceeding 200 ± 100 m (Supplementary Note 5). First, we analyzed long-term groundwater-level trends in our aquifers to test for spatial relationships between the prevalence of deep wells (indicative of wells that access fossil water) and declining (deepening) groundwater levels. We calculated Theil-Sen slopes to describe the rate of change in groundwater levels over time for each monitoring well that had sufficient data for analyses (Methods). Next, we present the median Theil-Sen slope for each aquifer system (depicted as shaded areas in Fig. 4; median calculated on the basis of Theil-Sen slopes among all monitoring wells within a delineated aquifer; for the schematic of the method see Supplementary Fig. 5). Finally, we evaluated correlations between the proportion of recorded wells within an aquifer system that exceed 200 ± 100 m versus two different metrics of groundwater level change over time: (i) median of all monitoring wells' Theil-Sen slopes, determined for any aquifers with sufficient data for analyses, and (ii) the proportion of all monitoring wells within the aquifer system that have Theil-Sen slope values indicative of groundwater level deepening over time.

We did not identify a consistently positive (or negative) correlation coefficient describing variations between the prevalence of deep wells and groundwater level changes over time (Supplementary Table 17), reinforcing that the use of fossil groundwater does not have to mean that groundwater use is non-renewable.

We do find examples of aquifer systems that are tapped by high proportions of wells deeper than 200 ± 100 m and have also experienced groundwater-level declines. For example, in California's Cuyama Valley (Fig. 1l), we find that 16–63% of wells are deeper than 200 ± 100 m, that low-$^{14}$C groundwater samples have been collected from wells deeper than 200 ± 100 m[20] (indicative of fossil well water), and that groundwater reserves are being depleted (i.e., the median groundwater level is deepening at a rate of 1–2 m/decade across all three-time intervals presented in Fig. 4). Here, fossil groundwater is accessed and groundwater stores are being depleted.

Conversely, we find examples of aquifers that are tapped by high proportions of wells deeper than 200 ± 100 m that have not experienced substantial and pervasive groundwater-level declines over recent decades. For example, the Black Warrior River Aquifer System (eastern Mississippi through Alabama; Fig. 1h) has a similar percentage of wells that are deeper than 200 ± 100 m as the Cuyama Valley (5–44%), and also contains groundwater with low-$^{14}$C activities that are indicative of fossil groundwater[21]. But, in contrast to the Cuyama Valley, the median groundwater-level trend in the Black Warrior River Aquifer System has remained near-zero among the three-time intervals we studied (Fig. 4a-c: (a) 0.0 m/decade (1950–1975), (b) +0.2 m/decade (1975–2000), and (c) −0.3 m/decade (2000–2015)). Here, fossil groundwater is likely tapped, but hydraulic heads have remained relatively stable over each of the three-time intervals we studied.

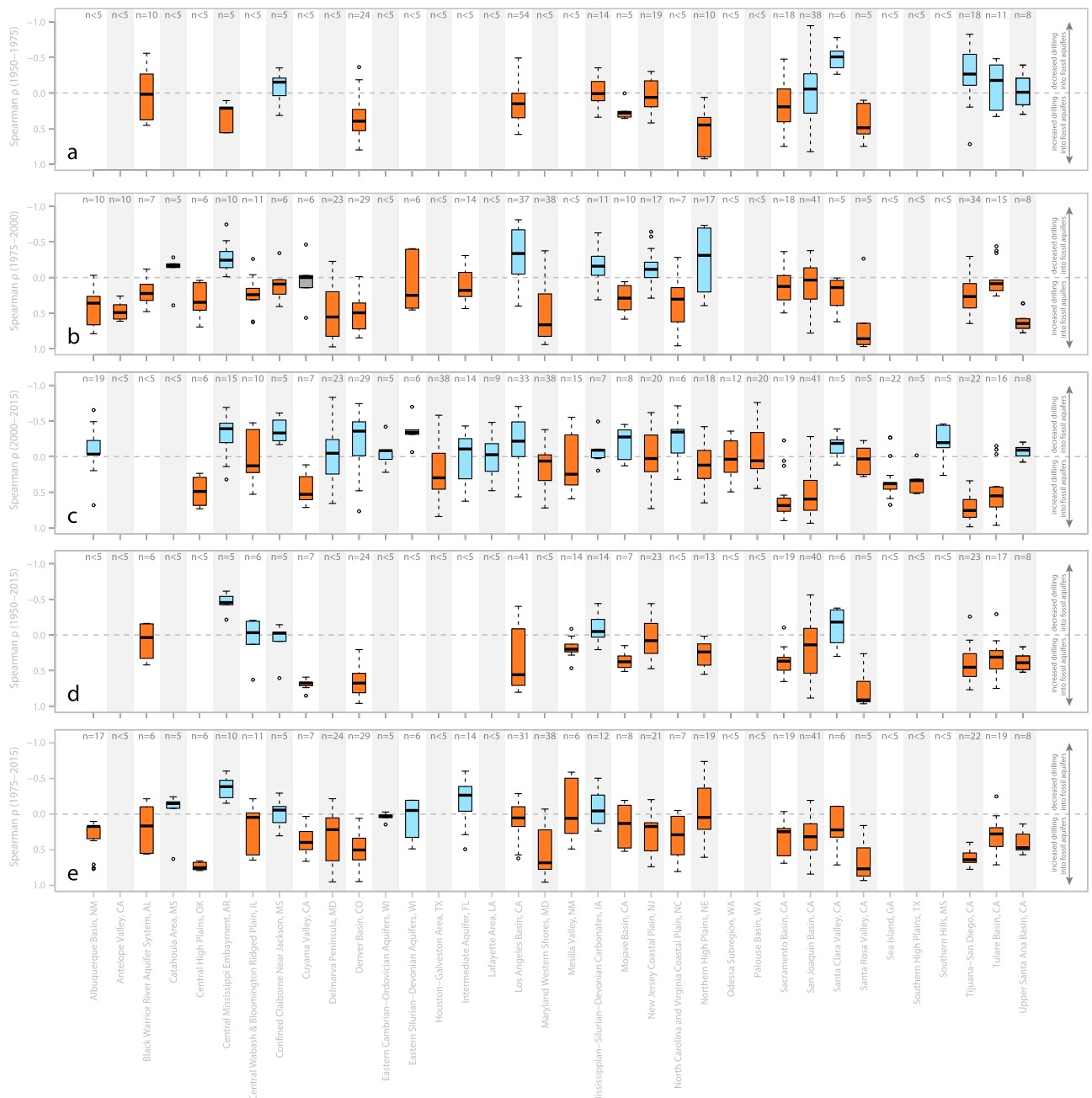

**Fig. 3 Temporal variations in well drilling depths in areas located nearby a well that has been reported to pump fossil water.** The Spearman rank correlation coefficients (ρ) were determined by correlating calendar year versus the fraction of newly constructed wells with depths exceeding the depth of a nearby (<20 km away) groundwater well that is known to pump fossil water. Each panel (**a**-**e**) represents correlations completed over one of five different time intervals: 1950–1975, 1975–2000, 2000–2015, 1950–2015 and 1975–2015. Each bar represents the statistical distribution of all study areas' (i.e., 20 km buffers around a groundwater well that has been reported to pump fossil water) correlation coefficients determined within a given aquifer system's boundaries; the thick horizontal black line represents the median ρ value for all areas with sufficient data within the aquifer system, the top and bottom of the shaded box represents the 25th–75th percentile range of ρ values, the dashed line and cap extends to the 10th–90th percentile range, and circles represent outlier points. Aquifer systems marked with orange-shaded boxplots have median ρ values exceeding zero (indicative of an increasing proportion of newly drilled wells that are deeper than the well that has been documented to pump fossil water); aquifer systems marked with blue-shaded boxplots have median ρ values of equal to or less than zero (indicative of an unchanging or decreasing proportion of newly drilled wells that are deeper than the well that has been documented to pump fossil water). The number (i.e., text reading: n = x) overlying each box plot represents number of study areas within the aquifer with sufficient data for analyses for a given time interval. We only present box plots for aquifer systems with at least five study areas with sufficient data to determine a rank correlation coefficient. The labels on the x-axis display the title of each aquifer system and the two-letter code for the state that the centroid of the aquifer system lies within (e.g., two-letter code CA denotes that the centroid of the aquifer system lies within California).

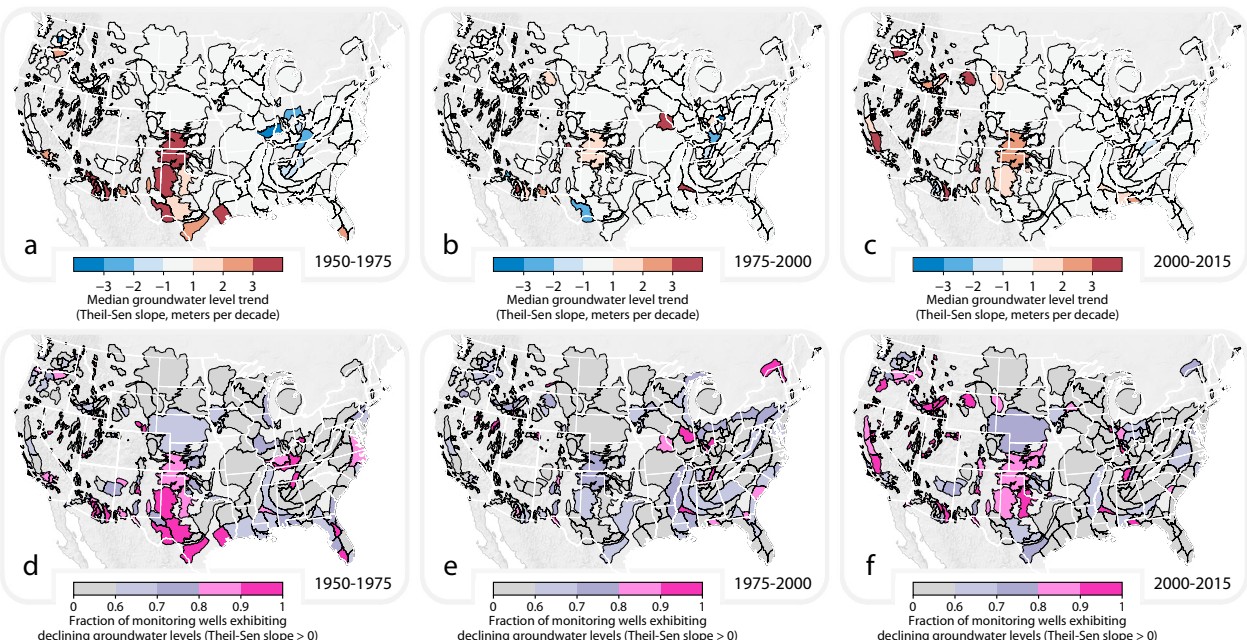

**Fig. 4 The median groundwater-level change rate (a–c) and the fraction of monitoring wells exhibiting declining groundwater levels (d–e) across US aquifer systems included in our US Aquifer Database (see Methods section entitled: Delineating aquifers across the US).** The upper row of figures displays the median Theil-Sen slope of the groundwater-level trend (expressed in meters per decade) analyzing all water level measurements within the years (**a**) 1950–1975, (**b**) 1975–2000, and (**c**) 2000–2015 (the median trend was calculated by determining the Theil-Sen slope for every monitoring well within the aquifer boundaries, and then calculating the median among these Theil-Sen slope values; see Supplementary Fig. 5 for schematic of method). The lower row of figures displays the proportion of all monitoring wells that have a Theil-Sen slope exceeding zero (panels **d–f**), thus presenting the fraction of monitoring wells within the aquifer boundaries that exhibit declining groundwater levels for the years (**d**) 1950–1975, (**e**) 1975–2000, and (**f**) 2000–2015. We only present aquifers for which we analyzed groundwater-level time series for at least five unique monitoring wells (over the specified time interval).

## Discussion

**Water quality ramifications of fossil groundwater use in the US.** We find that wells that likely tap fossil aquifers are widespread and becoming more common over time (Figs. 2 and 3). These results have implications for understanding and quantifying key processes influencing well water vulnerability to (a) surface-borne pollutants, (b) geogenic contaminants, and (c) groundwater salinity.

(a) Surface-borne pollutants contaminate groundwater resources when they percolate downward through the soil profile and unsaturated zone to enter the groundwater from above. Fossil groundwater tends to have lower concentrations of surface-borne pollutants than younger groundwater. For example, nitrate—a common surface-borne pollutant that is frequently associated with confined animal feeding operations, excessive fertilization and inadequate sanitation[22,23]—is more common in recently recharged (i.e., 'younger') groundwater than in older groundwater[14]. Because groundwater age tends to increase with depth, we compared tens-of-thousands of dissolved nitrate measurements in shallow (<50 m) and deep (100–300 m) wells to understand potential water quality ramifications of fossil groundwater use (Supplementary Note 6). We find that shallower wells tend to have high nitrate concentrations (>10 mg/L $NO_3$ as N) more frequently than deeper wells in most of the aquifer systems that we studied (Supplementary Fig. 13).

Thus, our finding that fossil groundwater access is increasing over time may suggest reduced exposure to surface-borne contaminants in some of these areas; nevertheless, we stress that deep wells can still be

vulnerable to contaminants because deep wells often contain mixtures of fossil groundwater and recent recharge[1,24]. The processes that lead to mixtures of fossil groundwater and recent recharge remain poorly understood, but may include (i) mixing along converging flow paths (e.g., Arava Valley in Israel[25]); (ii) cross-formational mixing (upconing and downwelling) induced as a result of borehole drilling and subsequent pumping (e.g., Diass Aquifer System in Senegal[26] and the North China Plain[27]); (iii) fast downward flow of recent precipitation along defective well casings to deeper depths where the well is perforated (e.g., Bohemian Cretaceous Basin in the Czech Republic[28] and the Aleppo and Steppe Basins in Syria[29]); (iv) pumping from wells with long perforated intervals that simultaneously draw groundwater from both shallow and deep depths (e.g., Malm Limestone Aquifer System in Poland[30]); (v) leakage through gaps in impermeable layers ('windows' in aquitards) separating fossil and modern groundwater; or (vi) relatively rapid vertical groundwater flow and mixing along geologic faults that may serve as conduits that connect fossil and modern groundwater. Because shallow contaminated groundwater can be drawn downward by pumping from deeper wells[31], developing data products that quantify not only total groundwater withdrawals in a region but also quantify the depths of the wells (and the depths of their screened intervals) from which groundwater is withdrawn will be key to understanding fossil aquifer contamination risk.

(b) Geogenic contamination can arise as groundwater interacts with the mineral skeleton of the geologic formations that it

flows through and resides within. For example, arsenic is a common geogenic pollutant that poses a challenge to the provision of fresh drinking water in multiple aquifer systems across the world[32]. Aqueous arsenic concentrations can differ between fossil versus younger groundwater, though the statistical relationships between groundwater age and arsenic concentrations vary among aquifer systems and are critically dependent on hydrostratigraphy, redox conditions, and flow path architectures[33,34]. For example, in California's Cuyama Valley, samples of fossil groundwater pumped from deep wells tend to have higher arsenic concentrations than samples of younger groundwater drawn from shallower wells[20]. Conversely, in the Bengal Basin (Bangladesh), groundwater found deeper than ~100 m tends to have lower arsenic concentrations than groundwater found at shallower depths[35]. Because groundwater age tends to increase with depth, we compared tens-of-thousands of dissolved arsenic measurements in shallow and deep wells (Supplementary Fig. 12). In some aquifer systems ($n = 47$ aquifers), shallower wells are contaminated by arsenic ($>10 \ \mu g/L$) more frequently than deeper wells. In other aquifer systems ($n = 54$), deep wells are contaminated by arsenic more frequently than shallower wells. The increasing reliance on fossil water demonstrated here may increase exposure to arsenic in some places, but reduce exposure in other places.

Our finding that fossil groundwater reliance is increasing as deep wells become more common may imply concomitant changes in exposure to high-arsenic groundwater; however, we stress that arsenic exposure assessments should be considered on an aquifer-by-aquifer basis because of the importance of local hydrochemical conditions and hydro-stratigraphy in determining groundwater arsenic concentrations[35]. Further, in some of the areas where we show the proportion of newly drilled wells that are deeper than $200 \pm 100$ m to be increasing, it is possible that excessive pumping from deep (semi)confined aquifers may alter aqueous arsenic concentrations in the groundwater; for example, in some areas, pumping has induced leakage of high-arsenic groundwater (or arsenic-mobilizing solutes) from aquitards into adjacent aquifers (e.g., some parts of California's Central Valley[36] and Vietnam's Mekong Delta[37]). Beyond arsenic, elevated activities of naturally occurring radioisotopes (e.g., $^{226}Ra$, $^{228}Ra$) have been identified in some samples of fossil groundwater, including those collected from the Disi Sandstone Aquifer of Jordan[38], the Saq Aquifer System and the Mega Aquifer System of the Arabian Peninsula[39,40], and the Nubian Sandstone Aquifer System of northeastern Africa[41]. Careful consideration of treatment options (e.g., blending, reverse osmosis) may be warranted in some of the areas where fossil groundwater is tapped for direct household use.

(c) Third, elevated salinity levels can render groundwater inadequate for drinking and irrigation. Salinization mechanisms are diverse, and different aquifer systems will have unique statistical relationships between groundwater age and salinity depending on the natural geologic setting, proximity to coastal waters, and historical land uses (e.g., irrigation practices; ref. [42]). Our finding that fossil groundwater use is widespread and increasing has a number of implications for understanding the vulnerability of groundwater users to high-salinity levels.

In coastal settings, there are multiple aquifer systems where fossil groundwater has been identified and where wells tap these fossil aquifers (e.g., portions of the North Atlantic Coastal Plain and the Floridan Aquifer System). Pumping water from deep aquifers in these settings can lower hydraulic heads below sea level, rendering these deep fossil aquifers vulnerable to landward incursions of seawater (e.g., North Atlantic Coastal Plain at Cape May, New Jersey[43]) or upconing of saline water from below (e.g., Floridan Aquifer System at Brunswick, Georgia[44]). Many deep wells in the US have water levels that lie below sea level, implying some deep aquifers tapped by these wells may be vulnerable to seawater intrusion[45].

Farther inland, deeper groundwater is generally more likely to be fossil in its age, and also more likely to be brackish or saline[46].

The increasing prevalence of deep wells in the majority of our study aquifers implies that groundwater wells may be encroaching on the depths at which some aquifer systems transition from shallow-and-fresh to deep-and-brackish conditions, likely limiting the effectiveness of drilling deeper wells indefinitely without concomitant treatment[19]. However, we stress that there are also aquifer systems where shallower wells are more likely to pump brackish water than deeper wells; our analysis of hundreds-of-thousands of total dissolved solids measurements identified a dozen such cases, most of which are arid alluvial basins in the western US (Supplementary Fig. 14). The high spatial variability in the statistical relationship between well water salinity and well depth highlights the importance of considering local hydrogeologic settings and historic land uses when examining connections between increased well drilling into fossil aquifers and the potential threat of salinity to groundwater users.

**Water quantity implications of fossil groundwater use in the US.** Unsustainable groundwater use is depleting groundwater stores in numerous aquifer systems around the world[47–51], with cascading ramifications for irrigated agriculture and food trade. Around the globe, there are examples of aquifer systems where fossil ground-water pumping coincides with groundwater depletion (e.g., Saq Aquifer System of Saudi Arabia; ref. [52]). Here, we identify US aquifer systems where wells likely tap fossil groundwater and groundwater levels have declined (e.g., California's Cuyama Valley). We also identify US aquifer systems where wells likely tap fossil groundwater but existing monitoring well networks have not captured con-comitant declines in groundwater stores (e.g., Black Warrior River Aquifer System). Our finding that fossil-groundwater-use hotspots do not always co-occur with groundwater-depletion hotspots rein-forces the point that the use of fossil groundwater does not have to mean that groundwater use is non-renewable (see Fossil Ground-water in Table 1 within ref. [15]). Depletion is a complex process that can be more readily informed by real-time withdrawal measure-ments (rather than estimated withdrawals), in addition to well depth and screen interval information that would allow withdrawal data to be linked back to specific geologic formations. Such information, when combined with groundwater recharge and discharge estimates, can provide a more nuanced understanding of depletion dynamics, which can assist in developing management frameworks.

Although the use of fossil groundwater does not have to mean that groundwater use is non-renewable, fossil groundwater does tend to be more common in deeper aquifers (Supplementary Fig. 1). Because deeper aquifers are more likely to be confined than shallower aquifers, fossil groundwater is disproportionately common in confined aquifers. Pumping groundwater from confined aquifers can have different ramifications on metrics of groundwater quantity (e.g., hydraulic heads, vertical hydraulic gradients) than similar amounts of pumping from unconfined aquifers, because confined aquifers typically have a lower

storativity than unconfined aquifers. Sustained pumping from confined aquifers can lead to leakage from surrounding geologic formations, substantial declines in hydraulic heads, land subsidence as adjoining confining units are compressed[53], or a combination of the aforementioned impacts.

From the perspective of groundwater quantity management, our research stresses the importance of moving beyond estimates of total groundwater pumping rates to include the depths at which (or geologic formations from which) groundwater is withdrawn from. To the best of our knowledge, available national-scale groundwater withdrawal estimates are two-dimensional data products that do not provide information about the vertical distribution of groundwater withdrawals. Three-dimensional groundwater withdrawal data could improve our understanding of the short- and long-term impacts of pumping fossil groundwater on hydraulic gradients, cross-formational flows, and land subsidence.

**Fossil groundwater use in the US.** Sustainably using finite fresh groundwater resources remains key to industrial productivity, irrigated agriculture, and the provision of clean, reliable, and convenient domestic water supplies. Fossil groundwater is common in US aquifer systems (Fig. 1). Widespread reliance on fossil groundwater (Fig. 2) in the US suggests that safeguarding the quality of deep and fresh fossil groundwater is key to modern water provision. Further, because the prevalence of deep wells tends to be increasing in most areas (Fig. 3), protecting fossil aquifers from overuse (Fig. 4) and pollution will be key to meeting future water demands.

## Methods
**Datasets analyzed**. We meet our objectives by combining four databases: (a) fossil groundwater prevalence determined on the basis of published analyses of groundwater radiocarbon data[1]; (b) records of groundwater well depths, construction dates, and purposes for millions of wells at continental scale[19,54]; (c) long-term groundwater level time-series recorded by the United States Geological Survey and California's Groundwater Ambient Monitoring and Assessment Program, and (d) a new geodatabase of hydrogeologic study areas (United States Aquifer Database; see Fig. 1a for aquifer system outlines).

**Delineating aquifers across the US (Fig. 1)**. We delineated boundaries for hundreds of aquifer systems across the United States by examining maps and reading descriptions within local- and regional-scale reports (e.g., United States Geological Survey reports). Methods and specific references consulted when developing the new geodatabase are detailed in Supplementary Note 3.1, which includes an extensive table detailing specific references and approaches applied to delineate each of our study aquifers (Supplementary Table 4).

We name our new aquifer boundary database the United States Aquifer Database. In places in this text we refer to these delineated, two-dimensional areas (i.e., polygons in Fig. 1a) as 'aquifers', although we stress that these two-dimensional areas are underlain by multiple geologic formations, each defined as separate local aquifers or aquitards that together form 'aquifer systems'. This geodatabase includes 440 aquifer systems across the US and addresses several shortcomings of existing nationwide aquifer spatial databases (Supplementary Note 3.1). Our newly delineated US Aquifer Database is preferable for our study over other databases (e.g., see Supplementary Note 3.2) for four reasons. Specifically, our United States Aquifer Database:

(i) subdivides broad aquifer systems (e.g., the entire High Plains, which, when considered as a single expansive area, is too expansive for locally relevant science) into smaller subareas (e.g., Northern High Plains, Central High Plains, Southern High Plains, Great Bend Prairie, and Equus Beds; Supplementary Fig. 6);

(ii) partitions separate valleys (i.e., basins) that are unlikely to share strong hydraulic connections into separate study areas (e.g., our US Aquifer Database treats individual valleys as separate study areas; Supplementary Fig. 7);

(iii) specifies and includes aquifer systems that have been widely studied for more than a century (e.g., the Dakota Aquifer System[55,56]; Supplementary Fig. 8); and,

(iv) has been informed by our nation-wide compilation of groundwater well drilling geospatial data[19], meaning these locally relevant hydrogeologic data were available to help guide the delineation of specific aquifers accessed by

actual wells (e.g., the 2D extensiveness of relatively deep wells helped us delineate areas where the Intermediate Aquifer (part of the broader Floridan Aquifer System) is accessed by wells in southwestern Florida).

Equipped with our new geospatial database of hydrogeologic study areas (Fig. 1a), we analyzed spatiotemporal variations in the prevalence of deep wells and observed groundwater-level fluctuations across these hydrogeologic study areas (see Methods sections entitled: Fossil groundwater accessed across US aquifers (Fig. 2), Fossil groundwater accessed more frequently over time (Fig. 3), and Groundwater depletion in places where deep wells tap fossil aquifers (Fig. 4)).

The previous lack of locally relevant and continent-wide aquifer geospatial data necessitated that the scope of previous well-completion depth studies[19] was limited to only a few expansive aquifer systems, rather than the 440 aquifer systems delineated here. Here, we pair well completion depth data[19] and radioisotope-based fossil groundwater prevalence data[1] to our new aquifer geodatabase (Fig. 1a); we develop a new method designed to explore well-drilling depth changes over time surrounding sites where fossil water has been identified (see Methods section entitled: Fossil groundwater accessed more frequently over time (Fig. 3)) to distinguish this study from previous works.

**Reports of fossil groundwater in the US (Fig. 1)**. In determining the proposed well depth threshold of 200 ± 100 m (see Methods subsections to follow), we considered our meta-analysis that documented studies reporting fossil well water (Supplementary Fig. 9) and our age-depth data analysis (Supplementary Fig. 1).

We identified $n = 114$ (of $n = 440$ study aquifers) US aquifers where at least one publication has reported that at least some sampled groundwater is more than 12 thousand years old (yellow polygons in Fig. 1a). The compiled studies base their interpretation that fossil water is present in the aquifer system on radioisotope measurements, such as $^{14}C$ and/or $^{36}Cl$.

We also present depth variations in fossil groundwater prevalence, based on water samples collected from approximately four thousand wells in the United States (see methods in ref. [1] and Supplementary Note 1.2). Fossil groundwater tends to be more common in deeper wells, especially wells with depths exceeding 200 ± 100 m (Supplementary Fig. 1). Specifically, more than ~30% of wells with depths at or exceeding 100 m contain detectable fossil water, and more than ~half (48%) of wells with depths at or exceeding 300 m contain detectable fossil water (Supplementary Fig. 1).

**Fossil groundwater accessed across US aquifers (Fig. 2)**. Uncertainty in our analysis of fossil groundwater reliance derives from the lack of groundwater radiocarbon data in many aquifer systems. However, our analysis of groundwater radiocarbon data that are available makes clear that fossil groundwater is often present in wells that are deeper than 200 ± 100 m (Supplementary Fig. 1). Therefore, we analyzed well completion reports derived from $n = 64$ state and sub-state databases to quantify spatial patterns of deep wells—defined here as those exceeding 200 ± 100 m—across US aquifers (extensive quality control procedures for each well completion database reported by ref. [19]). For each study aquifer containing at least $n = 10$ wells that met our quality control criteria (see Supplementary Information within ref. [19]), we calculated the fraction of all wells within the aquifer that are deeper than 100 m (Fig. 2a), 200 m (Fig. 2b), or 300 m (Fig. 2c).

We emphasize that well-screen interval data are not systematically reported in radioisotope reports nor in well construction reports. As a result, our analyses are based on the total depth of wells. Fossil groundwater is common, though not ubiquitous, in wells that are deeper than 200 ± 100 m, so we interpret the prevalence of wells that are deeper than 200 ± 100 m as a proxy for the prevalence of wells that access fossil groundwater. Fossil groundwater occurrence generally increases with depth across many major US aquifers (Supplementary Fig. 1; ref. [1]), but we acknowledge that there are likely aquifer systems where younger groundwater underlies fossil groundwater (see Supplementary Information within ref. [1]). These limitations highlight an important research gap in the water science community's collection of groundwater data.

**Fossil groundwater accessed more frequently over time (Fig. 3)**. To test if the proportion of newly drilled wells in the US that tap fossil aquifers has increased over time we completed a series of steps. First, (i) we mapped the locations of wells identified in ref. [1] to pump fossil water (minimum fraction of well water comprised of fossil groundwater exceeds zero). Second, we identified all records of well construction within a 20 km radius of the well that is known to pump fossil groundwater. Third, (iii) for each well construction event, we determined whether the total depth of the constructed well is shallower or deeper than the nearby 'fossil well'. That is, we compared the depth of each newly drilled well to that of the nearby 'fossil well', and describe the former in binary terms: (a) newly drilled well is shallower than the fossil well, or (b) newly drilled well is deeper than the fossil well. Fourth, (iv) for each calendar year where at least five wells were constructed within 20 km of the fossil well, we calculated the proportion of wells drilled in that year that are deeper than the fossil well, implying that many of these wells likely also pump fossil water, since they are deeper than a nearby well known to draw some fossil water (i.e., we calculated the fraction of wells drilled deeper than the well that is known to pump some fossil water for a given year). Fifth, (v) we calculated the Spearman rank correlation that describes variations in the fraction of wells that are deeper than the well that is known to pump

fossil water versus calendar year (see Supplementary Fig. 2 for schematic). Last, we determined spatial statistics of these increasing and decreasing trends for a number of aquifer systems across the contiguous United States (Fig. 3).

**Groundwater depletion in places where deep wells tap fossil aquifers (Fig. 4)**.
We compiled groundwater-level monitoring data from the United States Geological Survey and California's Groundwater Ambient Monitoring and Assessment Program. The downloaded data includes 16 million water level measurements from over 600,000 unique groundwater wells within US Aquifer Database areas. These data enabled us to evaluate groundwater-level fluctuations over three unique time intervals: (i) 1950–1975, (ii) 1975–2000, and (iii) 2000–2015.

To test for spatial correspondence between groundwater level declines and the existence of deep wells that likely tap fossil aquifers, we completed a series of steps. First, we calculated the average water level for any unique year with at least one water level measurement for every monitoring well in our database. Next, we filtered our dataset by considering only the monitoring wells that met both of these criteria: (a) at least five unique years within which at least one water level measurement was recorded within a given time interval, and (b) at least one water level measurement within both the first and the last five years of the time interval (e.g., for the time interval 1950–1975, we require at least one measurement between 1950 and 1955 and at least one measurement between 1970 and 1975 for us to consider the monitoring well in our analyses). For each monitoring well meeting these criteria, we calculated the Theil-Sen slope of the calendar year versus the average water level for a given calendar year. For each US Aquifer Database polygon, we calculated the median Theil-Sen slope among all monitoring wells meeting our criteria for analyses that are located within the aquifer bounds (Fig. 4a–c). We also calculated the fraction of all monitoring wells within an aquifer with a Theil-Sen slope indicative of deepening groundwater levels over time (Fig. 4d–f). Last, we compared these Theil-Sen slopes (i.e., groundwater-level variability through time) with the prevalence of wells exceeding 200 ± 100 m (see results section entitled: Fossil-groundwater-use hotspots do not always co-occur with groundwater depletion hotspots).

**Limitations to our results due to the lack of adequate groundwater age data**.
Our methodology is limited by the lack of widespread groundwater radioisotope measurements in deep and shallow wells (e.g., $^{14}$C, $^{36}$Cl). Therefore, we cannot easily resolve the depth below which most stored groundwater is fossil in age for each of our 440 study aquifers. Consequently, our analysis of fossil groundwater reliance depends on the use of well depths as a proxy for fossil water access, with the implicit assumption that deeper wells are more likely to draw fossil water than shallower wells. In an effort to overcome this data limitation, we report the fraction of wells deeper than a range of threshold depths: 100 m to 300 m. We emphasize that, for some aquifer systems, fossil water may not dominate at depths of ~300 m (e.g., ref. [57]), the deepest limit applied to our study.

## Data availability

Delineated aquifer system boundaries are available via CUAHSI's (Consortium of Universities for the Advancement of Hydrologic Science, Inc.) Hydroshare portal at the following website: http://www.hydroshare.org/resource/d2260651b51044d0b5cb2d293d21af08. For information on the development of this dataset see Supplementary Note 3.

## Code availability

Analyses presented here do not depend on specific code; the approach can be reproduced following the procedures described in the Methods section.

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

## Acknowledgements

This material is based upon work supported by the National Science Foundation under Grant No. EAR-2048227.

## Author contributions

M.G. wrote the first draft of the manuscript and led the statistical analyses. S.J. delineated aquifer system boundaries. M.G., D.P., S.J. co-developed methods, discussed results, and wrote the manuscript.

## Competing interests

The authors declare no competing interests.
