## [Peer Review File · Nature Communications]

Widespread and increased drilling of wells into fossil aquifers in the USAReviewers' Comments:

Reviewer #1:

Remarks to the Author:

Review of Widespread and increasing reliance on fossil groundwater in the USA by Merhawi GebreEgziabher GebreMichael, Scott Jasechko and Debra Perrone

The authors use an impressive data set of well depth data, radioisotope groundwater age data and an updated USA aquifer database to assess the fraction of wells likely pumping fossil groundwater (>12000 years), how this fraction developed over time and if fossil groundwater use is related to groundwater depletion.

I really like this study, particularly because it depends on so much great data. A nice bonus for the hydrogeological community is the updated aquifer data set (USAD) that would benefit large-scale groundwater modelers and would attract quite some citations.

This paper should be published after some of the following issues are resolved:

1. The first issue pertains to the rationale behind the importance of knowing whether groundwater pumped is fossil or not. This is based on differences in water quality focused on harmful constituents. I don't think that is a sufficiently strong argument. As discussed later, the concentration of harmful constituents (especially of geogenic nature) can be both higher and lower for fossil and non-fossil water, so what does it help us to know whether groundwater is fossil or not? What is missing from this analysis is that groundwater age is a quality by itself. First, bottlers of mineral water, beer breweries and spas pride themselves of using age old water. This water therefore has an intrinsic and economic value. Second, if some of the fossil water outcrops in springs or wetlands or discharges into lowland streams, its specific quality (e.g., being calcium rich and nutrient poor) may be important for certain ecosystems. These are additional strong reasons for wanting to know the nature of groundwater pumped.
2. The choice of 200+/-100 m need a bit more elaboration. Yes, it is based on age-depth data (Supplementary Figure 1), but what fraction of fossil groundwater is then a meaningful threshold. How robust are the results against this choice?
3. Depth is used as a proxy for age. That is understandable. But the analysis on how age (in fact depth) changes with time (objective 2 of the paper and Figure 3) seems just a redoing of the work done in the authors' paper (ref 15). Why repeat it here and not just state it as a reference. The additional analysis done by looking at wells surrounding wells with proven fossil water is new and could be presented as a sole result for this objective.
4. The statement that fossil groundwater is not necessarily non-renewable is a bit problematic when referring to Margat et al. (2006). As I understand the definition that is used here is based on mean renewable time (volume of aquifer divided by mean recharge rate) being larger than 1000 years. For aquifers with fossil groundwater, mean renewal time is likely very large as well and therefore non-renewable. However, what the authors likely mean is that no groundwater depletion has to occur when fossil groundwater is pumped. Therefore, it is better to avoid the term "groundwater as a non-renewable resource" and state that the use of fossil groundwater does not have to mean that groundwater use is non-renewable, i.e. leads to taking water out of storage that cannot be replenished over human time scales.

Reviewer #2:

Remarks to the Author:

GebreMichel et al. present a national analysis of trends in presence and increased drilling of deep groundwater wells (greater than 100m) that can be inferred to be pumping fossil groundwater. I commend the authors on a paper that is well written and easy to follow. The figures are also well done and very clearly illustrate the findings. While I find the analysis interesting, I have two major concerns that make me question whether this manuscript is suitable for publication here.

My primary concern is that this manuscript seems to me to be incremental progress from a recent publications of the two co-authors. I am not convinced about what the novel finding is here that warrants publication in Nature-communications. Perrone & Jesechko (2019) published analysis of the same national database that is evaluated here which specifically quantified national trends in wells being drilled deeper as well as connections between deeper well drilling and hot spots for groundwater depletion (i.e. two of the three results sections of this paper). What is different about the work presented here is that it is focusing specifically on trends for wells that are deeper than 100 meters rather than the entire dataset. However, the findings here appear to be quite consistent with the previous publication hence my concern about the novelty here. I don't see any novel methods or approaches being applied beyond just focusing on a subset of wells that was used for the 2019 paper. The discussion sections cover a lot of interesting implications of what increased fossil groundwater pumping could mean but none of this is actually explored in the paper.

Secondly, the majority of this work is premised on the assumption that deep wells access fossil groundwater. I am okay with this assumption and I agree with the authors that the data simply does not exist for us to be able to anything different. The part that I'm concerned about here is the meta analysis of the mapping of fossil groundwater supplies in the US. It's not really clear how this fits into the paper. On one hand, the main purpose of this analysis here seems to be to support the assumption that deep wells access fossil groundwater. However that assumption is already supported by a global analysis published by Jesechko et al (2017). It's not clear if this is intended to be a new finding or just additional analysis to support this idea. On the other hand, at some points in the results and discussion the authors go further to treat this meta-analysis more as a mapping of fossil aquifers in the US. I'm concerned about that application because documenting where people have identified fossil groundwater in previous work is not enough to determine that it is not present in aquifers where it hasn't been documented. If the primary purpose of the meta analysis is just to support assumptions that deep wells access fossil groundwater then I think this discussion can be moved from the introduction to the methods. If the purpose is to map fossil groundwater then I think much more focus and attention to this is needed and it would also warrant its own results section at least (if not its own paper). Regardless it would be good to be more explicit about how this mapping it is different from the previous global analysis published by Jesechko et al (2017). I assume that this is a more updated and detailed map for the US but some discussion of how findings relate across these two studies would be good.

Detailed comments:

- I think the title should be revised to better match the results. What this analysis is showing is the increased drilling of wells at depth however there is no pumping data here and the authors point out that shallow wells are still much more prevalent so focusing on our 'reliance' of the water pumped from these wells seems to be a few steps removed from what's actually presented here.
- Be careful in the use of groundwaters plural and make sure to use groundwaters and groundwater consistently.
- Per my comment above, figure 1 and the discussion of USAD seem like a bit of a tangent at the end of the introduction. Perhaps this is a separate results section? Or maybe just part of the methods if you feel these relationships are previously well established and don't warrant a separate results section.
- Line 195 – can you be more explicit here on the method for identifying wells known to draw some fossil groundwater? Is this not the depth approach you have been using so far?
- The methods associated with Figure 3 are not very clear in the results section. They are much clear in the methods I think some revision to the results text would help clarify.
- From the description on line 196 it sounds like this analysis is done based on proximity to individual wells but it's not clear what the points on Figure 2 are actually representing. Are these individual wells or points grouped by aquifer? (Here too this is described more clearly in the methods section)
- Line 235 – This paragraph is confusing and think needs some additional text to discuss aquifer classification. Referring to some aquifers as fossil aquifers is misleading as the meta analysis (as far as I understand it) could only be used to identify aquifers that are fossil aquifers and there is no way to

know if the other aquifers are or aren't because there isn't data for them. Is that correct? If so I think the wording here is very important because currently the implication is that the aquifers not identified as fossil aquifers do not have fossil water and I doubt that is true.

- Figure 4 is really focused on the groundwater depletion trends. Given that this section is all about spatial connections between well drilling depths and water levels I would like to see some maps or figures that connect these spatial patterns in drawdown with the well trends. It's covered in the discussion but given the importance of this as one of the results sections I think some quantitative and graphic analysis is warranted here.

- Sections 3.2 and 3.3 explores many interesting implications that could build from this work. However, it seems like most of the discussion here is really on implications and not on the results themselves.

References:

- Perrone, D. & Jasechko, S. Deeper well drilling an unsustainable stopgap to groundwater depletion. *Nat. Sustain.* 2, 773–782 (2019).
- Jasechko, S. et al. Global aquifers dominated by fossil groundwaters but wells vulnerable to modern contamination. *Nat. Geosci.* 10, 425–429 (2017)

In this reply letter we detail how we have modified our manuscript to embed recommendations made by Reviewer 1 and Reviewer 2. Reviewer comments are shown in bold black text. Our replies are in light blue text; specific quotes from our manuscript are identified by light blue italicized text within quotation marks (e.g., *“example of the formatting for text quoted from our manuscript”*).

REVIEWER COMMENTS

Reviewer #1 (Remarks to the Author):

Review of Widespread and increasing reliance on fossil groundwater in the USA by Merhawi GebreEgziabher GebreMichael, Scott Jasechko and Debra Perrone

The authors use an impressive data set of well depth data, radioisotope groundwater age data and an updated USA aquifer database to assess the fraction of wells likely pumping fossil groundwater (>12000 years), how this fraction developed over time and if fossil groundwater use is related to groundwater depletion.

I really like this study, particularly because it depends on so much great data.

We thank Reviewer 1 for their detailed and helpful review. We have made numerous changes to the manuscript that were motivated directly from Reviewer 1's helpful recommendations; our manuscript is improved by these changes. Thank you.

A nice bonus for the hydrogeological community is the updated aquifer data set (USAD) that would benefit large-scale groundwater modelers and would attract quite some citations.

We look forward to the opportunity to making these geospatial data available.

This paper should be published after some of the following issues are resolved:

Thank you.

The first issue pertains to the rationale behind the importance of knowing whether groundwater pumped is fossil or not. This is based on differences in water quality focused on harmful constituents. I don't think that is a sufficiently strong argument. As discussed later, the concentration of harmful constituents (especially of geogenic nature) can be both higher and lower for fossil and non-fossil water, so what does it help us to know whether groundwater is fossil or not?

Thank you for your comment and question. We completed new analyses examining tens-of-thousands of groundwater quality measurements across hundreds of our aquifer systems. Our new analyses of dissolved arsenic, nitrate and total dissolved solids data are included in a new supplementary section (Supplementary Note 6.2) and are referenced in the main text. Specifically, these analyses—displayed in Supplementary Figs. 12-14—highlight the spatial variability in contaminant prevalence in deeper versus shallower aquifers. An example of one of these Supplementary Figures is reproduced below:

What is missing from this analysis is that groundwater age is a quality by itself. First, bottlers of mineral water, beer breweries and spas pride themselves of using age old water. This water therefore has an intrinsic and economic value. Second, if some of the fossil water outcrops in springs or wetlands or discharges into lowland streams, its specific quality (e.g., being calcium rich and nutrient poor) may be important for certain ecosystems. These are additional strong reasons for wanting to know the nature of groundwater pumped.

Thank you for your comment and suggestion that we expand our explanation of the rationale for fossil water research. We agree that doing so helps improve this manuscript.

To incorporate your suggestion to be more clear about these other important ramifications of fossil water research, we have added text to (i) the introduction and (ii) added a new Supplementary Table providing some examples of webpages that market fossil water.

(i) In the introduction of our revised manuscript, we added the following new text:

"Understanding where wells access fossil groundwater has implications beyond well water quality assessments. First, fossil groundwater can capture human interest, garnering intrinsic and economic values beyond those ascribed to younger water (Supplementary Note 7). Second, fossil groundwater that discharges at springs or into lowland streams can play a critical role in sustaining vulnerable ecosystems¹². Third, mapping wells that tap fossil aquifers can enable better understanding of the prevalence of communities that rely on fossil groundwater resources¹³. Although multiple studies have commented on the sustainability of fossil groundwater use^{14–18}, the spatiotemporal patterns of fossil groundwater use and groundwater depletion remain unclear, partly because of a lack geospatial data with locally relevant information for aquifers in the US."

(ii) We added the following Supplementary Table showing how some companies market fossil water:

Supplementary Table 18. Examples of companies marketing water on the basis of its old age

#	Category	Quote from webpage	Website	Date Accessed
1	Beer and spirits	"From deep within the Earth's crust, the finest drinking water on the planet springs forth to Memphis. This unique aquifer supplies WISEACRE with the most necessary of ingredients for the production of crisp, light-colored lager. The rains that fell to earth 3000 years ago are filtered very slowly through hundreds of feet of fine grain sand, culminating in a huge underground lake filled with 57 trillion gallons of virtually mineral-free water. From this prestigious water reserve, we supply Memphis with Sands, our one-of-a-kind Lager. Beauteous in its simplicity, it is very low in bitterness and full of delicious flavors of bread and crackery malt."	https://wiseacrebrew.com/memphis-sands	Jan. 6, 2022
2	Beer and spirits	"Founded in 2005 by Martin Townshend, this genuine real ale brewery is based in the leafy green heart of the nations hop growing region. As well as Nelson hops, Townshend's also has access to an aquifer (sic) of ancient water to brew with, drawn up from the Motueka aquifer."	https://townshendbrewery.co.nz/pages/story?_pos=1&_sid=af12fc6b2&_ss=r	Jan. 6, 2022
3	Beer and spirits	"Our water is a cherished natural resource, drawn straight from the Southern Hills Aquifer. These springs are over two thousand years old and provide water that is pristine and naturally perfect for brewing, which is why we decided to call the brand Spring Loaded"	https://beerconnoisseur.com/articles/abita-brewing-co-launches-spring-loaded-spiked-sparkling-water	Jan. 6, 2022
4	Bottled water	" Water "up to 30,000 years old" has become a McCashin's signature ingredient for the Nelson brewery's beer, ciders and soft drinks." and "The water source has been scientifically carbon-dated as being between 14,000 and 30,000 years old."	https://www.rnz.co.nz/news/business/299274/bored-old-water-makes-for-exclusive brews	Jan. 6, 2022
5	Bottled water	" Fiji Water comes from an ancient aquifer deep within the earth on the island of Viti Levu, where it is protected and preserved from external impurities and remains untouched by man—until you unscrew the cap."	https://www.wonderful.com/brands/fiji/	Jan. 6, 2022

6	Spa	" Hot spring water is considered old water, fossil water , ancient and irreplaceable. Heated by geothermal processes and emerging at 105° F., the water carries dissolved mineral salts reputed to have healing powers. The water contains calcium carbonate, calcium sulfate, sodium sulfate, sodium chloride, and lithium."	https://www.nps.gov/places/langford-hot-springs.htm	Jan. 7, 2022
7	Bottled water	"The water of the source emerges at the surface from a Well that is over 235 metres (770 feet) deep. Scientific studies have shown that this water can take up to 750 years to percolate through the 355 million year old limestones and dolomite fractures and joints before being drawn to the surface. The groundwater dissolves the minerals out of the limestones and dolomites to give Ballygowan its particular hydrochemical attributes."	https://www.ballygowan.ie/our-water-story/	Jan. 13, 2022
8	Bottled water	"Since the Roman times, healing properties have been attributed to water from St. Ann's Well, a geothermal spring in Buxton, Derbyshire in England. When the Romans arrived in Buxton, they built a bath around the spring, which they named Aquae Arnemetiae (The Waters of the Goddess of the Grove). The well, which taps 5,000-year-old rainwater that traveled across mineral-rich rock, was first mentioned as a holy well by William Worcester around 1460. "	https://aleteia.org/2020/10/01/bottled-mineral-water-comes-from-original-st-anns-holy-well-in-england/	Jan. 13, 2022

2. The choice of 200+/-100 m need a bit more elaboration. Yes, it is based on age-depth data (Supplementary Figure 1), but what fraction of fossil groundwater is hen a meaningful threshold. How robust are the results against this choice?

Thank you for your recommendation. To better convey the sensitivity analysis of the range of depths, we made two modifications.

(i) First, we added the following new text to the main text Methods:

"Specifically, more than ~30% of wells with depths at or exceeding 100 m contain detectable fossil water, and more than ~half (48%) of wells with depths at or exceeding 300 m contain detectable fossil water (Supplementary Fig. 1)."

(ii) Second, we revised Supplementary Fig. 1—now complete with annotations (in text boxes) to support our new statement within the main text:

We also added the following statement in the caption for Supplementary Fig. 1:

“Specifically, more than 30% of wells with depths at or exceeding 100 m contain detectable fossil water, and more than ~half (48%) of wells with depths at or exceeding 300 m contain detectable fossil water (defined as minimum ‘fraction of well water sample comprised of fossil groundwater’ values that exceed zero, determined on the basis of carbon isotope compositions; see ref. ¹).”

3. Depth is used as a proxy for age. That is understandable. But the analysis on how age (in fact depth) changes with time (objective 2 of the paper and Figure 3) seems just a redoing of the work done in the authors’ paper (ref 15). Why repeat it here and not just state it as a reference. The additional analysis done by looking at wells surrounding wells with proven fossil water is new and could be presented as a sole result for this objective.

Thank you for recommending we present our analysis looking at wells surrounding wells with proven fossil water as a sole result for this objective. We agree that this analysis is new and can be presented to meet our second objective. We have made considerable changes to our original manuscript to incorporate this suggestion; specifically, we have made the following updates in our revised manuscript:

(i) First ,we wrote a new Methods section devoted to our new main text results addressing our objective 2; we described our approach as follows:

“To test if the proportion of newly drilled wells in the US that tap fossil aquifers has increased over time we completed a series of steps. First, (i) we mapped the locations of wells identified in ref. 1 to pump fossil water (minimal fraction of well water comprised of fossil groundwater exceeds zero). Second, we identified all records of well construction within a 20 km radius of the well that is known to pump fossil groundwater. Third, (iii) for each well construction event, we determined whether the total depth of the constructed well is shallower or deeper than the nearby ‘fossil well’. That is, we compared the depth of each newly drilled well to that of the nearby ‘fossil well’, and describe the former in binary terms: (a) newly drilled well is shallower than the fossil well, or (b) newly drilled well is deeper than the fossil well. Fourth, (iv) for each calendar year where at least five wells were constructed within 20 km of the fossil well within a given year, we calculated the proportion of wells drilled in that year that are deeper than the fossil well, implying that many of these wells likely also pump fossil water, since they are deeper than a nearby well known to draw some fossil water (i.e., we calculated the fraction of wells drilled deeper than the well that is known to pump some fossil water for a given year). Fifth, (v) we calculated the Spearman rank correlation that describes variations in the fraction of wells that are deeper than the well that is known to pump fossil water varies versus calendar year (see Supplementary Fig. 2 for schematic). Last, we determine spatial statistics of these increasing and decreasing trends for a number of aquifer systems across the contiguous United States (Fig. 3).”

(ii) Second, we wrote a new main text results section detailing our main findings (section 2.2 of the revised manuscript). Our revised results section reads as follows:

“We tested for changes in the prevalence with which wells tap fossil aquifers over time. We analyzed temporal changes in the fraction of newly constructed wells drilled deeper than a nearby well that is known to draw fossil water. Specifically, we identified wells known to draw some fossil groundwater (>0% as calculated by ref. 1); fossil well waters can be identified by attributing low ^{14}C activities to radioactive decay, after accounting for other potential carbon sources on the basis of $^{13}\text{C}/^{12}\text{C}$ measurements (for further details see Supplementary Note 1.2 and the methods section within ref. 1). Next, we define ‘study areas’ as areas within a 20 km radius of a well that is known to draw fossil water (see Supplementary Fig. 2). We identified newly drilled wells that are located within each study area. We then calculated the proportion of newly drilled wells that have depths that are deeper than the nearby well that draws fossil water. Last, we quantified temporal variations in the frequency with which new wells are constructed deeper than the nearby well known to draw some fossil water (see Supplementary Fig. 2 for schematic of statistical analysis). We analyzed five time intervals: 1950-1975, 1975-2000, 2000-2015, 1950-2015 and 1975-2015.

We show that the proportion of wells tapping fossil aquifers has increased over time in more places than it has decreased over time (Table 1). Specifically, we show that study areas where the proportion of wells tapping fossil aquifers has increased over time are ~ 1.4 - 3.4 times more common than study areas where the proportion of wells tapping fossil aquifers has decreased over time (Table 1). We conclude that the proportion of newly drilled wells that are sufficiently deep to tap fossil aquifers has increased over time in more places than it has decreased in the US.

On an aquifer-by-aquifer basis, we identified 36 aquifer systems that contain at least five wells known to draw fossil water with sufficient nearby (<20 km) well drilling data for analyses. We find evidence for an increase over time in the proportion of wells tapping fossil aquifers in the Black Warrior River Aquifer System, the Central High Plains and California’s Santa Rosa Valley (all of which have median rank correlation coefficients exceeding zero for all studied time intervals; Fig. 3).”

(ii) Third, we revised our supplementary information to report on our analysis of temporal variations in the proportion of wells deeper than 200 ± 100 m. As Reviewer 2 points out, this analysis has similar components to Perrone and Jasechko (2019; cited in paper) but differs as ***“it is focusing specifically on trends for wells that are deeper than 100 meters rather than the entire dataset.”*** (quote Reviewer 2 introductory comments). An additional distinction of this analysis from the previous analysis (Perrone and Jasechko, 2019) is that here we report our results in the context of our newly digitized aquifer system database (US Aquifer Database, reported for the first time here), rather than on the basis of (a) 10-kilometer-by-10-kilometer areas, (b) 100-km-by-100-km-areas, or (c) irregular polygons designed to each contain ~ 100 wells, which were the three types (i.e., aforementioned (a)-(c)) of study areas reported in that study. Therefore, we retain our aquifer-system-scale analysis as a supplementary, but we ensure that the primary basis for our main results is now the analyses based on 20 km radii around a well known to pump fossil water, rather than the aquifer-by-aquifer analysis. Our revised supplementary information contains the following text (and two supplementary figures):

“To further explore fossil groundwater access over time, we analyzed changes in the fraction of newly constructed wells with depths exceeding 200 ± 100 m over five different time intervals: 1950-1975, 1975-2000, 2000-2015, 1950-2015, and 1975-2015.

We evaluated temporal variations in the proportion of newly constructed wells that likely tap fossil aquifers following several steps. First, we identified all groundwater well completion records that present both a completion date (e.g., well completed on January 3, 1982) and a depth (e.g., 34 meters below ground). Next, we completed a spatial join to identify all wells within a single aquifer system (as defined in our US Aquifer Database). Then, for any calendar year within which at least 5 well completion records exist, we calculated the [proportion of all newly constructed wells that have a depth exceeding a ‘threshold depth’] (where ‘threshold depth’ is 100 m, 200 m or 300 m, a set of threshold depths that encompass the broad range at which many aquifer systems transition from young water (shallow) to fossil water (deep)). Last, we completed non-parametric regressions of the rank transforms of [well completion year] versus the rank transforms of [the proportion of all newly constructed wells that have a depth exceeding a ‘threshold depth’].

We only consider cases where all of the following criteria are met: (a) at least one calendar year met our criteria for analysis (i.e., at least five drilled wells) in the first five years of a studied time interval (e.g., for analyses of the time interval 1950-1975, we require at least one of the following five calendar years to meet our criteria for analysis: 1950, 1951, 1952, 1953 or 1954), (b) at least one calendar year met our criteria for analysis (i.e., at least five drilled wells) in the final five years of a studied time interval; and, (c) a minimum of at least five calendar years within the time interval met our criteria for analyses. The correlations were determined for five distinct time intervals: (i) 1950-1975, (ii) 1975-2000, (iii) 2000-2015, (iv) 1950-2015, and (v) 1975-2015. Positive Spearman rank correlation coefficients (ρ) imply that the proportion of drilled wells that are deeper than 100 m, 200 m, or 300 m (many of which are also likely pump fossil water) has increased over time (Supplementary Table 2 and Supplementary Fig. 3). Negative Spearman rank correlation coefficients imply that the proportion of drilled wells that are deeper than 100 m, 200 m, or 300 m has declined over time (Supplementary Fig. 3). Results for each time interval (i.e., the prevalence of blue versus red diamonds in Supplementary Fig. 3) are summarized in Supplementary Fig. 4.”

4. The statement that fossil groundwater is not necessarily non-renewable is a bit problematic when referring to Margat et al. (2006). As I understand the definition that is used here is based on mean renewable time (volume of aquifer divided by mean recharge rate) being larger than 1000 years. For aquifers with fossil groundwater, mean renewal time is likely very large as well and therefore non-renewable. However, what the authors likely mean is that no groundwater depletion has to occur when fossil groundwater is pumped. Therefore, it is better to avoid the term “groundwater as a non-renewable resource” and state that the use of fossil groundwater does not have to mean that groundwater use is non-renewable, i.e. leads to taking water out of storage that cannot be replenished over human time scales.

Thank you for your helpful feedback. We have made the following updates to our manuscript to avoid stating that fossil groundwater is not necessarily non-renewable. Specifically, we:

(i) We replaced our previous text in section 3.3 (which previously stated *“fossil groundwater is not necessarily a non-renewable resource”*). The full revised sentence in our revised manuscript reads as follows:

“Our finding that fossil-groundwater-use hotspots do not always co-occur with groundwater-depletion hotspots reinforces the point that the use of fossil groundwater does not have to mean that groundwater use is non-renewable (see “Fossil Groundwater” in Table 1 within ref. ¹⁵).“

(ii) We also changed another sentence in our original manuscript to capture the recommended wording from Reviewer 1. Specifically, our original manuscript stated: *“Although fossil groundwaters are not necessarily non-renewable resources, fossil groundwaters tend to be more common in deep aquifers (Supplementary Fig. 1)”* and our revised manuscript now reflects the reviewer’s recommendation, reading as follows in our revised manuscript:

“Although the use of fossil groundwater does not have to mean that groundwater use is non-renewable, fossil groundwater does tend to be more common in deeper aquifers”

(iii) We also added a new paragraph describing a new statistical analysis we completed in response to reviewer comments that examines, quantitatively, statistical relationships between groundwater level change over time and the fraction of wells in an aquifer that are deeper than 200+/-100 m. Within this paragraph, we included the recommended wording provided in this comment by Reviewer 1. Specifically, our revised manuscript contains the following statement:

“We did not identify a consistently positive (or negative) correlation coefficient describing variations between the prevalence of deep wells and groundwater level changes over time (Supplementary Table 17), reinforcing that the use of fossil groundwater does not have to mean that groundwater use is non-renewable.”

(iv) Last, to ensure the Reviewer’s recommendation is clear to potential readers of our manuscript, we revised our abstract and now include the following statement:

“However, this widespread and increased drilling of wells into fossil aquifers is not necessarily associated with groundwater depletion, emphasizing that the presence of fossil groundwater does not necessarily indicate a non-renewable water supply.”

Reviewer #2 (Remarks to the Author):

GebreMichel et al. present a national analysis of trends in presence and increased drilling of deep groundwater wells (greater than 100m) that can be inferred to be pumping fossil groundwater. I commend the authors on a paper that is well written and easy to follow. The figures are also well done and very clearly illustrate the findings.

We thank Reviewer 2 for their positive comments and for their detailed and highly helpful review of our manuscript. Below we detail how we have substantially updated our analyses to include recommendations made by Reviewer 2 into our revised manuscript.

While I find the analysis interesting, I have two major concerns that make me question whether this manuscript is suitable for publication here. My primary concern is that this manuscript seems to me to be incremental progress from a recent publication of the two co-authors. I am not convinced about what the novel finding is here that warrants publication in Nature-communications. Perrone & Jasechko (2019) published an analysis of the same national database that is evaluated here which specifically quantified national trends in wells being drilled deeper as well as connections between deeper well drilling and hot spots for groundwater depletion (i.e. two of the three results sections of this paper). What is different about the work presented here is that it is focusing specifically on trends for wells that are deeper than 100 meters rather than the entire dataset. However, the findings here appear to be quite consistent with the previous publication hence my concern about the novelty here. I don't see any novel methods or approaches being applied beyond just focusing on a subset of wells that was used for the 2019 paper.

We thank Reviewer #2 for their encouragement to better highlight how our manuscript differs from Perrone and Jasechko (2019). We agree that it is important that we highlight how our work goes beyond incremental progress, and clarifies the novelty of our findings and how they stand apart from previous publications. We have added the following statements to the introduction section of our revised manuscript to specify how our work differs from previous work.

“Compared to other countries, the US has relatively dense well water radioisotope measurements¹. Similarly, the quality of US well construction depth data is relatively good¹⁹. These two continental-scale datasets have never been merged at continental-scale, but could provide an opportunity to analyze spatiotemporal patterns of wells of sufficient depth to tap fossil groundwater if merged with a new geospatial dataset that has locally relevant information about aquifers in the US (Fig. 1a). The lack of locally relevant aquifer geospatial data has constrained the number of aquifer systems evaluated in previous studies^{1,19}, highlighting an opportunity to explore fossil water access more systematically now that these geospatial data are available.

Here, we combine (a) radioisotope-based fossil groundwater prevalence data¹, (b) well drilling data¹⁹, (c) groundwater level data, and (d) a novel geodatabase consisting of ~440 aquifer systems, providing locally relevant study areas in the contiguous US. Together, these data are used to meet three objectives.

To ensure we have made clear how our study differs and goes well beyond the scope of previous studies—e.g., by examining hundreds of aquifers rather than fewer than 10 (cf. Perrone and Jasechko, 2019) and applying a new methodology (Fig. 3 of revised manuscript)—we added another statement to the Methods:

“The previous lack of locally relevant and continent-wide aquifer geospatial data necessitated that the scope of previous well completion depth studies¹⁹ was limited to only a few expansive aquifer systems, rather than the novel geodatabase consisting of ~440 aquifer systems delineated here. Pairing these US well drilling data¹⁹ and radioisotope-based fossil groundwater prevalence data¹ to our new aquifer geodatabase (Fig. 1a) and a new method designed to explore well drilling depth changes over time surrounding sites where fossil water has been identified (see Methods section entitled “Fossil groundwater accessed more frequently over time (Fig. 3)”) distinguish this study from previous works.”

The discussion sections cover a lot of interesting implications of what increased fossil groundwater pumping could mean but none of this is actually explored in the paper.

We added substantial new analyses of groundwater quality to our work. Specifically, we (i) completed geospatial analyses of the vertical distribution of groundwater quality in hundreds of study aquifers and examining three analytes (each relevant to a different paragraph in our main text Discussion section 3.2), and (ii) made numerous additions to our main text to make reference to our new groundwater quality analyses.

(i) We added the following new Supplementary Note 6 describing an extensive analysis of tens-of-thousands of groundwater quality measurements in our study aquifer systems:

“Supplementary Note 6 – Groundwater quality in shallow versus deep wells

Supplementary Note 6.1 – Compilation of US groundwater quality data

Here we describe how groundwater quality data for the contiguous US were downloaded. We data from <https://www.waterqualitydata.us/#siteType=Well&mimeType=csv&sorted=no&providers=NWIS&providers=STEWARDS&providers=STORET>; specifically, we entered the following terms in the search field entitled “Characteristics”: “Arsenic (NWIS, STEWARDS, STORET)”, “Nitrate (NWIS, STORET)”, “Total Dissolved Solids (NWIS, STEWARDS, STORET)” (complete query URLs (entered January 11, 2022) are:

<https://www.waterqualitydata.us/#siteType=Well&characteristicName=Nitrate&mimeType=csv&sorted=no&dataProfile=resultPhysChem&providers=NWIS&providers=STEWARDS&providers=STORET> and

<https://www.waterqualitydata.us/#siteType=Well&characteristicName=Arsenic&mimeType=csv&sorted=no&dataProfile=resultPhysChem&providers=NWIS&providers=STEWARDS&providers=STORET> and

<https://www.waterqualitydata.us/#siteType=Well&characteristicName=Total%20dissolved%20solids&mimeType=csv&sorted=no&dataProfile=resultPhysChem&providers=NWIS&providers=STEWARDS&providers=STORET>

Here we detail steps to quality check each dataset: (i) arsenic, (ii) nitrate, (iii) total dissolved solids.

(i) To quality control the downloaded arsenic dataset, we excluded measurements returning a below detection result if the detection limit equaled or exceed 10 µg/L. We also any measurement that does not include the term “Dissolved” in the field entitled “ResultSampleFractionText (this step excluded measurements of “Total” arsenic, as we concluded that including “Total” arsenic measurements in our dataset would implicitly and inappropriately treat dissolved and total arsenic measurements as equivalent in our analyses). We excluded any measurements with a “ActivityMediaSubdivisionName” value other than “Groundwater”. A small number of measurements (n=7) reported “Present Above Quantification Limit”; these samples were excluded from further analyses, as we did not analyze data for an upper limit value for the analytical quantification limit. We excluded measurements with the flag “Detected Not Quantified” in the field “ResultDetectionConditionText”. Well depth data were compiled from the field entitled “WellDepthMeasure/MeasureValue” first and, if this field was empty, we compiled well depth data from the field entitled “WellHoleDepthMeasure/MeasureValue”. For stations with multiple measurements meeting the aforementioned quality control criteria, we analyzed only the most recent measurement.

(ii) To quality control the downloaded nitrate dataset, we excluded all measurements that does not include the term “Dissolved” in the field entitled “ResultSampleFractionText”. We excluded measurements that do not clearly specify sample media as “Groundwater” in the field “ActivityMediaSubdivisionName”. We excluded records reporting “Detected Not Quantified” or “Systematic Contamination” under the field “ResultDetectionConditionText”. We excluded measurements if the units did not specify ‘as NO₃’ or ‘as N’ (i.e., some records specify only “mg/l” or ‘ug/l’ or ‘ppm’, and were thus deleted as this information was insufficient to determine if the units are ‘as NO₃’ or ‘as N’). We converted nitrate data into consistent units of “nitrate as N” (as some measurements and stated detection limits are recorded in units of “mg/l as NO₃”). We excluded measurements recording a below detection limit measurement if the stated detection limit equaled or exceeded 10 mg/L NO₃- as N. Well depth data were compiled from the field entitled “WellDepthMeasure/MeasureValue” first and, if this field did not contain a number, we compiled well depth data from the field entitled “WellHoleDepthMeasure/MeasureValue” instead. For stations with multiple measurements meeting the aforementioned quality control criteria, we analyzed only the most recent measurement.

(iii) To quality control total dissolved solids (TDS) data, we excluded all records that does not specify “Dissolved” in the field “ResultSampleFractionText”. We excluded measurements that do not state “Groundwater” under the field entitled “ActivityMediaSubdivisionName”. We deleted any records with a below detection measurement if the detection limit was stated to be equal to or greater than 1000 mg/L. We excluded records reporting “Detected Not Quantified” or “Systematic Contamination” or “Present above Quantification Limit” under the field “ResultDetectionConditionText”. We converted units of “tons/ac ft” to mg/L by multiplying by the 735.47. We excluded a records reporting measurements in units of “tons/day”. Well depth data were compiled from the field entitled “WellDepthMeasure/MeasureValue” first and, if this field was empty, we compiled well depth data from the field entitled “WellHoleDepthMeasure/MeasureValue”. For stations with multiple measurements meeting the aforementioned quality control criteria, we analyzed only the most recent measurement.

Supplementary Note 6.2 – Groundwater quality in deep and shallow wells in aquifer systems

The following figures provide, for each of our three study analytes (dissolved arsenic, dissolved nitrate, total dissolved solids), information about the spatial distribution of well waters that exceed the following analyte-specific threshold concentrations: Dissolved arsenic (As): threshold concentration of 10 µg/L; Dissolved nitrate (NO₃): threshold concentration of 10 mg/L (measured as N; i.e., 10 mg/L NO₃-N); and, Total dissolved solids (TDS): threshold concentration of 3,000 mg/L.

The following figures’ panels provide the following information:

- a) Panel (a) presents a map of well water quality measurements made in wells with total depths shallower than 50 m. Colored points are those that exceed the analyte-specific concentration threshold (for example, a well water sample with more than 10 mg/L NO₃-N).
- b) Panel (b) presents a map of aquifers shaded by the fraction of shallow wells with groundwater quality data (<50 m; see panel (a)) that have concentrations exceeding the analyte-specific threshold (e.g., the proportion of wells within the aquifer system that have > 10 mg/L NO₃-N). We only shade areas for aquifer systems with at least n=20 water quality measurements.
- c) Panel (c) presents a map of well water quality measurements made in wells with total depths deeper than 100 m and shallower than 300 m. Colored points are those that exceed the analyte-specific concentration threshold (e.g., a well water sample with more than 10 mg/L NO₃-N).
- d) Panel (d) presents a map of aquifers shaded by the fraction of deep wells with groundwater quality data (100-300 m; see panel (c)) that have concentrations exceeding the analyte-specific threshold (e.g., the proportion of wells within the aquifer system that have > 10 mg/L NO₃-N). We only shade areas for aquifer systems with at least n=20 water quality measurements.

e) Panel (e) plots the fraction of wells that have concentrations exceeding the analyte-specific threshold. Downward-pointing dark blue triangles represent wells with depths between 100-300 m (i.e., values used to shade aquifers in panel d), whereas upward-pointing light blue triangles represent wells with depths shallower than 50 m (i.e., values used to shade aquifers in panel b). We only plot aquifer systems with at least $n=20$ water quality measurements for both shallow (<50 m) and deep (100-300 m) well depth intervals (i.e., aquifers that appear both panel (b) and panel (d)).”

Supplementary Fig. 12. Spatial distributions of dissolved arsenic in US well waters. (a) Well water arsenic measurements in wells with total depths shallower than 50 m; purple colored points are those that exceed 10 $\mu\text{g/L}$. (b) Aquifers shaded by the fraction of shallow wells (<50 m; see panel (a)) that have arsenic concentrations exceeding 10 $\mu\text{g/L}$. (c) Well water arsenic measurements in wells with total depths deeper than 100 m and shallower than 300 m; purple colored points are those that exceed 10 $\mu\text{g/L}$. (d) Aquifers shaded by the fraction of deep wells (100–300 m; see panel (c)) that have arsenic concentrations exceeding 10 $\mu\text{g/L}$. (e) The fraction of wells that have arsenic concentrations exceeding 10 $\mu\text{g/L}$ for individual aquifer systems (see labels at base of figure). Downward-pointing dark blue triangles represent wells with depths between 100–300 m (i.e., values used to shade aquifers in panel (d)), whereas upward-pointing light blue triangles represent wells with depths shallower than 50 m (i.e., values used to shade aquifers in panel (b)). Dashed vertical black lines mark differences between shallow wells (<50 m; upward-pointing light blue triangles) and deeper wells (100–300 m; downward-pointing dark blue triangles) in their respective frequencies with which these two groups of wells (shallow, deep) have high-arsenic concentrations (>10 $\mu\text{g/L}$).

Supplementary Fig. 13. Spatial distributions of dissolved nitrate (NO_3) in US well waters. (a) Well water NO_3 measurements in wells with total depths shallower than 50 m; green colored points are those that exceed 10 mg/L (NO_3 as N). (b) Aquifers shaded by the fraction of shallow wells (<50 m; see panel (a)) that have nitrate concentrations exceeding 10 mg/L. (c) Well water nitrate measurements in wells with total depths deeper than 100 m and shallower than 300 m; green colored points are those that exceed 10 mg/L. (d) Aquifers shaded by the fraction of deep wells (100-300 m; see panel (c)) that have nitrate concentrations exceeding 10 mg/L. (e) The fraction of wells that have nitrate concentrations exceeding 10 mg/L for individual aquifer systems (see labels at base of figure). Downward-pointing dark blue triangles represent wells with depths between 100-300 m (i.e., values used to shade aquifers in panel (d)), whereas upward-pointing light blue triangles represent wells with depths shallower than 50 m (i.e., values used to shade aquifers in panel (b)). Dashed vertical black lines mark differences between shallow wells (<50 m; upward-pointing light blue triangles) and deeper wells (100-300 m; downward-pointing dark blue triangles) in their respective frequencies with which these two groups of wells (shallow, deep) have high- NO_3 (>10 mg/L).

Supplementary Fig. 14. Spatial distributions of total dissolved solids (TDS) in US well waters. (a) Well water TDS measurements in wells with total depths shallower than 50 m; red colored points are those that exceed 3,000 mg/L. (b) Aquifers shaded by the fraction of shallow wells (<50 m; see panel (a)) that have TDS concentrations exceeding 3,000 mg/L. (c) Well water TDS measurements in wells with total depths deeper than 100 m and shallower than 300 m; red colored points are those that exceed 3,000 mg/L. (d) Aquifers shaded by the fraction of deep wells (100-300 m; see panel (c)) that have TDS concentrations exceeding 3,000 mg/L. (e) The fraction of wells that have TDS concentrations exceeding 3,000 mg/L for individual aquifer systems (see labels at base of figure). Downward-pointing dark blue triangles represent wells with depths between 100-300 m (i.e., values used to shade aquifers in panel (d)), whereas upward-pointing light blue triangles represent wells with depths shallower than 50 m (i.e., values used to shade aquifers in panel (b)). Dashed vertical black lines mark differences between shallow wells (<50 m; upward-pointing triangles) and deeper wells (100-300 m; downward-pointing triangles) in their respective frequencies with which these two groups of wells (shallow, deep) have high-TDS (>3,000 mg/L)."

(ii) We added new text to our discussion highlighting findings from our analysis of groundwater quality measurements in US aquifer systems. Specifically, we added the following statement to our Discussion section (3.2) paragraph focused on surface-borne pollutants:

“We analyzed tens-of-thousands of dissolved nitrate measurements in US wells (Supplementary Note 6); we found that shallow wells (<50 m) tend to have high nitrate concentrations (>10 mg/L NO₃ as N) more frequently than deeper wells (100-300 m) in most of the aquifer systems we studied (Supplementary Fig. 13).”

Further, we modified the following portion of our discussion section devoted to geogenic contaminants to read as follows:

“We analyzed tens-of-thousands of dissolved arsenic measurements in shallow (<50 m) and deep (100-300 m) wells in the US (Supplementary Fig. 12); we found an approximately equal number of aquifer systems where shallow wells are contaminated by arsenic more frequently than deep wells (n=47 aquifers) as the number of aquifer systems where deep wells are contaminated by arsenic more frequently than shallower wells (n=54).

Our finding that fossil groundwater reliance is increasing as deep wells become more common may imply concomitant changes in exposure to high-arsenic groundwater; however, we stress that arsenic exposure assessments should be considered on an aquifer-by-aquifer basis, because of the importance of local hydrochemical conditions and hydrostratigraphy in determining groundwater arsenic concentrations⁵⁵.”

Last, we also added the following to our discussion of groundwater salinity in the main text:

“However, we stress that there are also aquifer systems where shallower wells are more likely to pump brackish water than deeper wells; our analysis of hundreds-of-thousands of total dissolved solids measurements identified a dozen such cases, most of which are arid alluvial basins in the western US (Supplementary Fig. 14). The high spatial variability in the statistical relationship between well water salinity and well depth highlights the importance of considering local hydrogeologic settings and historic land uses when examining connections between increased well drilling into fossil aquifers and the potential threat of salinity to groundwater users.”

Secondly, the majority of this work is premised on the assumption that deep wells access fossil groundwater. I am okay with this assumption and I agree with the authors that the data simply does not exist for us to be able to anything different.

Thank you for your comment. We agree that depth is the best available option for analyzing fossil groundwater access by wells. We have further refined the age-depth relationship in response to this comment and a recommendation from Reviewer 1. Specifically, we have updated our Supplementary Fig. 1 to provide metrics as to the depth threshold of 100-300 m that we apply in our study. Our revised Supplementary Fig. 1 is shown here:

The part that I'm concerned about here is the meta analysis of the mapping of fossil groundwater supplies in the US. It's not really clear how this fits into the paper. On one hand, the main purpose of this analysis here seems to be to support the assumption that deep wells access fossil groundwater. However that assumption is already supported by a global analysis published by Jasechko et al (2017). It's not clear if this is intended to be a new finding or just additional analysis to support this idea.

On the other hand, at some points in the results and discussion the authors go further to treat this meta-analysis more as a mapping of fossil aquifers in the US. I'm concerned about that application because documenting where people have identified fossil groundwater in previous work is not enough to determine that it is not present in aquifers where it hasn't been documented.

If the primary purpose of the meta analysis is just to support assumptions that deep wells access fossil groundwater then I think this discussion can be moved from the introduction to the methods.

If the purpose is to map fossil groundwater then I think much more focus and attention to this is needed and it would also warrant its own results section at least (if not its own paper).

Thank you for your comments and helpful recommendation. We agree that we should have been clearer about the role of the meta-analysis. The primary purpose of the meta-analysis is to support the assumption that deep wells access fossil groundwater (i.e., the underlined text within Reviewer 2’s comment is correct), which is critical to the methodology developed within this paper.

We have followed Reviewer 2’s recommendation to move text from our original introduction section that described the meta-analysis. Specifically, our original manuscript stated “We meet our objectives by combining four databases: (a) fossil groundwater prevalence determined on the basis of our own analysis of groundwater radiocarbon data¹ and a meta-analysis of fossil groundwater occurrence (Fig. 1; Methods section entitled “Meta-analysis of fossil groundwater prevalence (Fig. 1)”)”; our revised manuscript deletes reference to the meta-analysis here and instead states the following: “We meet our objectives by combining three databases: (a) records of groundwater well depths...” (i.e., no reference to the meta-analysis in this portion of the introduction section that describes databases).

Regardless it would be good to be more explicit about how this mapping it is different from the previous global analysis published by Jasechko et al (2017). I assume that this is a more updated and detailed map for the US but some discussion of how findings relate across these two studies would be good.

Thank you. We have made two updates to our manuscript to better discuss of how findings relate across these two studies:

(i) First, we added the following sentence to our Fig. 1 caption to direct readers toward new Supplementary Figures that make clear how our meta-analysis differs from ref. ¹.

“For a comparison of the spatial distribution of fossil aquifers identified by ref. 1 and those identified in our meta-analysis see Supplementary Figs. 9 and 10.”

(ii) We updated Fig. 1a in the main text to provide a combination of data from ref. 1 and our latest meta-analysis (from Supplementary Table 1). We also included the specific locations of fossil well waters from ref. 1, as the display of these points helps us convey the information within our revised manuscript’s Fig. 3 (as it is based on 20 km radial distances from these points). Our revised Fig. 1 is shown here:

Detailed comments:

- I think the title should be revised to better match the results. What this analysis is showing is the increased drilling of wells at depth however there is no pumping data here and the authors point out that shallow wells are still much more prevalent so focusing on our 'reliance' of the water pumped from these wells seems to be a few steps removed from what's actually presented here.

Thank you for recommending we change the title and specifically the word 'reliance'. We agree. We have changed our previous manuscript's title (which was "*Widespread and increasing reliance on fossil groundwater in the USA*") to, instead, "*Widespread and increased drilling of wells into fossil aquifers in the USA*"

- Be careful in the use of groundwaters plural and make sure to use groundwaters and groundwater consistently.

Thank you for your feedback. We agree that we should have been more careful with our use of the plural term 'groundwaters' in our original manuscript. We have modified our manuscript; specifically, here are examples of corrections we made to our manuscript

- In the abstract, our original manuscript contained the sentence "*Most groundwaters have been under the ground more than ~12 thousand years*". Our revised manuscript states, instead, "*Most stored groundwater is 'fossil' in its age, having been under the ground for more than ~12 thousand years.*"
- Also, in its abstract, our original manuscript stated "*Because these fossil groundwaters often have distinct concentrations of contaminants relative to younger groundwaters*". Our revised manuscript has been revised to avoid the plural word 'groundwaters', and now reads as follows: "*Because fossil groundwater often has distinct concentrations of contaminants relative to younger groundwater*"
- In the introduction section of our original manuscript, we previously stated "*Fossil groundwaters—defined as waters that have been underground for more than 12,000 years—likely comprise more than half of global groundwater stored within 1000 m of the land surface*". We have revised our manuscript to avoid plural 'groundwaters', and now state the following instead: "*Fossil groundwater—defined as groundwater that has been underground for more than 12 thousand years—likely comprises more than half of global groundwater stored within 1000 m of the land surface*"
- Our original manuscript's introduction stated "*Studies reporting radioisotope measurements have identified fossil groundwaters in over one hundred aquifers around the globe*". We have revised our manuscript to read as follows instead: "*Radioisotope measurements have identified fossil groundwater in over one hundred aquifers around the globe*"
- Our previous (initial submission) manuscript contained the following statement: "*..little is known about the prevalence of wells that pump fossil groundwaters..*"; we have revised this statement to avoid use of 'groundwaters', and our revised manuscript reads as follows: "*little is known about the prevalence of wells that pump fossil groundwater*"
- Our previous manuscript stated the following: "*Dozens of publications that report on the occurrence of fossil groundwaters in US aquifers support this finding—that fossil groundwaters are common at depth in many aquifer systems—and highlight that fossil groundwaters occur in a wide variety of hydrogeologic settings*". We have revised this statement to avoid use of the term 'groundwaters' in all three cases; our revised manuscript reads as follows: "*Our meta-analysis of studies that report on the occurrence of fossil groundwater in US aquifers supports this finding (Fig. 1 panels b-m), and highlights that fossil groundwater occurs in a wide variety of hydrogeologic settings*"
- Our initial submission stated the following (in the Results section): "*where fossil groundwaters are common*"; our revised manuscript states the following instead "*where fossil groundwater is common*"

- **Our original manuscript stated that** *“This complementary analysis supports our finding: there is an increase in the proportion of wells being constructed to deep depths where fossil groundwater is common in the majority of aquifers that we studied”*
- **Our original manuscript stated that** *“For example, in California’s Cuyama Valley (Fig. 1L), we find that 16-63% of wells are deeper than 200±100 m, that low-¹⁴C groundwaters have been identified in wells deeper than 200±100 m”*; **our revised manuscript states the following instead:** *“For example, in California’s Cuyama Valley (Fig. 1L), we find that 16-63% of wells are deeper than 200±100 m, that low-¹⁴C groundwater samples have been collected from wells deeper than 200±100 m”*
- **Our original manuscript contained the following statement in its section 2.3:** *“For example, the Black Warrior River Aquifer System (eastern Mississippi through Alabama; Fig. 1h) has a similar percentage of wells that are deeper than 200±100 m as the Cuyama Valley (5-44%) and also contains low-¹⁴C groundwaters indicative of fossil groundwater”*. **We revised this statement to avoid the term ‘groundwaters’; our revised manuscript reads as follows:** *“For example, the Black Warrior River Aquifer System (eastern Mississippi through Alabama; Fig. 1h) has a similar percentage of wells that are deeper than 200±100 m as the Cuyama Valley (5-44%) and also contains groundwater with low-¹⁴C activities that are indicative of fossil groundwater”*
- **Our original manuscript stated** *“Surface-borne pollutants contaminate groundwater resources when they percolate downward through the soil profile and unsaturated zone to enter the groundwater from above. Fossil groundwaters tend to have lower concentrations of surface-borne pollutants than younger groundwaters. For example, nitrate—a common surface-borne pollutant that is frequently associated with confined animal feeding operations, excessive fertilization and inadequate sanitation^{41,42}—is more common in recently recharged (i.e., ‘younger’) groundwaters than in older groundwaters”*; **we revised our manuscript and replaced the word ‘groundwaters’ with ‘groundwater’.** **Our revised manuscript reads as follows:** *“Surface-borne pollutants contaminate groundwater resources when they percolate downward through the soil profile and unsaturated zone to enter the groundwater from above. Fossil groundwater tends to have lower concentrations of surface-borne pollutants than younger groundwater. For example, nitrate—a common surface-borne pollutant that is frequently associated with confined animal feeding operations, excessive fertilization and inadequate sanitation^{41,42}—is more common in recently recharged (i.e., ‘younger’) groundwater than in older groundwater”*
- **Our original manuscript stated** *“Geogenic groundwater contamination can arise due to the interactions of groundwaters with the mineral skeleton of the geologic formations that they flow through and reside within. For example, arsenic is a common geogenic pollutant that poses a challenge to the provision of fresh drinking water in multiple aquifer systems across the world⁵¹. Aqueous arsenic concentrations can differ between fossil versus younger groundwaters, though the statistical relationships between groundwater age and arsenic concentrations vary among aquifer systems and are critically dependent on hydrostratigraphy, redox conditions and flow path architectures^{52,53}. For example, in California’s Cuyama Valley, fossil groundwaters pumped from deep wells tend to have higher arsenic concentrations than younger groundwaters drawn from shallower wells²⁹. Conversely, in the Bengal Basin (Bangladesh), groundwaters deeper than ~100 m tend to have lower arsenic concentrations than some shallower groundwaters⁵⁴. Our finding that fossil groundwater reliance is increasing as deep wells become more common may imply concomitant changes in exposure to high-arsenic groundwaters”*; **we revised our manuscript by replacing all uses of the term ‘groundwaters’; our revised manuscript reads as follows:** *“Geogenic contamination can arise as groundwater interacts with the mineral skeleton of the geologic formations that it flows through and resides within. For example, arsenic is a common geogenic pollutant that poses a challenge to the provision of fresh drinking water in multiple aquifer systems across the world⁵². Aqueous arsenic concentrations can differ between fossil versus younger groundwater, though the statistical relationships between groundwater age and arsenic concentrations vary among aquifer systems and are critically*

dependent on hydrostratigraphy, redox conditions and flow path architectures^{53,54}. For example, in California's Cuyama Valley, samples of fossil groundwater pumped from deep wells tend to have higher arsenic concentrations than samples of younger groundwater drawn from shallower wells³⁰. Conversely, in the Bengal Basin (Bangladesh), groundwater found at depths of deeper than ~100 m tends to have lower arsenic concentrations than groundwater found at shallower depths⁵⁵. We analyzed tens-of-thousands of dissolved arsenic measurements in shallow (<50 m) and deep (100-300 m) wells in the US (Supplementary Fig. 12); we found an approximately equal number of aquifer systems where shallow wells are contaminated by arsenic more frequently than deep wells (n=47 aquifers) as the number of aquifer systems where deep wells are contaminated by arsenic more frequently than shallower wells (n=54).

- Our finding that fossil groundwater reliance is increasing as deep wells become more common may imply concomitant changes in exposure to high-arsenic groundwater; however, we stress that arsenic exposure assessments should be considered on an aquifer-by-aquifer basis, because of the importance of local hydrochemical conditions and hydrostratigraphy in determining groundwater arsenic concentrations⁵⁵."
- **The discussion section of our original manuscript stated** "Beyond arsenic, elevated activities of naturally occurring radioisotopes (e.g., ²²⁶Ra, ²²⁸Ra) have been identified in some fossil groundwaters, including those within the Disi Sandstone Aquifer of Jordan⁵⁷, the Saq Aquifer System and the Mega Aquifer System of the Arabian Peninsula^{58,59}, and the Nubian Sandstone Aquifer System of northeastern Africa⁶⁰. Careful consideration of treatment options (e.g., blending, reverse osmosis) may be warranted where these fossil groundwaters are tapped for household use."; **we revised our manuscript to replace the plural term 'groundwaters' with other options; the revised manuscript reads as follows:** "Beyond arsenic, elevated activities of naturally occurring radioisotopes (e.g., ²²⁶Ra, ²²⁸Ra) have been identified in some samples of fossil groundwater, including those collected from the Disi Sandstone Aquifer of Jordan⁵⁸, the Saq Aquifer System and the Mega Aquifer System of the Arabian Peninsula^{59,60}, and the Nubian Sandstone Aquifer System of northeastern Africa⁶¹. Careful consideration of treatment options (e.g., blending, reverse osmosis) may be warranted in some of the areas where fossil groundwater is tapped for direct household use."
- **We revised the following statement from our original manuscript** "elevated groundwater salinity levels can render groundwaters inadequate for drinking and irrigation."; **to instead read as follows in our revised manuscript:** "elevated salinity levels can render groundwater inadequate for drinking and irrigation."
- **Our original manuscript stated** "Our finding that fossil groundwater use is widespread and increasing has a number of implications for understanding the vulnerability of groundwater users to high-salinity groundwaters. In coastal settings, there are multiple aquifer systems where fossil groundwaters are known to exist and groundwater wells tap these deep aquifers"; **we revised our manuscript to remove the word 'groundwaters'; the revised manuscript reads as follows:** "Our finding that fossil groundwater use is widespread and increasing has a number of implications for understanding the vulnerability of groundwater users to high-salinity levels. In coastal settings, there are multiple aquifer systems where fossil groundwater has been identified and where wells tap these fossil aquifers."
- **We revised the following statement in our original manuscript** "Farther inland, deeper groundwaters are generally more likely to be fossil in their age," **to instead read as follows** "Farther inland, deeper groundwater is generally more likely to be fossil in its age"
- **Our original manuscript stated** "Although the use of fossil groundwater does not have to mean that groundwater use is non-renewable, fossil groundwaters tend to be more common in deep aquifers (Supplementary Fig. 1). Because deep aquifers are more likely to be confined than shallow aquifers, fossil groundwaters are disproportionately common in confined aquifers. Pumping groundwater from confined aquifers can have different ramifications on metrics of groundwater quantity (e.g., hydraulic heads, vertical hydraulic gradients) than similar amounts of pumping from unconfined aquifers, because confined aquifers have lower storativity than unconfined aquifers. Sustained pumping from confined aquifers—where fossil groundwaters are disproportionately common"; **we revised our manuscript to remove the word 'groundwaters'; the revised manuscript reads as follows:** "Although the use of fossil groundwater does not

have to mean that groundwater use is non-renewable, fossil groundwater does tend to be more common in deeper aquifers (Supplementary Fig. 1). Because deeper aquifers are more likely to be confined than shallower aquifers, fossil groundwater is disproportionately common in confined aquifers. Pumping groundwater from confined aquifers can have different ramifications on metrics of groundwater quantity (e.g., hydraulic heads, vertical hydraulic gradients) than similar amounts of pumping from unconfined aquifers, because confined aquifers typically have a lower storativity than unconfined aquifers. Sustained pumping from confined aquifers—where fossil groundwater is disproportionately common”

- **The final subsection of the main text in our original manuscript stated** “Widespread reliance on fossil groundwater in the US suggests that safeguarding the quality of deep and fresh fossil groundwaters is key to modern water provision”; **we modified our manuscript and replaced the word ‘groundwaters’ such that our revised manuscript reads as:** “Widespread reliance on fossil groundwater (Fig. 2) in the US suggests that safeguarding the quality of deep and fresh fossil groundwater is key to modern water provision.”
- **The Methods section in our original manuscript stated** “fossil groundwaters tend to be more common in deeper wells, especially wells with depths exceeding 200 ± 100 m”; **we revised our manuscript to remove the word ‘groundwaters’; the revised manuscript reads as follows:** “fossil groundwater tends to be more common in deeper wells, especially wells with depths exceeding 200 ± 100 m”
- **In its Methods section, our original manuscript stated** “Fossil groundwaters are common, though not ubiquitous, in wells that are deeper than 200 ± 100 m”; **we revised our manuscript to remove the word ‘groundwaters’; the revised manuscript reads as follows:** “Fossil groundwater is common, though not ubiquitous, in wells that are deeper than 200 ± 100 m”

In summary, all uses of the word “groundwaters” in the main text of our original manuscript have now been revised, such that the term ‘groundwaters’ no longer exists in the main text of our revised manuscript.

Per my comment above, figure 1 and the discussion of USAD seem like a bit of a tangent at the end of the introduction. Perhaps this is a separate results section? Or maybe just part of the methods if you feel these relationships are previously well established and don’t warrant a separate results section.

Thank you for your feedback, we agree. We have deleted almost all of the introduction text describing USAD and moved much of it to the Methods section; we have an entire section in the Methods devoted to describing these new geospatial data (as well as a lengthy supplementary section).

Specifically, here is the text from our original manuscript that we have deleted from the introduction: *“This new United States Aquifer Database (USAD)—available here as a supplementary geodatabase—delineates aquifer systems by digitizing aquifer system boundaries on the basis of an extensive literature review (Supplementary Table 4). Specifically, we georeferenced aquifer system outlines from maps presented within hundreds of locally relevant studies, and also use national-scale groundwater well data as a guide to better identify aquifer system boundaries. This geodatabase includes over 430 aquifers across the US and addresses several shortcomings of an existing nationwide aquifer dataset (Supplementary Note 3.1). Herein we refer to these delineated, two-dimensional areas (i.e., polygons in Fig. 1a) as ‘aquifers’, although we stress that these two-dimensional areas are underlain by multiple geologic formations each defined as separate local aquifers or aquitards that together form ‘aquifer systems’.”*

- Line 195 – can you be more explicit here on the method for identifying wells known to draw some fossil groundwater? Is this not the depth approach you have been using so far?

Thank you for your feedback. Our original manuscript contained the following statement on Line 195 *“Specifically, we identified wells known to draw some fossil groundwater (>0% as calculated by ref. 1)”*. We agree with the reviewer that our manuscript would be improved by more explicit statement on the method for identifying wells known to draw some fossil groundwater. We have thus revised our manuscript by adding a sentence designed to (i) provide greater detail on how fossil well waters were identified in ref. 1, and (ii) direct the reader to the Supplementary Note 1.2 where we have added a new ~1/2 page of text (see below) to better detail the approach published in ref. 1. Our revised sentence reads as follows:

“Specifically, we identified wells known to draw some fossil groundwater (>0% as calculated by ref. 1); fossil well waters can be identified by attributing low ^{14}C activities to radioactive decay, after accounting for other potential carbon sources on the basis of $^{13}\text{C}/^{12}\text{C}$ measurements (for further details see Supplementary Note 1.2 and the methods section within ref. 1)”

We have also added a new paragraph within Supplementary Note 1.2 (above Supplementary Fig. 1) that provides greater detail about the method applied in ref. 1 :

“Supplementary Fig. 1 presents the prevalence of detectable fossil well waters from ref. 1, identifying the frequency with which wells draw >0% fossil water. The proportion of well water samples made up of fossil groundwater was determined on the basis of carbon isotope compositions (for full method, see ref. 1). In brief, the fraction of fossil water in a well water sample (F_{Fossil}) was determined following:

$$F_{\text{Fossil}} = \frac{{}^{14}\text{C}_{\text{Sample}} - {}^{14}\text{C}_{\text{Fossil}}}{{}^{14}\text{C}_{\text{Holocene}} - {}^{14}\text{C}_{\text{Fossil}}}$$

where ${}^{14}\text{C}_{\text{Sample}}$ is the measured dissolved inorganic radiocarbon of a well water sample, ${}^{14}\text{C}_{\text{Fossil}}$ is the calculated range of ${}^{14}\text{C}$ activities for precipitation that infiltrated more than 11,700 years ago (i.e., before the Holocene), and ${}^{14}\text{C}_{\text{Holocene}}$ is the calculated range of ${}^{14}\text{C}$ activities for precipitation that infiltrated within the past 11,700 years. Both ${}^{14}\text{C}_{\text{Fossil}}$ and ${}^{14}\text{C}_{\text{Holocene}}$ have large ranges; we calculated a value for the minimum F_{Fossil} determined from the range of ${}^{14}\text{C}_{\text{Fossil}}$ and ${}^{14}\text{C}_{\text{Holocene}}$ values.

Ref. 1 also considered the potential for dissolution of carbonate bearing zero radiocarbon (e.g., dissolution of Paleozoic limestone) using stable carbon isotope compositions (for details see equations 1-3 in ref. ¹). There are other potential sources of inorganic C that are not explicitly unaccounted for such as endogenic CO₂ (e.g., Noseck, U., Rozanski, K., Dulinski, M., Havlová, V., Sracek, O., Brassler, T., Hercik, M., Buckau, G (2009). Carbon chemistry and groundwater dynamics at natural analogue site Ruprechtov, Czech Republic: Insights from environmental isotopes. *Applied Geochemistry*, 24, 1765–1776); however, even where endogenic carbon contributions to DIC may be high, their presence has been suggested to have little influence on F_{fossil} estimations (Wang, T., Chen, J., Zhang, C. (2021). Estimation of fossil groundwater mass fraction accounting for endogenic carbon input across California. *Journal of Hydrology*, 595, 126034). For further details on the method applied to identify fossil well waters see ref. ¹. The fraction of all wells with ¹⁴C data (reported in ref. ¹) where the minimal estimated F_{Fossil} exceeds zero are displayed in Supplementary Fig. 1 (as a function of total well depth on the y-axis)."

- The methods associated with Figure 3 are not very clear in the results section. They are much clear in the methods I think some revision to the results text would help clarify.

Thank you for your comment. We have substantially revised this portion of the manuscript; namely, we have moved our original manuscript's supplementary analysis into the main text in response to the following comment from Reviewer 1 (bold text quoted from Reviewer 1): **"The additional analysis done by looking at wells surrounding wells with proven fossil water is new and could be presented as a sole result for this objective."** We have revised our Methods to be clearer (i.e., both the methods associated with our new Fig. 3 and also our original manuscript's Fig. 3, which is part of Supplementary Note 2.2 in our revised submission).

- From the description on line 196 it sounds like this analysis is done based on proximity to individual wells but it's not clear what the points on Figure 2 are actually representing. Are these individual wells or points grouped by aquifer? (Here too this is described more clearly in the methods section)

We thank for your comment, we recognize our previous manuscript was insufficiently clear about Figure 2. We have made three modifications to our manuscript:

(i) First, we have added a new sentence to our results section to be clearer about what the points on Figure 2 are actually representing. Specifically, we added the following sentence to our Results (section 2.1):

"Each point in Fig. 2a-c presents the proportion of wells within an aquifer system that are deeper than 100 m (Fig. 2a), 200 m (Fig. 2b), or 300 m (Fig. 2c); that is, each of the points in Fig. 2a-c represents the proportion of all drilled wells that are deeper than 200±100 m within one of the aquifer polygons displayed in Fig. 1a."

(ii) Second, we have added a new sentence to the caption of Fig. 2 to improve the clarity of our approach in our revised manuscript. The new sentence in the Fig. 2 caption reads as follows:

"The red, orange, light yellow, light blue and dark blue diamonds in panels a, b, and c represent the proportion of all wells within a given aquifer system's boundaries that have been drilled deeper than 100 m (panel a), 200 m (panel b) or 300 m (panel c)."

(iii) Third, our original manuscript's line 196 stated *"Next, we identified all drilled wells within 20 km of the well that is known to pump fossil water, and quantified temporal variations in the frequency with which new wells are constructed deeper than the nearby well known to draw some fossil water (for a visual depicting this method see Supplementary Fig. 2)".* This sentence pertains to our Fig. 3, which presents changes over time in the proportion of deep wells. Following the recommendation of Reviewer 1, we have now replaced the main text Fig. 3 in our original manuscript with a figure similar to Supplementary Fig. 3 from our original manuscript. We have added the following new sentences near the original line 196 to provide more clarity on our methods.

“ We show that the proportion of wells tapping fossil aquifers has increased over time in more places than it has decreased over time (Table 1). Specifically, we show that study areas where the proportion of wells tapping fossil aquifers has increased over time are ~1.4-3.4 times more common than study areas where the proportion of wells tapping fossil aquifers has decreased over time (Table 1). We conclude that the proportion of newly drilled wells that are sufficiently deep to tap fossil aquifers has increased over time in more places than it has decreased in the US.”

- Line 235 – This paragraph is confusing and think needs some additional text to discuss aquifer classification. Referring to some aquifers as fossil aquifers is misleading as the meta analysis (as far as I understand it) could only be used to identify aquifers that are fossil aquifers and there is no way to know if the other aquifers are or aren't because there isn't data for them. Is that correct? If so I think the wording here is very important because currently the implication is that the aquifers not identified as fossil aquifers do not have fossil water and I doubt that is true.

Thank you. We agree that more text describing aquifer classification is needed. We have modified our paragraph (on Line 235 of our original manuscript) to include the following statement so that we do not inadvertently imply that aquifer systems where fossil groundwater has not (yet) been identified do not contain fossil water (as we expect they do, but these radiocarbon data are simply not yet available for the right wells). Specifically, we (i) deleted “fossil” from the paragraph that the reviewer specified (i.e., original line 235); the original manuscript read: *“Conversely, we also find examples of fossil aquifers that are tapped by high proportions..”* and the revised manuscript reads as: *“We do find examples of aquifer systems that are tapped by high proportions of wells deeper than 200±100 m and have also experienced groundwater-level declines.”*

We also identified another area in our manuscript where we would like to incorporate the Reviewer's recommendation, even though this part of our original manuscript was not specifically identified by their comment (which points to Line 235 of our original manuscript). We added the following statement to our introduction:

“Fossil groundwater expectedly exists in portions of most, if not all, aquifer systems, even those where we lack adequate radioisotope data to confirm the presence of fossil groundwater.”

- Figure 4 is really focused on the groundwater depletion trends. Given that this section is all about spatial connections between well drilling depths and water levels I would like to see some maps or figures that connect these spatial patterns in drawdown with the well trends. It's covered in the discussion but given the importance of this as one of the results sections I think some quantitative and graphic analysis is warranted here.

We found this recommendation particularly helpful. Thank you. We made two changes to our manuscript:

(1) We added a new statement in our main text describing the results of our many (n=24) calculations of correlations between metrics of groundwater level change over time and the prevalence of deep wells. Our new paragraph in the main text reads as follows:

“We compared spatial patterns of groundwater level variations over time (Fig. 4) and the proportion of wells within aquifers that have depths exceeding 200±100 m (Supplementary Note 5). ... We did not identify a consistently positive (or negative) correlation coefficient describing variations between the prevalence of deep wells and groundwater level changes over time (Supplementary Table 17), reinforcing that the use of fossil groundwater does not have to mean that groundwater use is non-renewable.”

(2) We added a new spatial statistical analysis via a new Supplementary Note. The contents of our new Supplementary note are reproduced here:

Supplementary Note 5 – Statistical relationships between groundwater level change over time and the prevalence of deep wells

We explore relationships between the prevalence of deep wells (>100 m to >300 m) and groundwater level changes over time. Specifically, we test two different types of relationships.

- a) We evaluate correlations between the proportion of recorded wells within an aquifer system's boundaries that exceed 200±100m versus the median groundwater level change over time (median of all monitoring wells' Sen slopes, determined for any aquifers with sufficient data for analyses as described in Supplementary Note 2.3 and Supplementary Fig. 6). Positive correlation coefficients (Spearman ρ) suggest that aquifer systems where a larger proportion of wells tend to be deep also tend to have higher magnitude groundwater level declines (and are less likely to host monitoring wells where the typical time series (i.e., median change over time) tends to show groundwater level shallowing over time)

- b) Second, we evaluate correlations between the proportion of recorded wells within an aquifer system's boundaries that exceed 200±100m versus the proportion of all monitoring wells within the aquifer system (with sufficient data for analysis, see Methods) that have Sen slope values indicative of groundwater level declines over time (i.e., the proportion of monitoring wells within the aquifer system bounds with positive Sen Slopes describing variations of 'depth to groundwater' versus 'measurement date'). Positive correlation coefficients (Spearman ρ) suggest that aquifer systems where a larger proportion of wells tend to be deep also tend to have a greater proportion of monitoring wells that indicate declining groundwater levels over time.

Because we have three thresholds for 'deep wells' (i.e., the (i) proportion of wells deeper than 100 m, (ii) proportion of wells deeper than 200 m, (iii) proportion of wells deeper than 300 m) and two different plots for which we determine rank correlation coefficients (i.e., "a") and "b") above in this Supplementary Note 5), we calculate six Spearman rank correlation coefficients for each studied time interval. Further, because we study five unique time intervals (i.e., (i) 1950-1975, (ii) 1975-2000, (iii) 2000-2015, (iv) 1950-2015, and (v) 1975-2015) we report a total of six correlation coefficients for each time interval and therefore complete a total of 24 correlation coefficient calculations (i.e., six correlation calculations per time interval, multiplied by five time-intervals).

The rank correlation coefficients are presented in the tables below (i.e., within this Supplementary Note 5). Further, we present a figure for one of the time intervals to provide an example of the substantial scatter in these statistical relationships and provide a clearer view of the statistical relationships we summarize in the tables within this Supplementary Note 5.

We stress that although positive correlation coefficients are more common than negative correlation coefficients (Supplementary Table 17), the use of fossil groundwater does not have to mean that groundwater use is non-renewable. Indeed, the prevalence of deep wells may also be indicative of responses of groundwater users; specifically, some groundwater users may construct deeper wells where groundwater levels are declining where hydrogeologic conditions enable such a construction activities.

Supplementary Table 17. Statistical relationships between the proportion of wells within an aquifer system with depths exceeding 100 m, 200 m or 300 m versus two metrics of groundwater level change over time for the time interval 2000-2015: (i) the fraction of all monitoring wells within an aquifer system indicating that groundwater levels have declined over time, and (ii) the median Sen slope determined on the basis of all monitoring wells within the aquifer system with sufficient data for analyses. Spearman rank correlation coefficients (ρ) are displayed. Each row of the table represents a different time interval (e.g., 1950-1975). Each gray-shaded set of columns represent a different well depth threshold (i.e., "100 m" corresponds to the proportion of wells within an aquifer system with depths exceeding 100 m). For an example of scatterplots of data associated with the time interval 2000-2015 (i.e., row 4 in this table) interval see Supplementary Fig. 11).

Timespan	Fraction of wells deeper than 100 m		Fraction of wells deeper than 200 m		Fraction of wells deeper than 300 m	
	Median* water level variation over time (median Sen slope among monitoring wells)	Proportion* of wells exhibiting water level declines (fraction of monitoring wells with Sen slopes of greater than zero)	Median* water level variation over time (median Sen slope among monitoring wells)	Proportion* of wells exhibiting water level declines (fraction of monitoring wells with Sen slopes of greater than zero)	Median* water level variation over time (median Sen slope among monitoring wells)	Proportion* of wells exhibiting water level declines (fraction of monitoring wells with Sen slopes of greater than zero)
1950-1975	$\rho = 0.294$	$\rho = 0.226$	$\rho = 0.279$	$\rho = 0.189$	$\rho = 0.262$	$\rho = 0.154$
1975-2000	$\rho = 0.063$	$\rho = 0.045$	$\rho = -0.008$	$\rho = -0.015$	$\rho = -0.069$	$\rho = -0.077$
2000-2015	$\rho = 0.208$	$\rho = 0.210$	$\rho = 0.185$	$\rho = 0.204$	$\rho = 0.125$	$\rho = 0.130$
1950-2015	$\rho = 0.299$	$\rho = 0.212$	$\rho = 0.251$	$\rho = 0.158$	$\rho = 0.192$	$\rho = 0.082$
1975-2015	$\rho = 0.222$	$\rho = 0.135$	$\rho = 0.176$	$\rho = 0.096$	$\rho = 0.119$	$\rho = 0.026$

Supplementary Fig. 11. Statistical relationships between the proportion of wells within an aquifer system with depths exceeding 200±100 m versus two metrics of groundwater level change over time: (i – left side column of plots with green points) the fraction of all monitoring wells within an aquifer system indicating that groundwater levels have declined over time (i.e., the proportion of monitoring wells within an aquifer system with a Sen slope exceeding zero, where the Sen slope reflects the statistical relationship between depth to groundwater versus measurement date for a given monitoring well); and, (ii – right side plots with blue points) the median Sen slope determined on the basis of all monitoring wells within the aquifer system with sufficient data for analyses (where the Sen slope reflects the statistical relationship between depth to groundwater versus measurement date for a given monitoring well). We only examined aquifer systems with at least 5 monitoring wells meeting our criteria for analyses for a given time interval. The Spearman rank correlation coefficient is displayed on top of each plot; these values are replicated in Supplementary Table 17.”

- Sections 3.2 and 3.3 explores many interesting implications that could build from this work. However, it seems like most of the discussion here is really on implications and not on the results themselves.

We agree. Thank you for your recommendation. We have completed considerable new groundwater quality data analyses to support our discussion of fossil groundwater and well water quality.

Specifically, we downloaded tens-of-thousands of groundwater quality measurements (various analytes to support each bullet point (a), (b) and (c) in discussion section 3.2). We have developed a new Supplementary Note and refer to these analyses focusing on differences in groundwater quality between water drawn from shallower wells versus water drawn from deeper wells in our studied aquifer systems.

Specifically, we made the following updates to (i) the Supplementary Information, and (ii) to the Discussion in the main text:

(i) New text added to the Supplement reads as follows:

“Supplementary Note 6 – Groundwater quality in shallow versus deep wells

Supplementary Note 6.1 – Compilation of US groundwater quality data

Here we describe how groundwater quality data for the contiguous US were downloaded. We data from <https://www.waterqualitydata.us/#siteType=Well&mimeType=csv&sorted=no&providers=NWIS&providers=STEWARDS&providers=STORET>; specifically, we entered the following terms in the search field entitled “Characteristics”: “Arsenic (NWIS, STEWARDS, STORET)”, “Nitrate (NWIS, STORET)”, “Total Dissolved Solids (NWIS, STEWARDS, STORET)” (complete query URLs (entered January 11, 2022) are:

<https://www.waterqualitydata.us/#siteType=Well&characteristicName=Nitrate&mimeType=csv&sorted=no&dataProfile=resultPhysChem&providers=NWIS&providers=STEWARDS&providers=STORET> and

<https://www.waterqualitydata.us/#siteType=Well&characteristicName=Arsenic&mimeType=csv&sorted=no&dataProfile=resultPhysChem&providers=NWIS&providers=STEWARDS&providers=STORET> and

<https://www.waterqualitydata.us/#siteType=Well&characteristicName=Total%20dissolved%20solids&mimeType=csv&sorted=no&dataProfile=resultPhysChem&providers=NWIS&providers=STEWARDS&providers=STORET>

Here we detail steps to quality check each dataset: (i) arsenic, (ii) nitrate, (iii) total dissolved solids.

(i) To quality control the downloaded arsenic dataset, we excluded measurements returning a below detection result if the detection limit equaled or exceed 10 µg/L. We also any measurement that does not include the term “Dissolved” in the field entitled “ResultSampleFractionText (this step excluded measurements of “Total” arsenic, as we concluded that including “Total” arsenic measurements in our dataset would implicitly and inappropriately treat dissolved and total arsenic measurements as equivalent in our analyses). We excluded any measurements with a “ActivityMediaSubdivisionName” value other than “Groundwater”. A small number of measurements (n=7) reported “Present Above Quantification Limit”; these samples were excluded from further analyses, as we did not analyze data for an upper limit value for the analytical quantification limit. We excluded measurements with the flag “Detected Not Quantified” in the field “ResultDetectionConditionText”. Well depth data were compiled from the field entitled “WellDepthMeasure/MeasureValue” first and, if this field was empty, we compiled well depth data from the field entitled “WellHoleDepthMeasure/MeasureValue”. For stations with multiple measurements meeting the aforementioned quality control criteria, we analyzed only the most recent measurement.

(ii) To quality control the downloaded nitrate dataset, we excluded all measurements that does not include the term “Dissolved” in the field entitled “ResultSampleFractionText”. We excluded measurements that do not clearly specify sample media as “Groundwater” in the field “ActivityMediaSubdivisionName”. We excluded records reporting “Detected Not Quantified” or “Systematic Contamination” under the field “ResultDetectionConditionText”. We excluded measurements if the units did not specify ‘as NO₃’ or ‘as N’ (i.e., some records specify only “mg/l” or “ug/l” or “ppm”, and were thus deleted as this information was insufficient to determine if the units are ‘as NO₃’ or ‘as N’). We converted nitrate data into consistent units of “nitrate as N” (as some measurements and stated detection limits are recorded in units of “mg/l as NO₃”). We excluded measurements recording a below detection limit measurement if the stated detection limit equaled or exceeded 10 mg/L NO₃- as N. Well depth data were compiled from the field entitled “WellDepthMeasure/MeasureValue” first and, if this field did not contain a number, we compiled well depth data from the field entitled “WellHoleDepthMeasure/MeasureValue” instead. For stations with multiple measurements meeting the aforementioned quality control criteria, we analyzed only the most recent measurement.

(iii) To quality control total dissolved solids (TDS) data, we excluded all records that does not specify "Dissolved" in the field "ResultSampleFractionText". We excluded measurements that do not state "Groundwater" under the field entitled "ActivityMediaSubdivisionName". We deleted any records with a below detection measurement if the detection limit was stated to be equal to or greater than 1000 mg/L. We excluded records reporting "Detected Not Quantified" or "Systematic Contamination" or "Present above Quantification Limit" under the field "ResultDetectionConditionText". We converted units of "tons/ac ft" to mg/L by multiplying by the 735.47. We excluded a records reporting measurements in units of "tons/day". Well depth data were compiled from the field entitled "WellDepthMeasure/MeasureValue" first and, if this field was empty, we compiled well depth data from the field entitled "WellHoleDepthMeasure/MeasureValue". For stations with multiple measurements meeting the aforementioned quality control criteria, we analyzed only the most recent measurement.

Supplementary Note 6.2 – Groundwater quality in deep and shallow wells in aquifer systems

The following figures provide, for each of our three study analytes (dissolved arsenic, dissolved nitrate, total dissolved solids), information about the spatial distribution of well waters that exceed the following analyte-specific threshold concentrations: Dissolved arsenic (As): 10 µg/L; Dissolved nitrate (NO₃): 10 mg/L (measured as N; i.e., 10 mg/L NO₃-N); and, Total dissolved solids (TDS): 3,000 mg/L.

The following figures' panels provide the following information:

Panel (a) presents a map of well water quality measurements made in wells with total depths shallower than 50 m. Colored points are those that exceed the analyte-specific concentration threshold (for example, a well water sample with more than 10 mg/L NO₃-N).

Panel (b) presents a map of aquifers shaded by the fraction of shallow wells with groundwater quality data (<50 m; see panel (a)) that have concentrations exceeding the analyte-specific threshold (e.g., the proportion of wells within the aquifer system that have > 10 mg/L NO₃-N). We only shade areas for aquifer systems with at least n=20 water quality measurements.

Panel (c) presents a map of well water quality measurements made in wells with total depths deeper than 100 m and shallower than 300 m. Colored points are those that exceed the analyte-specific concentration threshold (e.g., a well water sample with more than 10 mg/L NO₃-N).

Panel (d) presents a map of aquifers shaded by the fraction of deep wells with groundwater quality data (100-300 m; see panel (c)) that have concentrations exceeding the analyte-specific threshold (e.g., the proportion of wells within the aquifer system that have > 10 mg/L NO₃-N). We only shade areas for aquifer systems with at least n=20 water quality measurements.

Panel (e) plots the fraction of wells that have concentrations exceeding the analyte-specific threshold. Downward-pointing dark blue triangles represent wells with depths between 100-300 m (i.e., values used to shade aquifers in panel d), whereas upward-pointing light blue triangles represent wells with depths shallower than 50 m (i.e., values used to shade aquifers in panel b). We only plot aquifer systems with at least n=20 water quality measurements for both shallow (<50 m) and deep (100-300 m) well depth intervals (i.e., aquifers that appear both panel (b) and panel (d)).

Supplementary Fig. 12. Spatial distributions of dissolved arsenic in US well waters. (a) Well water arsenic measurements in wells with total depths shallower than 50 m; purple colored points are those that exceed 10 µg/L. (b) Aquifers shaded by the fraction of shallow wells (<50 m; see panel (a)) that have arsenic concentrations exceeding 10 µg/L. (c) Well water arsenic measurements in wells with total depths deeper than 100 m and shallower than 300 m; purple colored points are those that exceed 10 µg/L. (d) Aquifers shaded by the fraction of deep wells (100–300 m; see panel (c)) that have arsenic concentrations exceeding 10 µg/L. (e) The fraction of wells that have arsenic concentrations exceeding 10 µg/L for individual aquifer systems (see labels at base of figure). Downward-pointing dark blue triangles represent wells with depths between 100–300 m (i.e., values used to shade aquifers in panel (d)), whereas upward-pointing light blue triangles represent wells with depths shallower than 50 m (i.e., values used to shade aquifers in panel (b)). Dashed vertical black lines mark differences between shallow wells (<50 m; upward-pointing light blue triangles) and deeper wells (100–300 m; downward-pointing dark blue triangles) in their respective frequencies with which these two groups of wells (shallow, deep) have high-arsenic concentrations (>10 µg/L).

Supplementary Fig. 13. Spatial distributions of dissolved nitrate (NO_3) in US well waters. (a) Well water NO_3 measurements in wells with total depths shallower than 50 m; green colored points are those that exceed 10 mg/L (NO_3 as N). (b) Aquifers shaded by the fraction of shallow wells (<50 m; see panel (a)) that have nitrate concentrations exceeding 10 mg/L. (c) Well water nitrate measurements in wells with total depths deeper than 100 m and shallower than 300 m; green colored points are those that exceed 10 mg/L. (d) Aquifers shaded by the fraction of deep wells (100-300 m; see panel (c)) that have nitrate concentrations exceeding 10 mg/L. (e) The fraction of wells that have nitrate concentrations exceeding 10 mg/L for individual aquifer systems (see labels at base of figure). Downward-pointing dark blue triangles represent wells with depths between 100-300 m (i.e., values used to shade aquifers in panel (d)), whereas upward-pointing light blue triangles represent wells with depths shallower than 50 m (i.e., values used to shade aquifers in panel (b)). Dashed vertical black lines mark differences between shallow wells (<50 m; upward-pointing light blue triangles) and deeper wells (100-300 m; downward-pointing dark blue triangles) in their respective frequencies with which these two groups of wells (shallow, deep) have high- NO_3 (>10 mg/L).

Supplementary Fig. 14. Spatial distributions of total dissolved solids (TDS) in US well waters. (a) Well water TDS measurements in wells with total depths shallower than 50 m; red colored points are those that exceed 3,000 mg/L. (b) Aquifers shaded by the fraction of shallow wells (<50 m; see panel (a)) that have TDS concentrations exceeding 3,000 mg/L. (c) Well water TDS measurements in wells with total depths deeper than 100 m and shallower than 300 m; red colored points are those that exceed 3,000 mg/L. (d) Aquifers shaded by the fraction of deep wells (100-300 m; see panel (c)) that have TDS concentrations exceeding 3,000 mg/L. (e) The fraction of wells that have TDS concentrations exceeding 3,000 mg/L for individual aquifer systems (see labels at base of figure). Downward-pointing dark blue triangles represent wells with depths between 100-300 m (i.e., values used to shade aquifers in panel (d)), whereas upward-pointing light blue triangles represent wells with depths shallower than 50 m (i.e., values used to shade aquifers in panel (b)). Dashed vertical black lines mark differences between shallow wells (<50 m; upward-pointing triangles) and deeper wells (100-300 m; downward-pointing triangles) in their respective frequencies with which these two groups of wells (shallow, deep) have high-TDS (>3,000 mg/L)."

(ii) We inserted text into the revised Discussion section of our manuscript. Here we now highlight the new findings from our analysis of groundwater quality data. First, we added the following to our Discussion paragraph focused on surface-borne pollutants:

“We find that shallower wells tend to have high nitrate concentrations (>10 mg/L NO₃ as N) more frequently than deeper wells in most of the aquifer systems we studied (Supplementary Fig. 13).”

Next, we revised our discussion section on to geogenic contaminants. Our revised manuscript reads as follows:

“Because groundwater age tends to increase with depth, we compared tens-of-thousands of dissolved arsenic measurements in shallow and deep wells (Supplementary Fig. 12). In some aquifer systems (n=47 aquifers), shallower wells are contaminated by arsenic (>10 µg/L) more frequently than deeper wells. In other aquifer systems (n=54), deep wells are contaminated by arsenic more frequently than shallower wells. The increasing reliance on fossil water demonstrated here may lead increase exposure to arsenic in some places, but reduce exposure in other places.

Our finding that fossil groundwater reliance is increasing as deep wells become more common may imply concomitant changes in exposure to high-arsenic groundwater; however, we stress that arsenic exposure assessments should be considered on an aquifer-by-aquifer basis because of the importance of local hydrochemical conditions and hydrostratigraphy in determining groundwater arsenic concentrations⁵⁵.”

Third, we revised our discussion section devoted to groundwater salinity to include our new analyses. It reads:

“However, we stress that there are also aquifer systems where shallower wells are more likely to pump brackish water than deeper wells; our analysis of hundreds-of-thousands of total dissolved solids measurements identified a dozen such cases, most of which are arid alluvial basins in the western US (Supplementary Fig. 14). The high spatial variability in the statistical relationship between well water salinity and well depth highlights the importance of considering local hydrogeologic settings and historic land uses when examining connections between increased well drilling into fossil aquifers and the potential threat of salinity to groundwater users.”

References:

- Perrone, D. & Jasechko, S. Deeper well drilling an unsustainable stopgap to groundwater depletion. *Nat. Sustain.* **2**, 773–782 (2019).
- Jasechko, S. et al. Global aquifers dominated by fossil groundwaters but wells vulnerable to modern contamination. *Nat. Geosci.* **10**, 425–429 (2017)

Manuscript ~~Revised manuscript~~ **entitled:** Widespread and ~~increasing reliance on~~ increased drilling of wells into fossil groundwater aquifers in the USA

Submitted to: *Nature Communications*

Authors: Merhawi GebreEgziabher GebreMichael^{a*}, Scott Jasechko^a, Debra Perrone^b

Affiliation ~~Affiliations:~~

^a Bren School of Environmental Science and Management, University of California, Santa Barbara, California, 93106, USA

^b Environmental Studies Program, University of California, Santa Barbara, California, 93106, USA

Words in abstract: ~~450~~ 153

Words in main text: ~~4770~~ 5506

Words in methods: ~~4656~~ 1904

Number of figures: 4

Number of tables: 1

Number of references: ~~75~~ 77

*** Corresponding author:**

Merhawi GebreEgziabher GebreMichael
University of California at Santa Barbara
2400 Bren Hall, Santa Barbara, California, 93106
Email: gebremichael@ucsb.edu

Keyword:

groundwater, aquifer, well, fossil groundwater

Formatted: Font: 12 pt, Superscript

Formatted: Font: 12 pt, Superscript

Formatted: Font: 12 pt, Superscript

Abstract.

Most ~~groundwaters have~~ stored groundwater is 'fossil' in its age, having been under the ground for more than ~12 thousand years. Because ~~these 'fossil' groundwaters~~ fossil groundwater often ~~have~~ has distinct concentrations of contaminants relative to younger ~~groundwaters~~ groundwater, mapping where wells tap fossil aquifers is critical for evaluating contaminant exposure; ~~but~~. Nevertheless, the prevalence of wells that tap fossil aquifers is not known. Here we ~~analyze millions of groundwater well completion records across the US~~. We show that wells that are sufficiently deep to tap fossil aquifers are widespread, though they remain outnumbered by shallower wells in most areas. Moreover, the proportion of newly drilled wells ~~that are~~ deep enough to tap fossil aquifers has increased over recent decades, ~~suggesting a growing reliance on~~. However, ~~this widespread and increased drilling of wells into fossil groundwater~~. We also ~~analyze groundwater level time series and find that fossil groundwater use~~ aquifers is not ~~always~~ necessarily associated with groundwater depletion, ~~emphasizing that the presence of fossil groundwater does not necessarily indicate a non-renewable water supply~~. Our results emphasize the importance of safeguarding fossil groundwater quality and quantity to meet present and future water demands.

Formatted: Font color: Black

Formatted: Font: Not Bold, Font color: Text 1

1. Introduction

Fossil groundwaters—defined as waters that have been underground for more than 12,000 years—likely comprise more than half of global groundwater stored within 1000 m of the land surface¹. Our current understanding of fossil groundwater distributions is based primarily on well water radioisotope measurements. Studies reporting radioisotope measurements have identified fossil groundwaters in over one hundred aquifers around the globe, including the Nubian Sandstone Aquifer System (Chad, Egypt, Libya, Sudan²), the Karoo Aquifer (South Africa³), the Paris Basin (France⁴), the Chalk Aquifer (United Kingdom⁵), the Great Artesian Basin (Australia⁶), and the North China Plain (China⁷). Identifying places where fossil groundwater withdrawals are common or increasing over time is important because older and younger well waters often have different susceptibilities to contaminants, including fluoride (e.g., Brazil's Botucatu Aquifer⁸), arsenic (Mexico's Comarca Lagunera Granular Aquifer⁹), salinity (Tunisia's Sfax Basin¹⁰) and nitrate (e.g., California's Central Valley¹¹). Despite the importance of identifying wells that pump fossil water for understanding their susceptibility to different contaminants, little is known about the prevalence of wells that pump fossil groundwaters, because well water radioisotope measurements and continental-scale well construction depth data are not systematically collected, hindering comparison. Further, although multiple studies have commented on the sustainability of fossil groundwater use^{12–17}, the spatial patterns of fossil groundwater use and groundwater depletion remain unclear.

The United States offers an opportunity to evaluate both the spatio-temporal patterns of wells deep enough to tap fossil groundwater and the spatial relationships between fossil groundwater use and groundwater level changes over time. Compared to other countries, the US has a relatively high density of well water radioisotope measurements¹. Further, the quality of well construction depth data in the US is relatively good, although records are disaggregated among many state and local scale datasets¹⁵. Previous calculations of fossil groundwater prevalence in four thousand wells across the US¹ demonstrated that fossil waters are common, though not ubiquitous, in wells with depths that exceed 200±100 m (Supplementary Fig. 1). Dozens of

publications that report on the occurrence of fossil groundwaters in US aquifers support this finding—that fossil groundwaters are common at depth in many aquifer systems—and highlight that fossil groundwaters occur in a wide variety of hydrogeologic settings (Fig. 1; see meta-analysis in Supplementary Table 1). Pairing compiled US well data¹⁵ with radioisotope data provides an opportunity to better understand the spatio-temporal distributions of wells that tap fossil aquifers and groundwater level changes over time.

This study has three objectives. (i) Our first objective is to evaluate the spatial distributions of wells accessing fossil water, using wells deeper than 200–100 m as a proxy for fossil water access (results). Fossil groundwater—defined as groundwater that has been underground for more than 12 thousand years—likely comprises more than half of global groundwater stored within 1000 m of the land surface¹. Our current understanding of fossil groundwater distributions is based primarily on well water radioisotope measurements. Radioisotope measurements have identified fossil groundwater in over one hundred aquifers around the globe, including the Nubian Sandstone Aquifer System (Chad, Egypt, Libya, Sudan²), the Karoo Aquifer (South Africa³), the Paris Basin (France⁴), the Chalk Aquifer (United Kingdom⁵), the Great Artesian Basin (Australia⁶), and the North China Plain (China⁷). Fossil groundwater expectedly exists in portions of most, if not all, aquifer systems, even those where we lack adequate radioisotope data to confirm the presence of fossil groundwater.

Identifying places where fossil groundwater withdrawals are common or increasing over time is important, because older and younger well waters often have different susceptibilities to contaminants, including fluoride (e.g., Brazil's Botucatu Aquifer⁸), arsenic (Mexico's Comarca Lagunera Granular Aquifer⁹), salinity (Tunisia's Sfax Basin¹⁰) and nitrate (e.g., California's Central Valley¹¹). Despite the importance of identifying wells that pump fossil groundwater for understanding contamination risk, little is known about the prevalence of wells that pump fossil groundwater. One reason for this is that well water radioisotope measurements and continental-scale well construction depth data are not systematically collected, hindering comparison.

Understanding where wells access fossil groundwater has implications beyond well water quality assessments. First, fossil groundwater can capture human interest, garnering intrinsic and economic values beyond those ascribed to younger water (Supplementary Note 7). Second, fossil groundwater that discharges at springs or into lowland streams can play a critical role in sustaining vulnerable ecosystems¹². Third, mapping wells that tap fossil aquifers can enable better understanding of the prevalence of communities that rely on fossil groundwater resources¹³. Although multiple studies have commented on the sustainability of fossil groundwater use^{14–18}, the spatiotemporal patterns of fossil groundwater use and groundwater depletion remain unclear, partly because of a lack geospatial data with locally relevant information for aquifers in the US.

The United States has sufficient data to evaluate both the spatiotemporal patterns of wells deep enough to tap fossil groundwater, and the spatial relationships between fossil groundwater use and groundwater level changes over time. Compared to other countries, the US has relatively dense well water radioisotope measurements¹. Similarly, the quality of US well construction depth data is relatively good¹⁹. These two continental-scale datasets have never been merged at continental-scale, but could provide an opportunity to analyze spatiotemporal patterns of wells of sufficient depth to tap fossil groundwater if merged with a new geospatial dataset that has locally relevant information about aquifers in the US (Fig. 1a). The lack of locally relevant aquifer

geospatial data has constrained the number of aquifer systems evaluated in previous studies^{1,19}, highlighting an opportunity to explore fossil water access more systematically now that these geospatial data are available.

Here, we combine (a) radioisotope-based fossil groundwater prevalence data¹, (b) well drilling data¹⁹, (c) groundwater level data, and (d) a novel geodatabase consisting of ~440 aquifer systems, providing locally relevant study areas in the contiguous US. Together, these data are used to meet three objectives.

(i) Our first objective is to evaluate the spatial distributions of wells accessing fossil groundwater, using wells deeper than 200±100 m as a proxy for fossil water access (Results section entitled “Fossil groundwater accessed across US aquifers”). We use deep wells as a proxy for fossil water access, because densely distributed well depth data are available for the great majority of our study aquifers, and wells that have been drilled deeper than 200±100 m frequently tend to draw some fossil groundwater when pumped (Supplementary Fig. 1; Fig. 1). Furthermore, continent-wide and densely distributed groundwater ¹⁴C measurements are not available. As a result, we cannot evaluate locally relevant depths below which fossil water dominates storage in each aquifer system (see Methods section entitled “Limitations to our results due to the lack of adequate groundwater age data”).

(ii) Our second objective is to test for an increase or decrease in whether the frequency with which wells access fossil groundwater in aquifer systems across the United States by exploring how the prevalence of wells that have been drilled deeper than 200±100 m has varied has increased or decreased over time (results Results section entitled “Fossil groundwater accessed more frequently over time”).

(iii) Our third objective is to test if groundwater level declines are disproportionately common in aquifer systems that are tapped by where wells that are sufficiently deep to access fossil groundwater (results Results section entitled “Fossil groundwater use hotspots do not always co-occur with groundwater depletion hotspots”).

We meet our objectives by combining four databases: (a) fossil groundwater prevalence determined on the basis of our own analysis of groundwater radiocarbon data¹ and a meta-analysis of fossil groundwater occurrence (Fig. 1; Methods section entitled “Meta-analysis of fossil groundwater prevalence (Fig. 1)”), (b) records of groundwater well depths, construction dates and purposes for millions of wells at continental scale (refs.^{15,18}; Methods sections entitled “Fossil groundwater accessed across US aquifers (Fig. 2)” and “Fossil groundwater accessed more frequently over time (Fig. 3)”), (c) long-term groundwater level time series recorded by the United States Geological Survey and California’s Groundwater Ambient Monitoring and Assessment Program (Methods section entitled “Groundwater depletion in places where deep wells tap fossil aquifers (Fig. 4)”), and (d) a new geodatabase of hydrogeologic study areas that we term the ‘United States Aquifer Database’ (USAD; see Fig. 1a for aquifer system outlines, and Methods section entitled “Delineating aquifers across the US” for steps applied to delineate each aquifer system).

This new United States Aquifer Database (USAD)—available here as a supplementary geodatabase—delineates aquifer systems by digitizing aquifer system boundaries on the basis of

an extensive literature review (Supplementary Table 4). Specifically, we georeferenced aquifer system outlines from maps presented within hundreds of locally relevant studies, and also use national scale groundwater well data as a guide to better identify aquifer system boundaries. This geodatabase includes over 430 aquifers across the US and addresses several shortcomings of an existing nation-wide aquifer dataset (Supplementary Note 3.1). Herein we refer to these delineated, two-dimensional areas (i.e., polygons in Fig. 1a) as ‘aquifers’, although we stress that these two-dimensional areas are underlain by multiple geologic formations each defined as separate local aquifers or aquitards that together form ‘aquifer systems’.

Fig. 1. Documented occurrence of fossil groundwater in the US and boundaries of our newly created United States Aquifer Database (USAD; see Supplementary Table 4). (a) Boundaries of our 437 locally relevant study areas delineated after reviewing hundreds of primary literature sources describing aquifer boundaries (see Supplementary Table 4 for references). Yellow polygons represent aquifer systems where fossil water has been identified (Supplementary Table 1); pink polygons represent other aquifer systems that have also been included in our analyses. (b) The Milk River Aquifer System is dominated by clastic sedimentary aquitards and aquifers, with fossil groundwater reported in some wells with depths exceeding ~150 m¹⁹. (c) The Denver Basin is a multi-layered clastic sedimentary aquifer system, with fossil groundwater reported in some wells with depths exceeding ~150 m²⁰. (d) The Dakota Aquifer System is comprised of carbonate and clastic sedimentary rocks overlying endogenous bedrock, with fossil groundwater reported in some wells in southeastern South Dakota at depths

exceeding $\sim 60\text{ m}^{24}$ and also in a parallel flow system in Nebraska in some wells with depths exceeding $\sim 170\text{ m}^{22}$. (e) The central portion of the High Plains Aquifer System consists of unconsolidated deposits overlying sedimentary rocks (mostly clastic rocks; e.g., sandstones and mudstones of the Dakota Formation), with fossil groundwater reported in some wells with depths exceeding $\sim 150\text{ m}^{23}$. (f) The North Atlantic Coastal Plain is a multi-layered sedimentary aquifer system underlain by endogenous bedrock, with fossil groundwater reported in some wells with depths exceeding $\sim 80\text{ m}^{24}$. (g) The Floridan Aquifer System consists of a surficial aquifer that is underlain by sedimentary rocks including widespread carbonate aquifers interbedded with confining layers, with fossil water reported in some wells with depths exceeding $\sim 180\text{ m}^{35}$. (h) The Black Warrior River Aquifer System is dominated by clastic consolidated or semi-consolidated aquifers and Paleozoic bedrock, with fossil groundwater reported in some wells with depths exceeding $\sim 150\text{ m}^{26}$. (i) The central portion of the Mississippi Embayment Aquifer System consists of unconsolidated alluvium overlying consolidated clastic sedimentary rocks, with fossil groundwater reported in some wells with depths exceeding $\sim 100\text{ m}^{1}$. (j) The western portion of the Carrizo-Wilcox Aquifer System is a multi-layered sedimentary aquifer system, with fossil water reported at depths exceeding $\sim 400\text{ m}^{27}$. (k) The Mojave Basin consists of alluvium overlying endogenous rock, with fossil groundwater reported in some wells with depths exceeding $\sim 200\text{ m}^{28}$. (l) The Cuyama Valley is comprised of alluvial materials overlying (semi)consolidated clastic bedrock, with fossil water reported in some wells with depths exceeding $\sim 200\text{ m}^{29}$. (m) The northern portion of California's Central Valley Aquifer System is comprised is comprised of alluvial materials overlying (semi)consolidated clastic bedrock, with fossil water reported in some wells with depths of 115 to 300 m^{30} . Each of the twelve cross-sections (panels b-m) are based on descriptions and figures presented by refs. ^{19,29,31-40}. See Supplementary Tables S5-S16 for detailed descriptions of hydrostratigraphy; see Supplementary Fig. 9 for an expanded version of this figure.

Fig. 1. Documented occurrence of fossil groundwater in the US and boundaries of our newly created United States Aquifer Database. (a) Boundaries of our 440 locally relevant study areas delineated after reviewing hundreds of primary literature sources describing aquifer boundaries (see Supplementary Table 4 for references). Yellow polygons represent aquifer systems where fossil water has been identified by ref. ¹ or in our meta-analysis. For a comparison of the spatial distribution of fossil aquifers identified by ref. ¹ and those identified in our meta-analysis see Supplementary Figs. 9 and 10. Pink polygons represent other aquifer systems that have also been included in our analyses. Gold circles represent wells where fossil groundwater has been identified in ref. ¹. (b) The Milk River Aquifer System is dominated by clastic sedimentary aquitards and aquifers, with fossil groundwater reported in some wells with depths exceeding ~150 m²⁰. (c) The Denver Basin is a multi-layered clastic sedimentary aquifer system, with fossil groundwater reported in some wells with depths exceeding ~150 m²¹. (d) The Dakota Aquifer System is comprised of carbonate and clastic sedimentary rocks overlying endogenous bedrock, with fossil groundwater reported in some wells in southeastern South Dakota at depths exceeding ~60 m²² and also in a parallel flow system in Nebraska in some wells with depths exceeding ~170 m²³. (e) The central portion of the High Plains Aquifer System consists of unconsolidated deposits overlying sedimentary rocks (mostly clastic rocks; e.g., sandstones and mudstones of the Dakota Formation), with fossil groundwater reported in some wells with depths exceeding ~150 m²⁴. (f) The

North Atlantic Coastal Plain is a multi-layered sedimentary aquifer system underlain by endogenous bedrock, with fossil groundwater reported in some wells with depths exceeding ~80 m²⁵. (g) The Floridan Aquifer System consists of a surficial aquifer that is underlain by sedimentary rocks including widespread carbonate aquifers interbedded with confining layers, with fossil water reported in some wells with depths exceeding ~180 m²⁶. (h) The Black Warrior River Aquifer System is dominated by clastic consolidated or semi-consolidated aquifers and Paleozoic bedrock, with fossil groundwater reported in some wells with depths exceeding ~150 m²⁷. (i) The central portion of the Mississippi Embayment Aquifer System consists of unconsolidated alluvium overlying consolidated clastic sedimentary rocks, with fossil groundwater reported in some wells with depths exceeding ~100 m¹. (j) The western portion of the Carrizo-Wilcox Aquifer System is a multi-layered sedimentary aquifer system, with fossil water reported at depths exceeding ~400 m²⁸. (k) The Mojave Basin consists of alluvium overlying endogenous rock, with fossil groundwater reported in some wells with depths exceeding ~200 m²⁹. (l) The Cuyama Valley is comprised of alluvial materials overlying (semi)consolidated clastic bedrock, with fossil water reported in some wells with depths exceeding ~200 m³⁰. (m) The northern portion of California's Central Valley Aquifer System is comprised of alluvial materials overlying (semi)consolidated clastic bedrock, with fossil water reported in some wells with depths of 115 to 300 m³¹. Each of the twelve cross sections (panels b-m) are based on descriptions and figures presented by refs. ^{20,30,32-41}. See Supplementary Tables 5-16 for detailed descriptions of hydrostratigraphy; see Supplementary Figs. 9-10 for alternate and enlarged versions of this figure.

2. Results

2.1. Fossil groundwater accessed across the US

To better understand the spatial distribution of wells that potentially ~~access~~ pump fossil groundwater, we analyzed ~~more than ~~~5.3 million groundwater wells in ~~437-440~~ aquifer systems across the contiguous United States. We interpret the prevalence of deep wells—defined as wells deeper than 200±100 m—as a proxy for the prevalence with which wells tap fossil groundwater (Fig. 2). ~~We show that wells drilled deeper than 100 m, 200 m, and 300 m—where fossil groundwaters are common (ref. ¹; Supplementary Fig. 1)—are widespread in many US aquifers (Fig. 2a-c).~~ Previous calculations of fossil groundwater prevalence in four thousand wells across the US¹ demonstrated that fossil groundwater is common in wells with depths that exceed 200±100 m (Supplementary Fig. 1). Our meta-analysis of studies that report on the occurrence of fossil groundwater in US aquifers supports this finding (Fig. 1 panels b-m), and highlights that fossil groundwater occurs in a wide variety of hydrogeologic settings (Fig. 1; meta-analysis in Supplementary Table 1; Supplementary Figs. 9 and 10 for spatial relationship of fossil well water presented in ref. ¹ and the meta-analysis presented here).

We analyzed spatial patterns of wells that are deeper than 200±100 m, and, therefore, likely to pump fossil groundwater. We show that wells drilled deeper than 100 m, 200 m, and 300 m are widespread in many US aquifers (Fig. 2a-c). Each point in Fig. 2a-c presents the proportion of wells within an aquifer system that are deeper than 100 m (Fig. 2a), 200 m (Fig. 2b), or 300 m (Fig. 2c); that is, each of the points in Fig. 2a-c represents the proportion of all drilled wells that are deeper than 200±100 m within one of the aquifer polygons displayed in Fig. 1a. Specifically, more than one-in-ten wells have depths that exceed 100 m in 67% of our study aquifers, exceed 200 m in 17% of aquifers, and exceed 300 m in 4.6% of aquifers (Fig. 2). ~~Further, more~~ More than one-in-five wells have depths that exceed 100 m in 49% of our study aquifers, exceed 200 m in 8.0% of study aquifers, and exceed 300 m in 2.1% of our study aquifers (Fig. ~~2)-2~~).

Nevertheless, deep wells that tap fossil aquifers are far outnumbered by shallower wells with depths less than 200±100 m in most aquifers.

Formatted: Space Before: 12 pt

Aquifer systems that have high proportions of wells with depths exceeding 200±100 m include layered sedimentary aquifer systems in the northern Great Plains (e.g., eastern portion of South Dakota, where wells tap the Dakota Aquifer), southern Texas (e.g., Carrizo-Wilcox Aquifer System), central Texas (e.g., the Stockton Plateau and the Balcones Fault Zone, each part of the broader Edwards-Trinity Aquifer System), and alluvial basins in Arizona (e.g., Picacho Basin, Maricopa-Stanfield Basin; Harquahalla Basin, and Little Chino Valley), Nevada (e.g., Boulder Valley, Coyote Springs Valley) and California (e.g., Cuyama Valley, Los Angeles Basin, and Santa Clara-Calleguas Basin; Fig. 2d-f). By contrast, areas where vast majority of wells are shallower than 200±100 m include the Puget Sound Lowlands (Washington), Great Bend Prairie and Equus Beds of the east-central High Plains Aquifer System (Kansas), Central Lowland Till Plain aquifers (Illinois, Indiana, Ohio, Kentucky), the Michigan Basin (Michigan), the Cape Cod Aquifer System (Massachusetts), and the Biscayne Aquifer System (Florida; Fig. 2).

Fig. 2. The fraction of wells that are deeper than 100 m, 200 m or 300 m in US aquifer systems. Panels (a-c) present the fraction of wells deeper than 100 m, 200 m or 300 m for each aquifer (i.e., each diamond represents one the analysis of all wells within a particular aquifer). The data are ranked from highest (i.e., largest proportion of wells within an aquifer that are deeper than 100 m, 200 m or 300 m) to lowest y-axis values. Panels (d-e) display spatial patterns of the fraction of wells deeper than 100 m, 200 m or 300 m. Each polygon represents one aquifer system. Red and orange shades mark aquifers with higher proportions of wells constructed to deep depths. The red, orange, light yellow, light blue and dark blue diamonds in panels a, b, and c represent the proportion of all wells within a given aquifer system's boundaries that have been drilled deeper than 100 m (panel a), 200 m (panel b) or 300 m (panel c).

2.2. Fossil groundwater accessed more frequently over time

We tested for changes in the prevalence with which wells tap fossil aquifers over time. We analyzed temporal changes in the fraction of newly constructed wells drilled deeper than a nearby well that is known to draw fossil water. Specifically, we identified wells known to draw some fossil groundwater (>0% as calculated by ref. To explore fossil groundwater access over time, we analyzed changes in the fraction of newly constructed wells with depths exceeding 200±100 m over five different time intervals: 1950-1975, 1975-2000, 2000-2015, 1975-2015, and 1950-2015. Among study aquifers with sufficient data (see Methods), we find an increase over time in the fraction of wells drilled deeper than 200±100 m in 1.2-1.4 times more aquifers than we find a decrease over time in the fraction of wells drilled deeper than 200±100 m (Fig. 3; Supplementary Table 3). Specifically, we calculated correlation coefficients of the rank transforms of “groundwater well construction year” versus “the proportion of all wells drilled within a given year that are deeper than 100 m, 200 m, or 300 m”¹); fossil well waters can be identified by attributing low ¹⁴C activities to radioactive decay, after accounting for other potential carbon sources on the basis of ¹³C/¹²C measurements (for further details see Supplementary Note 1.2 and the methods section within ref. ¹). Next, we define ‘study areas’ as areas within a 20 km radius of a well that is known to draw fossil water (see Supplementary Fig. 2). We identified newly drilled wells that are located within each study area. We then calculated the proportion of newly drilled wells that have depths that are deeper than the nearby well that draws fossil water. Last, we quantified temporal variations in the frequency with which new wells are constructed deeper than the nearby well known to draw some fossil water (see Supplementary Fig. 2 for schematic of statistical analysis). We analyzed five time intervals: 1950-1975, 1975-2000, 2000-2015, 1950-2015 and 1975-2015.

We show that the proportion of wells tapping fossil aquifers has increased over time in more places than it has decreased over time (Table 1). Specifically, we show that study areas where the proportion of wells tapping fossil aquifers has increased over time are ~1.4-3.4 times more common than study areas where the proportion of wells tapping fossil aquifers has decreased over time (Table 1). We conclude that the proportion of newly drilled wells that are sufficiently deep to tap fossil aquifers has increased over time in more places than it has decreased in the US.

On an aquifer-by-aquifer basis, we identified 36 aquifer systems that contain at least five wells known to draw fossil water with sufficient nearby (<20 km) well drilling data for analyses. We find evidence for an increase over time in the proportion of wells tapping fossil aquifers in the Black Warrior River Aquifer System, the Central High Plains and California’s Santa Rosa Valley (all of which have median rank correlation coefficients exceeding zero for all studied time intervals; Fig. 3).

In addition to the above analysis, we completed a complementary analysis for each of our aquifer systems by calculating temporal variations in the proportion of newly drilled wells that exceed 200±100 m (Supplementary Note 2.2). Specifically, we calculated correlation coefficients of the rank transforms of “groundwater well construction year” versus “the proportion of all wells drilled within a given year that are deeper than 200±100 m”. These Spearman rank correlation coefficients (i.e., ρ values) demonstrate that the fraction of new wells that are drilled deeper than 200±100 m has increased over time in more than half of all aquifers with sufficient data for analyses (Supplementary Fig. 4). If we limit our analyses to consider only significant correlations (Spearman ρ values of less than 0.05), our conclusion becomes even stronger (see

diamond symbols in Fig. 3). Our analysis suggests that the proportion of newly drilled wells that are sufficiently deep to tap fossil aquifers has increased over time in many areas across the US (Figs. 3 and 4).

To further test our finding—that the proportion of wells tapping fossil aquifers has increased over time in more places than it has decreased over time—we completed a complementary analysis that focuses on well drilling depth time series in areas that surround a well that is known to draw fossil groundwater (see Supplementary Note 2.1). Among aquifer systems with sufficient data (see Methods), 1.2-14 times more aquifers show an increase over time in the fraction of wells drilled deeper than 200 ± 100 m than those that show a decrease (Supplementary Fig. 4; Supplementary Table 2). If we limit our analyses to consider only significant correlations (Spearman P-values of less than 0.05), our conclusion becomes stronger (see large diamond symbols denoting significant (P -value < 0.05) correlations in Supplementary Fig. 3). Specifically, we identified wells known to draw some fossil groundwater ($> 0\%$ as calculated by ref.). This complementary analysis supports our finding: there is an increase in the proportion of wells being constructed to deep depths where fossil groundwater is common in the majority of aquifers that we studied (see Supplementary Fig. 4). We conclude that the proportion of wells tapping fossil groundwater resources is likely increasing across the US.

Table 1. Temporal variations in the proportion newly drilled wells that are deeper than a nearby (within 20 km) well that pumps fossil water (Supplementary Fig. 2 for schematic of analysis)

Time interval	Total number of areas* analyzed	Number of areas* where the rank correlation coefficient is greater than zero (i.e., the proportion of wells being drilled deeper than a nearby well that is known to contain fossil water is, if anything, increasing over time)	Number of areas* where the rank correlation coefficient is less than or equal to zero (i.e., the proportion of wells being drilled deeper than a nearby well that is known to contain fossil water is, if anything, decreasing over time)	Number of areas where the rank correlation coefficient is undefined	(the number of areas with rank correlation coefficient of greater than zero) divided by (the number of areas with rank correlation coefficient of less than zero)
1950-1975	436	230	144	62	1.6
1975-2000	703	400	190	113	2.1
2000-2015	855	433	314	108	1.4
1950-2015	427	303	100	24	3.0
1975-2015	643	451	131	61	3.4

* defined as land area within a 20 km radius of a well that has been identified¹ to pump fossil water

Fig. 4). Next, we identified all drilled wells within 20 km of the well that is known to pump fossil water, and quantified temporal variations in the frequency with which new wells are constructed deeper than the nearby well known to draw some fossil water (for a visual depicting this method see Supplementary Fig. 2). This complementary analysis supports our finding: there is an increase in the proportion of wells being constructed to deep depths where fossil groundwaters are common in the majority of aquifers we studied (see Supplementary Fig. 3).

Formatted: Font: Calibri, 11 pt, Bold

Fig. 3. Variability in the proportion of wells deep enough to tap fossil water over time; points plotted in the red background indicate that the proportion of newly drilled wells that are deep enough to tap fossil aquifers has increased over recent decades in more places than it has decreased. The y-axis presents [the number of aquifers where the proportion of newly drilled wells that are deeper than 200±100 m is increasing over time (i.e., number of aquifers with $p > 0$)] divided by [the number of aquifers where the proportion of newly drilled wells that are deeper than 200±100 m is decreasing over time (i.e., number of aquifers with $p < 0$)]. The rank correlation coefficients for each aquifer were determined on the basis of statistical relationships between [the proportion of newly drilled wells in a given year that are deeper than a given threshold depth (Spearman rank correlation coefficients ($p > 0$))] and [well completion year]. Each point represents a unique combination of a studied time interval over which our correlation coefficients were determined and a given threshold depth (i.e., threshold depths are: 100 m, 200 m, and 300 m). Green points reflect temporal variability in the proportion of wells deeper than 100 m, yellow points reflect temporal variability in the proportion of wells deeper than 200 m, and pink points reflect temporal variability in the proportion of wells deeper than 300 m. Circles depict all results (i.e., any Spearman p value); diamonds depict only significant correlations (i.e., only those correlations with a Spearman p value of less than 0.05).

3. Temporal variations in well drilling depths in areas located nearby a well that has been reported to pump fossil water. The Spearman rank correlation coefficients (ρ) were determined by correlating calendar year versus the fraction of newly constructed wells with depths exceeding the depth of a nearby (<20 km away) groundwater well that is known to pump fossil water. Each panel (a-e) represents correlations completed over one of five different time intervals: 1950-1975, 1975-2000, 2000-2015, 1950-2015 and 1975-2015. Each bar represents the statistical distribution of all study areas (i.e., 20 km buffers around a groundwater well that has been reported to pump fossil water) correlation coefficients determined within a given aquifer system's boundaries; the thick horizontal black line represents the median ρ value for all areas with sufficient data within the aquifer system, the top and bottom of the shaded box represents the 25th-75th percentile range of ρ values, the dashed line and cap extends to the 10th-90th percentile range, and circles represent outlier points. Aquifer systems marked with orange-shaded boxplots have median ρ values exceeding zero (indicative of an increasing proportion of newly drilled wells that are deeper than the well that has been documented to pump fossil water); aquifer systems marked with blue-shaded boxplots have median ρ values of equal to or less than zero (indicative of an unchanging or increasing proportion of newly drilled wells that are deeper than the well that has been documented to pump fossil water). The "n=" text overlying each box plot represents

number of study areas within the aquifer with sufficient data for analyses for a given time interval. We only present box plots for aquifer systems with at least five study areas with sufficient data to determine a rank correlation coefficient. The labels on the x-axis display the title of each aquifer system and the two-letter code for the state that the centroid of the aquifer system lies within (e.g., two-letter code "CA" denotes that the centroid of the aquifer system lies within California).

2.3. Fossil groundwater use hotspots do not always co-occur with groundwater depletion hotspots

We compared spatial patterns of groundwater level variations over time (Fig. 4) and the proportion of wells within aquifers that have depths exceeding 200±100 m (Supplementary Note 5). First, we analyzed long-term groundwater-level trends in our study aquifers to test for spatial relationships between the prevalence of deep wells (indicative of wells that access fossil water) and declining groundwater levels. We calculated Sen slopes ~~describing to describe~~ the rate of change in groundwater levels over time for each monitoring well ~~with that has~~ sufficient data for analyses (Methods). Next, we present the median Sen slope for each aquifer system (depicted as shaded areas in Fig. 4; median calculated on the basis of all Sen slopes ~~(i.e., one slope for each monitoring well)~~ within a ~~given delineated~~ aquifer; for schematic of method see Supplementary Fig. S5). Finally, we evaluated correlations between the proportion of recorded wells within an aquifer system's boundaries that exceed 200±100 m versus two different metrics of groundwater level change over time ((i) median of all monitoring wells' Sen slopes, determined for any aquifers with sufficient data for analyses, and (ii) the proportion of all monitoring wells within the aquifer system that have Sen slope values indicative of groundwater level declines over time).

We find examples of aquifers We did not identify a consistently positive (or negative) correlation coefficient describing variations between the prevalence of deep wells and groundwater level changes over time (Supplementary Table 17), reinforcing that the use of fossil groundwater does not have to mean that groundwater use is non-renewable.

We do find examples of aquifer systems that are tapped by high proportions of wells deeper than 200±100 m and have also experienced groundwater-level declines. For example, in California's Cuyama Valley (Fig. 1L), we find that 16-63% of wells are deeper than 200±100 m, that low-¹⁴C ~~groundwaters groundwater samples~~ have been ~~identified in collected from~~ wells deeper than 200±100 m^{29,30} (indicative of fossil well water), and that groundwater reserves are being depleted (i.e., the median groundwater level is deepening at a rate of 1-2 m/decade across all three time intervals presented in Fig. 4). Here, fossil groundwater is accessed and groundwater stores are being depleted.

Conversely, we ~~also~~ find examples of ~~fossil~~ aquifers that are tapped by high proportions of wells deeper than 200±100 m that have not experienced substantial and pervasive groundwater-level declines over recent decades. For example, the Black Warrior River Aquifer System (eastern Mississippi through Alabama; Fig. 1h) has a similar percentage of wells that are deeper than 200±100 m as the Cuyama Valley (5-44%~~%)~~, and also contains ~~groundwater with~~ low-¹⁴C ~~groundwaters activities that are~~ indicative of fossil groundwater^{26,27}. But, in contrast to the Cuyama Valley, the median groundwater-level trend in the Black Warrior River Aquifer System has remained near-zero among the three time intervals we studied (Fig. 4a-c: (a) 0.0 m/decade

(1950-1975), (b) +0.2 m/decade (1975-2000), and (c) -0.3 m/decade (2000-2015)). Here, fossil groundwater is likely tapped, but hydraulic heads have remained relatively stable over each of the three time intervals we studied.

Fig. 4. The median groundwater-level change rate (a-c) and the fraction of monitoring well exhibiting declining groundwater levels (d-e) across US aquifer systems included in our US Aquifer Database (USAD; see Methods section entitled “Delineating aquifers across the US”). The upper row of figures displays the median Sen slope of the groundwater-level trend (expressed in meters per decade) analyzing all water level measurements within the years (a) 1950-1975, (b) 1975-2000, and (c) 2000-2015 (the median trend was calculated by determining the Sen slope for every monitoring well within the aquifer boundaries, and then calculating the median among these Sen slope values; see Supplementary Fig. 5 for methods schematic of method). The lower row of figures displays the proportion of all monitoring wells that have a Sen slope exceeding zero (panels d-f), thus presenting the fraction of monitoring wells within the aquifer boundaries that exhibit declining groundwater levels for the years (d) 1950-1975, (e) 1975-2000, and (f) 2000-2015. We only present aquifers for which we analyzed groundwater-level time series for at least five unique monitoring wells (over the specified time interval).

3. Discussion

3.1. Wells tap Water quality ramifications of fossil aquifers across groundwater use in the US

In most aquifer systems, fossil groundwaters tend to be more common at deeper depths (We find Fig. 1; Supplementary Fig. 1). Our meta analysis confirms that fossil groundwaters are common at depths exceeding 200-100 m in many aquifers across the contiguous United States—including aquifers in the semi arid western US (e.g., California Central Valley), the humid south central US (e.g., Mississippi Embayment Aquifer System) and the subtropical southeastern US (e.g., Floridan Aquifer System; Fig. 1).

Formatted: Font: Calibri, 11 pt, Bold

We analyzed spatial patterns of wells that are deeper than 200±100 m, and, therefore, likely to pump fossil groundwater. We show that more than 1 in 10 wells have depths exceeding 200±100 m deep in many US aquifers (e.g., more than 1 in 10 wells exceed 100 m in 67% of study aquifers, and more than 1 in 10 wells exceed 300 m in 4.6% of study aquifers; Fig. 1), suggesting that fossil groundwater is currently accessed by wells for domestic, agricultural, and industrial purposes in many aquifers across the US. Nevertheless, deep wells that tap fossil aquifers are far outnumbered by shallower wells with depths less than 200±100 m in most aquifers.

Formatted: Font color: Auto

We also analyzed temporal variations in the prevalence of deep wells in US aquifers over five time intervals, and show that the proportion of newly drilled wells that are deeper than 200±100 m is increasing over time in more places than it is decreasing (Fig. 3). These results are consistent with, but distinct from, results showing that newer wells are being drilled deeper than older wells across most of the US¹⁵. We conclude that the proportion of wells tapping fossil groundwater resources is likely increasing across the US.

3.2. Water quality ramifications of fossil groundwater use in the US

Our findings suggest that wells that likely tap fossil aquifers are already widespread and are widespread and becoming more common over time have implications for groundwater quality. Specifically, these (Figs. 2 and 3). These results have implications for understanding and quantifying key processes influencing well water vulnerability to (a) surface-borne pollutants, (b) geogenic contaminants, and (c) groundwater salinity.

(a) Surface-borne pollutants contaminate groundwater resources when they percolate downward through the soil profile and unsaturated zone to enter the groundwater from above. Fossil groundwaters tend to have lower concentrations of surface-borne pollutants than younger groundwaters. For example, nitrate—a common surface-borne pollutant that is frequently associated with confined animal feeding operations, excessive fertilization and inadequate sanitation^{41,42}—is more common in recently recharged (i.e., ‘younger’) groundwaters than in older groundwaters¹². Our finding that fossil groundwater access is increasing over time may suggest reduced exposure to surface-borne contaminants in some of these areas; nevertheless, we stress that deep wells can be vulnerable to contaminants because deep wells often contain mixtures of fossil groundwater and recent recharge⁴³. Fossil groundwater tends to have lower concentrations of surface-borne pollutants than younger groundwater. For example, nitrate—a common surface-borne pollutant that is frequently associated with confined animal feeding operations, excessive fertilization and inadequate sanitation^{42,43}—is more common in recently recharged (i.e., ‘younger’) groundwater than in older groundwater¹⁴. Because groundwater age tends to increase with depth, we compared tens-of-thousands of dissolved nitrate measurements in shallow (<50 m) and deep (100-300 m) wells to understand potential water quality ramifications of fossil groundwater use (Supplementary Note 6). We find that shallower wells tend to have high nitrate concentrations (>10 mg/L NO₃ as N) more frequently than deeper wells in most of the aquifer systems we studied (Supplementary Fig. 13). The processes that lead to mixtures of fossil groundwater and recent recharge remain poorly understood, but may include (i) mixing along converging flow paths (e.g., Arava Valley in Israel⁴⁴); (ii) cross-formational mixing (upconing and downwelling) induced as a result of borehole drilling and subsequent pumping (e.g., Diass Aquifer System in Senegal⁴⁵ and the North China Plain⁴⁶); (iii) fast

downward flow of recent precipitation along defective well casings to deeper depths where the well is perforated (e.g., Bohemian Cretaceous Basin in the Czech Republic⁴⁷, and the Aleppo and Steppe Basins in Syria⁴⁸); (iv) pumping from wells with long perforated intervals that simultaneously draw groundwater from both shallow and deep depths (e.g., Malm Limestone Aquifer System in Poland⁴⁹); (v) leakage through gaps in impermeable layers ('windows' in aquitards) separating fossil and modern groundwater; or (vi) relatively rapid vertical groundwater flow and mixing along geologic faults that may serve as conduits that connect fossil and modern groundwater. Because shallow contaminated groundwater can be drawn downward by pumping from deeper wells⁵⁰, developing data products that quantify not only total groundwater withdrawals in a region but also quantify the depths of the wells from which groundwater is withdrawn will be key to understanding fossil aquifer contamination risk.

Formatted: Font color: Text 1

Thus, our finding that fossil groundwater access is increasing over time may suggest reduced exposure to surface-borne contaminants in some of these areas; nevertheless, we stress that deep wells can be vulnerable to contaminants because deep wells often contain mixtures of fossil groundwater and recent recharge^{1,44}. The processes that lead to mixtures of fossil groundwater and recent recharge remain poorly understood, but may include (i) mixing along converging flow paths (e.g., Arava Valley in Israel)(b) Geogenic groundwater contamination can arise due to the interactions of groundwaters with the mineral skeleton of the geologic formations that they flow through and reside within. For example, arsenic is a common geogenic pollutant that poses a challenge to the provision of fresh drinking water in multiple aquifer systems across the world⁵¹. Aqueous arsenic concentrations can differ between fossil versus younger groundwaters, though the statistical relationships between groundwater age and arsenic concentrations vary among aquifer systems and are critically dependent on hydrostratigraphy, redox conditions and flow path architectures^{52,53}. For example, in California's Cuyama Valley, fossil groundwaters pumped from deep wells tend to have higher arsenic concentrations than younger groundwaters drawn from shallower wells²⁹. Conversely, in the Bengal Basin (Bangladesh), groundwaters deeper than ~100 m tend to have lower arsenic concentrations than some shallower groundwaters⁵⁴. Our finding that fossil groundwater reliance is increasing as deep wells become more common may imply concomitant changes in exposure to high arsenic groundwaters; however, we stress that arsenic exposure assessments should be considered on an aquifer-by-aquifer basis, because of the importance of local hydrochemical conditions and hydrostratigraphy in determining groundwater arsenic concentrations⁵⁴. Further, in some of the areas where we show the proportion of wells deeper than 200±100 m to be increasing, it is possible that excessive pumping from deep (semi)confined aquifers may alter aqueous arsenic concentrations in the groundwater; for example, in some areas, pumping has induced leakage of high-arsenic groundwater (or arsenic-mobilizing solutes) from aquitards into adjacent aquifers (e.g., some parts of California's Central Valley⁵⁵ and Vietnam's Mekong Delta⁵⁶). Beyond arsenic, elevated activities of naturally occurring radioisotopes (e.g., ²²⁶Ra, ²²⁸Ra) have been identified in some fossil groundwaters, including those within the Disi Sandstone Aquifer of Jordan⁵⁷, the Saq Aquifer System and the Mega Aquifer System of the Arabian Peninsula^{58,59}, and the Nubian Sandstone Aquifer System of northeastern Africa⁶⁰. Careful consideration of treatment options (e.g., blending, reverse osmosis) may be warranted where these fossil groundwaters are tapped for household use.

(e) Third, elevated groundwater salinity levels can render groundwaters inadequate for drinking and irrigation. Salinization mechanisms are diverse, and different aquifer systems will have unique statistical relationships between groundwater age and salinity depending on the natural

geologic setting, proximity to coastal waters and historical land uses (e.g., irrigation practices; ref. ⁶⁴). Our finding that fossil groundwater use is widespread and increasing has a number of implications for understanding the vulnerability of groundwater users to high salinity groundwaters. In coastal settings, there are multiple aquifer systems where fossil groundwaters are known to exist and groundwater wells tap these deep aquifers (e.g., portions of the North Atlantic Coastal Plain and the Floridan Aquifer System). Pumping water from deep aquifers in these settings can lower hydraulic heads below sea level, rendering these deep fossil aquifers vulnerable to landward incursions of seawater (e.g., North Atlantic Coastal Plain at Cape May, New Jersey⁶²) or upconing of saline water from below (e.g., Floridan Aquifer System at Brunswick, Georgia⁶³). Many deep wells in the US have water levels that lie below sea level, implying some of the deep aquifers that the wells tap may be vulnerable to seawater intrusion⁶⁴. Farther inland, deeper groundwaters are generally more likely to be fossil in their age, yet also more likely to be brackish or saline⁶⁵. The increasing prevalence of deep wells in the majority of our study aquifers implies that groundwater wells may be encroaching on the depths at which some aquifer systems transition from shallow and fresh to deep and brackish conditions, likely limiting the effectiveness of drilling deeper wells indefinitely without concomitant treatment¹⁵.

3.3⁴⁵); (ii) cross-formational mixing (upconing and downwelling) induced as a result of borehole drilling and subsequent pumping (e.g., Diass Aquifer System in Senegal⁴⁶ and the North China Plain⁴⁷); (iii) fast downward flow of recent precipitation along defective well casings to deeper depths where the well is perforated (e.g., Bohemian Cretaceous Basin in the Czech Republic⁴⁸, and the Aleppo and Steppe Basins in Syria⁴⁹); (iv) pumping from wells with long perforated intervals that simultaneously draw groundwater from both shallow and deep depths (e.g., Malm Limestone Aquifer System in Poland⁵⁰); (v) leakage through gaps in impermeable layers ('windows' in aquitards) separating fossil and modern groundwater; or (vi) relatively rapid vertical groundwater flow and mixing along geologic faults that may serve as conduits that connect fossil and modern groundwater. Because shallow contaminated groundwater can be drawn downward by pumping from deeper wells⁵¹, developing data products that quantify not only total groundwater withdrawals in a region but also quantify the depths of the wells from which groundwater is withdrawn will be key to understanding fossil aquifer contamination risk.

(b) Geogenic contamination can arise as groundwater interacts with the mineral skeleton of the geologic formations that it flows through and resides within. For example, arsenic is a common geogenic pollutant that poses a challenge to the provision of fresh drinking water in multiple aquifer systems across the world⁵². Aqueous arsenic concentrations can differ between fossil versus younger groundwater, though the statistical relationships between groundwater age and arsenic concentrations vary among aquifer systems and are critically dependent on hydrostratigraphy, redox conditions, and flow path architectures^{53,54}. For example, in California's Cuyama Valley, samples of fossil groundwater pumped from deep wells tend to have higher arsenic concentrations than samples of younger groundwater drawn from shallower wells³⁰. Conversely, in the Bengal Basin (Bangladesh), groundwater found deeper than ~100 m tends to have lower arsenic concentrations than groundwater found at shallower depths⁵⁵. Because groundwater age tends to increase with depth, we compared tens-of-thousands of dissolved arsenic measurements in shallow and deep wells (Supplementary Fig. 12). In some aquifer systems (n=47 aquifers), shallower wells are contaminated by arsenic (>10 µg/L) more frequently than deeper wells. In other aquifer systems (n=54), deep wells are contaminated by arsenic more frequently than shallower wells. The increasing reliance on fossil water

demonstrated here may lead increase exposure to arsenic in some places, but reduce exposure in other places.

Our finding that fossil groundwater reliance is increasing as deep wells become more common may imply concomitant changes in exposure to high-arsenic groundwater; however, we stress that arsenic exposure assessments should be considered on an aquifer-by-aquifer basis because of the importance of local hydrochemical conditions and hydrostratigraphy in determining groundwater arsenic concentrations⁵⁵. Further, in some of the areas where we show the proportion of newly drilled wells that are deeper than 200±100 m to be increasing, it is possible that excessive pumping from deep (semi)confined aquifers may alter aqueous arsenic concentrations in the groundwater; for example, in some areas, pumping has induced leakage of high-arsenic groundwater (or arsenic-mobilizing solutes) from aquitards into adjacent aquifers (e.g., some parts of California's Central Valley⁵⁶ and Vietnam's Mekong Delta⁵⁷). Beyond arsenic, elevated activities of naturally occurring radioisotopes (e.g., ²²⁶Ra, ²²⁸Ra) have been identified in some samples of fossil groundwater, including those collected from the Disi Sandstone Aquifer of Jordan⁵⁸, the Saq Aquifer System and the Mega Aquifer System of the Arabian Peninsula^{59,60}, and the Nubian Sandstone Aquifer System of northeastern Africa⁶¹. Careful consideration of treatment options (e.g., blending, reverse osmosis) may be warranted in some of the areas where fossil groundwater is tapped for direct household use.

(c) Third, elevated salinity levels can render groundwater inadequate for drinking and irrigation. Salinization mechanisms are diverse, and different aquifer systems will have unique statistical relationships between groundwater age and salinity depending on the natural geologic setting, proximity to coastal waters and historical land uses (e.g., irrigation practices; ref. ⁶²). Our finding that fossil groundwater use is widespread and increasing has a number of implications for understanding the vulnerability of groundwater users to high-salinity levels.

In coastal settings, there are multiple aquifer systems where fossil groundwater has been identified and where wells tap these fossil aquifers (e.g., portions of the North Atlantic Coastal Plain and the Floridan Aquifer System). Pumping water from deep aquifers in these settings can lower hydraulic heads below sea level, rendering these deep fossil aquifers vulnerable to landward incursions of seawater (e.g., North Atlantic Coastal Plain at Cape May, New Jersey⁶³) or upconing of saline water from below (e.g., Floridan Aquifer System at Brunswick, Georgia⁶⁴). Many deep wells in the US have water levels that lie below sea level, implying some of the deep aquifers that the wells tap may be vulnerable to seawater intrusion⁶⁵.

Farther inland, deeper groundwater is generally more likely to be fossil in its age, and also more likely to be brackish or saline⁶⁶.

The increasing prevalence of deep wells in the majority of our study aquifers implies that groundwater wells may be encroaching on the depths at which some aquifer systems transition from shallow-and-fresh to deep-and-brackish conditions, likely limiting the effectiveness of drilling deeper wells indefinitely without concomitant treatment¹⁹. However, we stress that there are also aquifer systems where shallower wells are more likely to pump brackish water than deeper wells; our analysis of hundreds-of-thousands of total dissolved solids measurements identified a dozen such cases, most of which are arid alluvial basins in the western US (Supplementary Fig. 14). The high spatial variability in the statistical relationship between well

water salinity and well depth highlights the importance of considering local hydrogeologic settings and historic land uses when examining connections between increased well drilling into fossil aquifers and the potential threat of salinity to groundwater users.

3.2. Water quantity implications of fossil groundwater use in the US

Unsustainable groundwater use is depleting groundwater stores in numerous aquifer systems around the world⁶⁶⁻⁷⁰, with cascading ramifications for irrigated agriculture and food trade. Around the globe, there are examples of aquifer systems where fossil groundwater pumping coincides with groundwater depletion (e.g., Saq Aquifer System of Saudi Arabia; ref. ⁷⁴). Here, we identify US aquifer systems where wells likely tap fossil groundwater and groundwater levels have declined (e.g., California's Cuyama Valley). We also identify US aquifer systems where wells likely tap fossil groundwater but existing monitoring well networks have not captured concomitant declines in groundwater stores (e.g., Black Warrior River Aquifer System). Our finding that fossil groundwater use hotspots do not always co-occur with groundwater depletion hotspots reinforces the point that fossil groundwater is not necessarily a non-renewable resource (see "Fossil Groundwater" in Table 1 within ref. ¹³).

Unsustainable groundwater use is depleting groundwater stores in numerous aquifer systems around the world⁶⁷⁻⁷¹, with cascading ramifications for irrigated agriculture and food trade. Around the globe, there are examples of aquifer systems where fossil groundwater pumping coincides with groundwater depletion (e.g., Saq Aquifer System of Saudi Arabia; ref. ⁷²). Here, we identify US aquifer systems where wells likely tap fossil groundwater and groundwater levels have declined (e.g., California's Cuyama Valley). We also identify US aquifer systems where wells likely tap fossil groundwater but existing monitoring well networks have not captured concomitant declines in groundwater stores (e.g., Black Warrior River Aquifer System). Our finding that fossil-groundwater-use hotspots do not always co-occur with groundwater-depletion hotspots reinforces the point that the use of fossil groundwater does not have to mean that groundwater use is non-renewable (see "Fossil Groundwater" in Table 1 within ref. ¹⁵). Depletion is a complex process that can be more readily informed by real-time withdrawal measurements (versus estimated withdrawals), in addition to well depth and screen interval information that would allow withdrawal data to be linked back to specific geologic formations. Such information, when combined with groundwater recharge and discharge estimates, can provide a more nuanced understanding of depletion dynamics, which can assist in developing management frameworks.

Although the use of fossil groundwaters are groundwater does not necessarily have to mean that groundwater use is non-renewable resources, fossil groundwaters groundwater does tend to be more common in deep deeper aquifers (Supplementary Fig. 1). Because deep deeper aquifers are more likely to be confined than shallow shallower aquifers, fossil groundwaters are groundwater is disproportionately common in confined aquifers. Pumping groundwater from confined aquifers can have different ramifications on metrics of groundwater quantity (e.g., hydraulic heads, vertical hydraulic gradients) than similar amounts of pumping from unconfined aquifers, because confined aquifers typically have a lower storativity than unconfined aquifers. Sustained pumping from confined aquifers—where fossil groundwaters are groundwater is disproportionately common—can lead to leakage from surrounding geologic formations, substantial declines in hydraulic heads, land subsidence as adjoining confining units are

compressed⁷², or a combination of the aforementioned impacts⁷³, or a combination of the aforementioned impacts.

Formatted: Font: Bold

From the perspective of groundwater quantity management, our research stresses the importance of moving beyond estimates of total groundwater pumping rates to include the depths at which (or geologic formations) groundwater is withdrawn from. To the best of our knowledge, available national-scale groundwater withdrawal estimates are two-dimensional data products that do not provide information about the vertical distribution of groundwater withdrawals across the US. Three-dimensional groundwater withdrawal data could improve our understanding of the short- and long-term impacts of pumping fossil groundwater on hydraulic gradients, cross-formational flows, and land subsidence.

3.43. Fossil groundwater use in the US

Formatted: Font: Not Bold

Sustainably using finite fresh groundwater resources remains key to industrial productivity, irrigated agriculture, and the provision of clean, reliable, and convenient domestic water supplies. Widespread reliance on fossil groundwater (Fig. 2) in the US suggests that safeguarding the quality of deep and fresh fossil groundwaters groundwater is key to modern water provision (Fig. 2). Further, because the prevalence of deep wells tends to be increasing in most areas (Fig. 3), protecting fossil aquifers from overuse (Fig. 4) and pollution will be key to meeting future water demands.

Methods

Datasets analyzed

We meet our objectives by combining four databases: (a) fossil groundwater prevalence determined on the basis of published analyses of groundwater radiocarbon data¹; (b) records of groundwater well depths, construction dates and purposes for millions of wells at continental scale^{19,74}; (c) long-term groundwater level time series recorded by the United States Geological Survey and California's Groundwater Ambient Monitoring and Assessment Program, and (d) a new geodatabase of hydrogeologic study areas (United States Aquifer Database; see Fig. 1a for aquifer system outlines).

Delineating aquifers across the US (Fig. 1)

We delineated boundaries for hundreds of aquifer systems across the United States by examining maps and reading descriptions within local- and regional-scale reports (e.g., United States Geological Survey reports). Methods and specific references consulted when developing the new geodatabase are detailed in Supplementary Note 3.1, which includes an extensive table detailing specific references and approaches applied to delineate each of our study aquifers (Supplementary Table 4).

We term our new aquifer boundary database the United States Aquifer Database (USAD). In places in this text we refer to these delineated, two-dimensional areas (i.e., polygons in Fig. 1a) as 'aquifers', although we stress that these two-dimensional areas are underlain by multiple geologic formations each defined as separate local aquifers or aquitards that together form

'aquifer systems'. This geodatabase includes 440 aquifer systems across the US and addresses several shortcomings of an existing nation-wide aquifer spatial dataset (Supplementary Note 3.1). Our newly delineated US Aquifer Database (USAD) was preferable for our study over the USGS' other databases (e.g., "Principal Aquifers database"; see Supplementary Note 3.2) for four reasons: Specifically, our United States Aquifer Database (USAD):

(i) subdivides broad aquifer systems (e.g., the entire High Plains, which the "Principal Aquifers" database defines as a single aquifer system that is too expansive for locally relevant science) into smaller subareas (e.g., Northern High Plains, Central High Plains, Southern High Plains, Great Bend Prairie, and Equus Beds; Supplementary Fig. 6);

(ii) partitions separate valleys (i.e., basins) that are unlikely to share strong hydraulic connections into separate study areas (e.g., part of the Basin and Range aquifer system delineated in the USGS' "Principal Aquifers" database includes valleys in northwestern Nevada and southeastern Arizona in a single polygon, even though these areas are separated by hundreds of kilometers; USADour US Aquifer Database treats these individual valleys as separate study areas; Supplementary Fig. 7);

(iii) specifies and includes aquifer systems that have been widely studied for more than a century yet remain absent in the "Principal Aquifers" database (e.g., "Darton's Dakota Aquifer"^{73,74}; Supplementary Note 3.1, Supplementary Fig. 8); and,^{75,76} Supplementary Fig. 8); and,

(iv) has been informed by our nation-wide compilation of groundwater well drilling geospatial data⁴⁵¹⁹, meaning these locally relevant hydrogeologic data were available to help guide the delineation of specific aquifers accessed by actual wells (e.g., the 2D extensiveness of relatively deep wells helped us delineate areas where the "Intermediate Aquifer" (part of the broader Floridan Aquifer System) is accessed by wells in southwestern Florida).

Equipped with our new geospatial database of hydrogeologic study areas (Fig. 1a), we analyzed spatiotemporal variations in the prevalence of deep wells (i.e., wells with depths exceeding 200±100 m) and observed groundwater-level fluctuations across these hydrogeologic study areas (see Methods sections "Fossil groundwater accessed across US aquifers (Fig. 2)", "Fossil groundwater accessed more frequently over time (Fig. 3)" and "Groundwater depletion in places where deep wells tap fossil aquifers (Fig. 4)").

Meta-analysis of fossil groundwater prevalence (Fig. 1)

We identified 45 (of ~430)The previous lack of locally relevant and continent-wide aquifer geospatial data necessitated that the scope of previous well completion depth studies¹⁹ was limited to only a few expansive aquifer systems, rather than the novel geodatabase consisting of ~440 aquifer systems delineated here. Pairing these US well drilling data¹⁹ and radioisotope-based fossil groundwater prevalence data¹ to our new aquifer geodatabase (Fig. 1a) and a new method designed to explore well drilling depth changes over time surrounding sites where fossil

water has been identified (see Methods section entitled “Fossil groundwater accessed more frequently over time (Fig. 3)”) distinguish this study from previous works.

Reports of fossil groundwater in the US (Fig. 1)

In determining the proposed well depth threshold of 200±100 m (see Methods subsections to follow), we considered our meta-analysis that documented studies reporting fossil well water (Supplementary Fig. 9) and our age-depth data analysis (Supplementary Fig. 1).

We identified n=114 (of 440 study aquifers) US aquifers where at least one publication has reported that at least some sampled groundwater is more than 12,000 thousand years old (yellow polygons in Fig. 1b; Supplementary Information Note 1.1a). The compiled studies base their interpretation (i.e., that fossil water is present in the aquifer system) on radioisotope measurements, such as ¹⁴C and/or ³⁶Cl. Aquifer systems where fossil water has been identified/documented previously are presented in Fig. 1a (yellow shaded aquifer systems in Fig. 1a).

We also present depth variations in fossil groundwater prevalence, based on approximately four thousand wells in the United States (see methods by in ref. ⁴); fossil groundwaters tend¹ and Supplementary Note 1.2). Fossil groundwater tends to be more common in deeper wells, especially wells with depths exceeding 200±100 m (Supplementary Fig. ~~4~~1). Specifically, more than ~30% of wells with depths at or exceeding 100 m contain detectable fossil water, and more than ~half (48%) of wells with depths at or exceeding 300 m contain detectable fossil water (Supplementary Fig. 1).

Fossil groundwater accessed across US aquifers (Fig. 2)

A ~~key uncertainty~~Uncertainty in our analysis of fossil groundwater reliance ~~is~~derives from the lack of groundwater radiocarbon data in many aquifer systems. However, our analysis of groundwater radiocarbon data that are available makes clear that fossil groundwater is often present in wells that are deeper than 200±100 m (Supplementary Fig. 1). Therefore, we analyzed well completion reports derived from n=64 state and sub-state databases to quantify spatial patterns of deep wells—defined here as those exceeding 200±100 m—across US aquifers (extensive quality control procedures for each well completion database reported by ref. ⁴⁵,¹⁹). For each study aquifer containing at least n=10 wells that met our quality control criteria (see Supplementary Information by ref. ⁴⁵ within ref. ¹⁹), we calculated the fraction of all wells within the aquifer that are deeper than 100 m (Fig. 2a), 200 m (Fig. 2b), or 300 m (Fig. 2c).

We emphasize that well screen interval data are not systematically reported in radioisotope reports nor in well construction reports. As a result, our analyses are based on the total depth of wells. Fossil ~~groundwaters are~~groundwater is common, though not ubiquitous, in wells that are deeper than 200±100 m, so we interpret the prevalence of wells that are deeper than 200±100 m as a proxy for the prevalence of wells that access fossil ~~water~~groundwater. Fossil groundwater occurrence generally increases with depth across many major US aquifers (Supplementary Fig. 1; ref. ⁴¹), but we acknowledge that there are likely aquifer systems where younger ~~waters~~

~~underlie~~groundwater ~~underlies~~ fossil ~~waters~~groundwater (see ~~supplementary information~~Supplementary Information within ref. ¹). These limitations highlight an important research gap in the water science community's collection of groundwater data.

Field Code Changed

Fossil groundwater accessed more frequently over time (Fig. 3)

~~We evaluated temporal variations in the proportion of newly constructed wells that likely tap fossil aquifers following several steps. First, we identified all groundwater well completion records that present both a completion date (e.g., well completed on January 3, 1982) and a depth (e.g., 34 meters below ground). Next, we completed a spatial join to identify all wells within a single aquifer system (as defined in our US Aquifer Database: USAD). Then, for any calendar year within which at least 5 well completion records exist, we calculated the [proportion of all newly constructed wells that have a depth exceeding a 'threshold depth'] (where 'threshold depth' is 100 m, 200 m or 300 m, a set of threshold depths that encompass the broad range at which many aquifer systems transition from young water (shallow) to fossil water (deep)). Last, we completed non-parametric regressions of the rank transforms of [well completion year] versus the rank transforms of [the proportion of all newly constructed wells that have a depth exceeding a 'threshold depth'].~~

~~We only consider cases where all of the following criteria are met: (a) at least one calendar year met our criteria for analysis (i.e., at least five drilled wells) in the first five years of a studied time interval (e.g., for analyses of the time interval 1950–1975, we require at least one of the following five calendar years to meet our criteria for analysis: 1950, 1951, 1952, 1953 or 1954), (b) at least one calendar year met our criteria for analysis (i.e., at least five drilled wells) in the final five years of a studied time interval; and, (c) a minimum of at least five calendar years within the time interval met our criteria for analyses. The correlations were determined for five time intervals: (i) 1950–1975, (ii) 1975–2000, (iii) 2000–2015, (iv) 1950–2015, and (v) 1975–2015. Positive Spearman rank correlation coefficients (ρ) imply that the proportion of drilled wells that are deeper than 100 m, 200 m, or 300 m (many of which are also likely pump fossil water) has increased over time (Fig. 3). Negative Spearman rank correlation coefficients imply that the proportion of drilled wells that are deeper than 100 m, 200 m, or 300 m has declined over time (Fig. 3).~~

~~To test if the proportion of newly drilled wells in the US that tap fossil aquifers has increased over time we completed a series of steps. First, (i) we mapped the locations of wells identified in ref. ¹ to pump fossil water (minimal fraction of well water comprised of fossil groundwater exceeds zero). Second, we identified all records of well construction within a 20 km radius of the well that is known to pump fossil groundwater. Third, (iii) for each well construction event, we determined whether the total depth of the constructed well is shallower or deeper than the nearby 'fossil well'. That is, we compared the depth of each newly drilled well to that of the nearby 'fossil well', and describe the former in binary terms: (a) newly drilled well is shallower than the fossil well, or (b) newly drilled well is deeper than the fossil well. Fourth, (iv) for each calendar year where at least five wells were constructed within 20 km of the fossil well within a given year, we calculated the proportion of wells drilled in that year that are deeper than the fossil well.~~

implying that many of these wells likely also pump fossil water, since they are deeper than a nearby well known to draw some fossil water (i.e., we calculated the fraction of wells drilled deeper than the well that is known to pump some fossil water for a given year). Fifth, (v) we calculated the Spearman rank correlation that describes variations in the fraction of wells that are deeper than the well that is known to pump fossil water varies versus calendar year (see Supplementary Fig. 2 for schematic). Last, we determine spatial statistics of these increasing and decreasing trends for a number of aquifer systems across the contiguous United States (Fig. 3).

Groundwater depletion in places where deep wells tap fossil aquifers (Fig. 4)

We compiled groundwater-level monitoring data from the United States Geological Survey. The downloaded data includes 16 million water level measurements from 611,445 unique groundwater wells within US Aquifer Database (USAD) areas. These data enabled us to evaluate temporal groundwater-level fluctuations. ~~We did so for over~~ three unique time intervals: (i) 1950-1975, (ii) 1975-2000, and (iii) 2000-2015; ~~our goal is to test for spatial correspondence between groundwater depletion and the existence of deep wells that likely tap fossil aquifers.~~

To test for spatial correspondence between groundwater level declines and the existence of deep wells that likely tap fossil aquifers, we completed a series of steps. First, we calculated the average water level for any unique year with at least one water level measurement for every monitoring well in our database. Next, we filtered our dataset by considering only the monitoring wells that met both of these criteria: (a) at least five unique years within which at least one water level measurement was recorded within a given time interval, and (b) at least one water level measurement within both the first and the last five years of the time interval (e.g., for the time interval 1950-1975, we require at least one measurement between 1950-1955 and at least one measurement between 1970-1975 for us to consider the monitoring well in our analyses). For each monitoring well meeting these criteria, we calculated the Sen slope of “calendar year” versus “average water level for a given calendar year”. For each aquifer (i.e., each USADUS Aquifer Database polygon), we calculated the median Sen slope among all monitoring wells meeting our criteria for analyses located within the aquifer (Figs. 4a-c). We also calculated the fraction of all monitoring wells within an aquifer with a Sen slope indicative of deepening groundwater levels over time (Figs. 4d-f). Last, we compared these Sen slopes (i.e., groundwater-level variability through time) with the prevalence of wells exceeding 200±100 m (see results section entitled “Fossil groundwater use hotspots do not always co-occur with groundwater depletion hotspots”).

Limitations to our results due to the lack of adequate groundwater age data

Our methodology ~~has a number of limitations. Foremost, because of~~ is limited by the lack of widespread groundwater radioisotope measurements in deep and shallow wells (e.g., ^{14}C , ^{36}Cl); Therefore, we cannot easily resolve the depth below which most stored groundwater is fossil in age for each of our 437440 study aquifers. Consequently, our analysis of fossil groundwater reliance depends on the use of well depths as a proxy for fossil water access, with the implicit assumption that deeper wells are more likely to draw fossil water than shallower wells. In an

Formatted: Space Before: 0 pt

effort to overcome this data limitation, we report the fraction of wells deeper than a wide range of threshold depths: 100 m to 300 m. We also emphasize that, for some aquifers aquifer systems, fossil water may not dominate at depths of ~300 m (the deepest limit applied to our study (e.g. ref. 75,77), the deepest limit applied to our study).

References

1. Jasechko, S. *et al.* Global aquifers dominated by fossil groundwaters but wells vulnerable to modern contamination. *Nat. Geosci.* **10**, 425–429 (2017).
2. Yokochi, R. *et al.* Radiokrypton unveils dual moisture sources of a deep desert aquifer. *Proc. Natl. Acad. Sci.* **116**, 16222–16227 (2019).
3. Harkness, J. S. *et al.* Pre-drill Groundwater Geochemistry in the Karoo Basin, South Africa. *Groundwater* **56**, 187–203 (2018).
4. Lavastre, V. *et al.* Establishing constraints on groundwater ages with ³⁶Cl, ¹⁴C, ³H, and noble gases: A case study in the eastern Paris basin, France. *Appl. Geochem.* **25**, 123–142 (2010).
5. Elliot, T., Andrews, J. N. & Edmunds, W. M. Hydrochemical trends, palaeorecharge and groundwater ages in the fissured Chalk aquifer of the London and Berkshire Basins, UK. *Appl. Geochem.* **14**, 333–363 (1999).
6. Bentley, H. W. *et al.* Chlorine 36 dating of very old groundwater: 1. The Great Artesian Basin, Australia. *Water Resour. Res.* **22**, 1991–2001 (1986).
7. Matsumoto, T. *et al.* Application of combined ⁸¹Kr and ⁴He chronometers to the dating of old groundwater in a tectonically active region of the North China Plain. *Earth Planet. Sci. Lett.* **493**, 208–217 (2018).
8. Aurelia, A., Silva, K. E., Rebouças, A. da C., Marlucia, M. & Santiago, F. ¹⁴C Analyses of Groundwater From the Botucatu Aquifer System in Brazil. *Radiocarbon* **31**, 926–933 (1989).

9. Gonzalez Hita, L., Sanchez Diaz, F. & Mata Arellano, I. Estudio hidrogeoquímico e isotópico del acuífero granular de la Comarca Lagunera, México. In Estudios de hidrología isotópica en América Latina 1994. Vienna Austria Int. At. Energy Agency IAEA-TECDOC-835 237–276 (1995).
10. Ayadi, R. *et al.* Hydrogeological and hydrochemical investigation of groundwater using environmental isotopes (^{18}O , ^2H , ^3H , ^{14}C) and chemical tracers: a case study of the intermediate aquifer, Sfax, southeastern Tunisia. *Hydrogeol. J.* **26**, 983–1007 (2018).
11. Ransom, K. M. *et al.* A hybrid machine learning model to predict and visualize nitrate concentration throughout the Central Valley aquifer, California, USA. *Sci. Total Environ.* **601–602**, 1160–1172 (2017).
12. ~~Jasechko, S. Global Isotope Hydrogeology—Review. *Rev. Geophys.* **57**, 835–965 (2019)~~ Chambers, L. A., White, I., Ferguson, J., Radke, B. M. & Evans, W. R. Brine Chemistry of Groundwater Discharge Zones in the Murray Basin, Australia. *Water Resour. Res.* **31**, 1343–1353 (1995).
- ~~13-13.~~ Mir, R., Azizyan, G., Massah, A. & Gohari, A. Fossil water: Last resort to resolve long-standing water scarcity? *Agric. Water Manag.* **261**, 107358 (2022).
14. Jasechko, S. Global Isotope Hydrogeology—Review. *Rev. Geophys.* **57**, 835–965 (2019).
15. Margat, J., Foster, S. & Droubi, A. Concept and importance of non-renewable resources, in Non-Renewable Groundwater Resources: A Guidebook on Socially-Sustainable Management for Water-Policy Makers, vol. 10, edited by S. Foster and D. P. Loucks, pp. 13–24, U. N. Educ., Sci. and Cult. Organ., Paris. 97 (2006).
1416. Bierkens, M. F. P. & Wada, Y. Non-renewable groundwater use and groundwater depletion: a review. *Environ. Res. Lett.* **14**, 063002 (2019).
- ~~15-17~~ ~~—Perrone, D. & Jasechko, S. Deeper well drilling an unsustainable stopgap to groundwater depletion. *Nat. Sustain.* **3**, 773–782 (2019).~~

- ~~46~~. Ferguson, G., Cuthbert, M. O., Befus, K., Gleeson, T. & McIntosh, J. C. Rethinking groundwater age. *Nat. Geosci.* **13**, 592–594 (2020).
- ~~17~~~~18~~. Gleeson, T., Cuthbert, M., Ferguson, G. & Perrone, D. Global Groundwater Sustainability, Resources, and Systems in the Anthropocene. *Annu. Rev. Earth Planet. Sci.* **48**, 431–463 (2020).
- ~~18~~~~19~~. Perrone, D. & Jasechko, S. Deeper well drilling an unsustainable stopgap to groundwater depletion. *Nat. Sustain.* **2**, 773–782 (2019).
- ~~Perrone, D. & Jasechko, S. Dry groundwater wells in the western United States. *Environ. Res. Lett.* **12**, 104002 (2017).~~
- ~~19~~~~20~~. Pétré, M.-A., Rivera, A., Lefebvre, R., Hendry, M. J. & Fohnagy, A. J. B. A unified hydrogeological conceptual model of the Milk River transboundary aquifer, traversing Alberta (Canada) and Montana (USA). *Hydrogeol. J.* **24**, 1847–1871 (2016).
- ~~20~~~~21~~. Musgrove, M., Beck, J. A., Paschke, S., Bauch, N. J. & Mashburn, S. L. *Quality of groundwater in the Denver Basin aquifer system, Colorado, 2003-5*. 123 <http://pubs.er.usgs.gov/publication/sir20145051> (2014).
- ~~21~~~~22~~. Iles, D. L. & Rich, T. *Examination of isotopes in selected waters in eastern South Dakota*. 38 <http://www.sdgs.usd.edu/pubs/pdf/RI-118.pdf> (2017).
- ~~22~~~~23~~. Stotler, R., Harvey, F. E. & Gosselin, D. C. A Black Hills-Madison Aquifer Origin for Dakota Aquifer Groundwater in Northeastern Nebraska. *Groundwater* **48**, 448–464 (2010).
- ~~23~~~~24~~. Clark, J. F., Davisson, M. L., Hudson, G. B. & Macfarlane, P. A. Noble gases, stable isotopes, and radiocarbon as tracers of flow in the Dakota aquifer, Colorado and Kansas. *J. Hydrol.* **211**, 151–167 (1998).
- ~~24~~~~25~~. Aeschbach-Hertig, W., Stute, M., Clark, J. F., Reuter, R. F. & Schlosser, P. A paleotemperature record derived from dissolved noble gases in groundwater of the Aquia Aquifer (Maryland, USA). *Geochim. Cosmochim. Acta* **66**, 797–817 (2002).

- 2526. Clark, J. F., Stute, M., Schlosser, P., Drenkard, S. & Bonani, G. A tracer study of the Floridan Aquifer in southeastern Georgia: Implications for groundwater flow and paleoclimate. *Water Resour. Res.* **33**, 281–289 (1997).
- 2627. Lee, R. W. Geochemistry of Ground Water in the Southeastern Coastal Plain Aquifer System in Mississippi, Alabama, Georgia, and South Carolina. 81 (1993).
- 2728. Castro, M. C. & Goblet, P. Noble gas thermometry and hydrologic ages: Evidence for late Holocene warming in Southwest Texas. *Geophys. Res. Lett.* **30**, (2003).
- 2829. Kulongoski, J. T., Hilton, D. R. & Izbicki, J. A. Helium isotope studies in the Mojave Desert, California: implications for groundwater chronology and regional seismicity. *Chem. Geol.* **202**, 95–113 (2003).
- 2930. Everett, R. *et al.* Geology, Water-Quality, Hydrology, and Geomechanics of the Cuyama Valley Groundwater Basin, California, 2008–12: U.S. Geological Survey Scientific Investigations Report 2013–5108. <https://pubs.usgs.gov/sir/2013/5108/> (2013).
- 3031. Davisson, M. L. & Criss, R. E. Stable isotope imaging of a dynamic groundwater system in the southwestern Sacramento Valley, California, USA. *J. Hydrol.* **144**, 213–246 (1993).
- 3132. Robson, S. G. & Banta, E. R. *Ground Water Atlas of the United States: Segment 2, Arizona, Colorado, New Mexico, Utah*. C1C32 <http://pubs.er.usgs.gov/publication/ha730C> (1995).
- 3233. Smith, W. B., Miller, G. R. & Sheng, Z. Assessing aquifer storage and recovery feasibility in the Gulf Coastal Plains of Texas. *J. Hydrol. Reg. Stud.* **14**, 92–108 (2017).
- 3334. Konikow, L. F. & Neuzil, C. E. A method to estimate groundwater depletion from confining layers. *Water Resour. Res.* **43**, (2007).
- 3435. Williams, L. J. & Kuniansky, E. L. *Revised hydrogeologic framework of the Floridan aquifer system in Florida and parts of Georgia, Alabama, and South Carolina*. 156 <http://pubs.er.usgs.gov/publication/pp1807> (2015).

- ~~35~~36. Masterson, J. P. *et al.* *Hydrogeology and hydrologic conditions of the Northern Atlantic Coastal Plain aquifer System from Long Island, New York, to North Carolina*. 88
<http://pubs.er.usgs.gov/publication/sir20135133> (2013).
- ~~36~~37. Mallory, M. J. *Hydrogeology of the southeastern coastal plain aquifer system in parts of eastern Mississippi and western Alabama. Regional aquifer-system analysis. Southeastern coastal plain. Professional paper*. <https://www.osti.gov/biblio/161972> (1993).
- ~~37~~38. Stamos, C. L., Martin, P., Nishikawa, T. & Cox, B. F. *Simulation of Ground-Water Flow in the Mojave River Basin, California*. (2001).
- ~~38~~. Page, R. W. *Base and thickness of the Post-Eocene continental deposits in the Sacramento Valley, California*. <http://pubs.er.usgs.gov/publication/wri7345> (1974).
- ~~39~~-~~39~~. EIS/EIR. *Environmental Water Account: Draft Environmental Impact Statement Environmental Impact Report*. 317 <https://usbr.gov/mp/ewa/docs/v1-draft-enviro-impact-statement-environmental-impact-report.pdf> (2003).
- ~~40~~. Macfarlane, P. A. *An analysis of the upper part of the regional flow system along the southern ground-water flow "corridor" in the Dakota aquifer using a steady-state, vertical profile flow model*. http://www.kgs.ku.edu/Hydro/Publications/1996/OF96_1d/index.html (1996).
- ~~40~~41. Hosman, R. L. & Weiss, J. S. *Geohydrologic units of the Mississippi embayment and Texas coastal uplands aquifer systems, south-central United States*. <http://pubs.er.usgs.gov/publication/pp1416B> (1991).
- ~~41~~42. Katz, B. G., Chelette, A. R. & Pratt, T. R. Use of chemical and isotopic tracers to assess nitrate contamination and ground-water age, Woodville Karst Plain, USA. *J. Hydrol.* **289**, 36–61 (2004).
- ~~42~~43. Reaver, K. M. *et al.* Drinking Water Quality and Provision in Six Low-Income, Peri-Urban Communities of Lusaka, Zambia. *GeoHealth* **5**, e2020GH000283 (2021).

- 4344. Han, D., Currell, M. J. & Cao, G. Deep challenges for China's war on water pollution. *Environ. Pollut.* **218**, 1222–1233 (2016).
- 4445. Burg, A., Zilberbrand, M. & Yechieli, Y. Radiocarbon Variability in Groundwater in an Extremely Arid Zone—The Arava Valley, Israel. *Radiocarbon* **55**, 963–978 (2013).
- 4546. Madioune, D. H. *et al.* Application of isotopic tracers as a tool for understanding hydrodynamic behavior of the highly exploited Diass aquifer system (Senegal). *J. Hydrol.* **511**, 443–459 (2014).
- 4647. Kreuzer, A. M. *et al.* A record of temperature and monsoon intensity over the past 40 kyr from groundwater in the North China Plain. *Chem. Geol.* **259**, 168–180 (2009).
- 4748. Jiráková, H., Huneau, F., Hrkal, Z., Celle-Jeanton, H. & Le Coustumer, P. Carbon isotopes to constrain the origin and circulation pattern of groundwater in the north-western part of the Bohemian Cretaceous Basin (Czech Republic). *Appl. Geochem.* **25**, 1265–1279 (2010).
- 4849. Stadler, S., Geyh, M. A., Ploethner, D. & Koeniger, P. The deep Cretaceous aquifer in the Aleppo and Steppe basins of Syria: assessment of the meteoric origin and geographic source of the groundwater. *Hydrogeol. J.* **20**, 1007–1026 (2012).
- 4950. Zuber, A., Weise, S. M., Motyka, J., Osenbrück, K. & Rózański, K. Age and flow pattern of groundwater in a Jurassic limestone aquifer and related Tertiary sands derived from combined isotope, noble gas and chemical data. *J. Hydrol.* **286**, 87–112 (2004).
- 5051. Levy, Z. F. *et al.* Critical Aquifer Overdraft Accelerates Degradation of Groundwater Quality in California's Central Valley During Drought. *Geophys. Res. Lett.* **48**, e2021GL094398 (2021).
- 5152. Podgorski, J. & Berg, M. Global threat of arsenic in groundwater. *Science* **368**, 845–850 (2020).
- 5253. Fendorf, S., Michael, H. A. & Geen, A. van. Spatial and Temporal Variations of Groundwater Arsenic in South and Southeast Asia. *Science* (2010).
- 5354. Burgess, W. G. *et al.* Vulnerability of deep groundwater in the Bengal Aquifer System to contamination by arsenic. *Nat. Geosci.* **3**, 83–87 (2010).

- 5455. Hoque, M. A., Burgess, W. G., Shamsudduha, M. & Ahmed, K. M. Delineating low-arsenic groundwater environments in the Bengal Aquifer System, Bangladesh. *Appl. Geochem.* **26**, 614–623 (2011).
- 5556. Smith, R., Knight, R. & Fendorf, S. Overpumping leads to California groundwater arsenic threat. *Nat. Commun.* **9**, 2089 (2018).
- 5657. Erban, L. E., Gorelick, S. M., Zebker, H. A. & Fendorf, S. Release of arsenic to deep groundwater in the Mekong Delta, Vietnam, linked to pumping-induced land subsidence. *Proc. Natl. Acad. Sci.* **110**, 13751–13756 (2013).
- 5758. Vengosh, A. *et al.* High Naturally Occurring Radioactivity in Fossil Groundwater from the Middle East. *Environ. Sci. Technol.* **43**, 1769–1775 (2009).
- 5859. Faraj, T., Ragab, A. & El Alfy, M. Geochemical and hydrogeological factors influencing high levels of radium contamination in groundwater in arid regions. *Environ. Res.* **184**, 109303 (2020).
- 5960. Sultan, M. *et al.* Assessment of age, origin, and sustainability of fossil aquifers: A geochemical and remote sensing-based approach. *J. Hydrol.* **576**, 325–341 (2019).
- 6061. Sherif, M. I. & Sturchio, N. C. Elevated radium levels in Nubian Aquifer groundwater of Northeastern Africa. *Sci. Rep.* **11**, 78 (2021).
- 6162. Foster, S. *et al.* Impact of irrigated agriculture on groundwater-recharge salinity: a major sustainability concern in semi-arid regions. *Hydrogeol. J.* **26**, 2781–2791 (2018).
- 6263. Lacombe, P. J. & Carleton, G. B. *Hydrogeologic Framework, Availability of Water Supplies, and Saltwater Intrusion, Cape May County, New Jersey*. (U.S. Department of the Interior, U.S. Geological Survey, 2002).
- 6364. Cherry, G. S. & Peck, M. *Saltwater intrusion in the Floridan aquifer system near downtown Brunswick, Georgia, 1957–2015*. <http://pubs.er.usgs.gov/publication/ofr20171010> (2017).

- 6465. Jasechko, S., Perrone, D., Seybold, H., Fan, Y. & Kirchner, J. W. Groundwater level observations in 250,000 coastal US wells reveal scope of potential seawater intrusion. *Nat. Commun.* **11**, 3229 (2020).
- 6566. Stanton, J. S. *et al.* *Brackish groundwater in the United States*. 202
<http://pubs.er.usgs.gov/publication/pp1833> (2017).
- 6667. Shah, T., Molden, D., Sakthivadivel, R. & Seckler, D. Global Groundwater Situation: Opportunities and Challenges. *Econ. Polit. Wkly.* **36**, 4142–4150 (2001).
- 6768. Konikow, L. F. & Kendy, E. Groundwater depletion: A global problem. *Hydrogeol. J.* **13**, 317–320 (2005).
- 6869. Wada, Y. Modeling Groundwater Depletion at Regional and Global Scales: Present State and Future Prospects. *Surv. Geophys.* **37**, 419–451 (2016).
- 6970. Rodell, M. *et al.* Emerging trends in global freshwater availability. *Nature* **557**, 651–659 (2018).
- 7071. Dalin, C., Wada, Y., Kastner, T. & Puma, M. J. Groundwater depletion embedded in international food trade. *Nature* **543**, 700–704 (2017).
- 7172. Othman, A. & Abotalib, A. Z. Land subsidence triggered by groundwater withdrawal under hyper-arid conditions: case study from Central Saudi Arabia. *Environ. Earth Sci.* **78**, 243 (2019).
- 7273. Herrera-García, G. *et al.* Mapping the global threat of land subsidence. *Science* (2021).
- 74. Perrone, D. & Jasechko, S. Dry groundwater wells in the western United States. *Environ. Res. Lett.* **12**, 104002 (2017).
- 7375. Darton, N. H. *Geology and Underground Waters of South Dakota*. (Department of the Interior, United States Geological Survey, U.S. Government Printing Office, 1909).
- 7476. Bredehoeft, J. D., Neuzil, C. E. & Milly, P. C. D. *Regional Flow in the Dakota Aquifer: A Study of the Role of Confining Layers*. (U.S. Government Printing Office, 1983).

7577. Eastoe, C. J., Gu, A. & Long, A. The origins, ages and flow paths of groundwater in Tucson basin: results of a study of multiple isotope systems. *Groundw. Recharge Desert Environ. Southwest. U. S.* 217–234 (2004).

Acknowledgments. This material is based upon work supported by the National Science Foundation under Grant No. EAR-2048227.

Author Contributions. M.G.G. wrote the first draft of the manuscript and led the statistical analyses. S.J. delineated aquifer system boundaries. M.G.G., D.P., S.J. co-developed methods, discussed results and wrote the manuscript.

Data Availability. Delineated aquifer system boundaries are presented as a Supplementary Dataset (shapefile). [**note: ~~these~~ link to the geospatial dataset (posted to a public data sharing repository e.g., Dataverse, FigShare) will be included and posted as a supplementary file made available should our work be accepted for publication; for peer review purposes only, the geospatial data can be accessed via the following confidential hyperlink <https://app.box.com/s/o7vot58cja1452651ivi8j3jgoxbxyxj>*]

Reviewers' Comments:

Reviewer #1:

Remarks to the Author:

The authors have responded positively to my comments and recommendations and have changed the manuscript accordingly, even including additional analyses. I congratulate the authors with this paper and advise acceptance as is.

Reviewer #2:

Remarks to the Author:

I would like to commend the authors on a very thorough response to my comments. I had several significant concerns with the first submission, and I appreciate that the authors have done considerable work to respond to all of my comments. In particular, I would like to highlight the following:

- I was concerned about the novelty of the manuscript with respect to the authors previous publications. In response they have clarified the specific novelty much more clearly and have also added additional quantitative analysis that does a better job of setting this work apart.
- I was also concerned that the discussion section was not sufficiently supported by the results and the authors have added multiple quantitative analyses to better support the assertions they were making. This includes additional water quality analysis as well as better analysis of spatial trends slopes and connections to groundwater drawdown hot spots.
- The meta analysis has been framed much more clearly and the authors have revised language to be more clear on what this analysis can and can't show both in response to my comments and the comments from reviewer 1.

I only have one remaining small comment. In response to reviewer 1 the authors have done a more thorough job explaining all the reasons why we might care about fossil groundwater in the introduction and in the discussion. However, in the abstract only water quality is listed. In the following sentence:

"Because fossil groundwater often has distinct concentrations of contaminants to younger groundwater, mapping where wells tap fossil aquifers is critical for evaluating contaminant exposure."

Also, the focus on 'contaminant exposure' implies that fossil groundwaters have higher level of contaminants, but as the authors have clearly outlined they are likely just different contaminants as shallow groundwater contamination can have different sources.

I suggest the authors consider revising this sentence to be more general and better match the rest of the manuscript.

Reviewer #1 (Remarks to the Author):

The authors have responded positively to my comments and recommendations and have changed the manuscript accordingly, even including additional analyses. I congratulate the authors with this paper and advise acceptance as is.

We thank Reviewer #1 for their helpful and constructive comments. We feel that our manuscript was improved substantially from their input.

Reviewer #2 (Remarks to the Author):

I would like to commend the authors on a very thorough response to my comments. I had several significant concerns with the first submission, and I appreciate that the authors have done considerable work to respond to all of my comments. In particular, I would like to highlight the following:

- I was concerned about the novelty of the manuscript with respect to the authors previous publications. In response they have clarified the specific novelty much more clearly and have also added additional quantitative analysis that does a better job of setting this work apart.
- I was also concerned that the discussion section was not sufficiently supported by the results and the authors have added multiple quantitative analyses to better support the assertions they were making. This includes additional water quality analysis as well as better analysis of spatial trends slopes and connections to groundwater drawdown hot spots.
- The meta analysis has been framed much more clearly and the authors have revised language to be more clear on what this analysis can and can't show both in response to my comments and the comments from reviewer 1.

Thank you to Reviewer #2; your input and encouragement helped us to take our analyses further, and we feel that our manuscript was strengthened by your comments.

I only have one remaining small comment. In response to reviewer 1 the authors have done a more thorough job explaining all the reasons why we might care about fossil groundwater in the introduction and in the discussion. However, in the abstract only water quality is listed. In the following sentence:

“Because fossil groundwater often has distinct concentrations of contaminants to younger groundwater, mapping where wells tap fossil aquifers is critical for evaluating contaminant exposure.”

Also, the focus on ‘contaminant exposure’ implies that fossil groundwaters have higher level of contaminants, but as the authors have clearly outlined they are likely just different contaminants as shallow groundwater contamination can have different sources.

I suggest the authors consider revising this sentence to be more general and better match the rest of the manuscript.

Thank you. We agree that a more general framing is appropriate. We have modified our abstract to now read as follows:

“Mapping where wells tap fossil aquifers is relevant for water quality and quantity management.”